# PATTERN-GUIDED DIFFUSION MODELS

## ABSTRACT

Diffusion models have shown promise in forecasting future data from multivariate time series. However, few existing methods account for recurring structures, or patterns, that appear within the data. We present Pattern-Guided Diffusion Models (PGDM), which leverage inherent patterns within temporal data for forecasting future time steps. PGDM first extracts patterns using archetypal analysis and estimates the most likely next pattern in the sequence. By guiding predictions with this pattern estimate, PGDM makes more realistic predictions that fit within the set of known patterns. We additionally introduce a novel uncertainty quantification technique based on archetypal analysis, and we dynamically scale the guidance level based on the pattern estimate uncertainty. We apply our method to two well-motivated forecasting applications, predicting visual field measurements and motion capture frames. On both, we show that pattern guidance improves PGDM's performance (MAE / CRPS) by up to 40.67% / 56.26% and 14.12% / 14.10%, respectively. PGDM also outperforms baselines by up to 65.58% / 84.83% and 93.64% / 92.55%.

## 1 INTRODUCTION

Diffusion models are a class of generative models that perform generation by iteratively removing noise from a noisy sample. These models are easier to train and generate higher quality images compared to the previous state-of-the-art, generative adversarial networks (Dhariwal & Nichol, 2021). Recent work has found success in using diffusion models to forecast future steps of temporal data (Chang et al., 2024; Feng et al., 2024; Gu et al., 2022; Hu et al., 2024; Li et al., 2022; Lv et al., 2024; Rasul et al., 2021; Wen et al., 2023). Such methods, however, rarely leverage the recurring structures that often manifest in temporal data. These appear, for example, in medical modalities due to the physiology and anatomy of the human body. Basketball videos also contain repeated structures due to standardized courts, player positions, and strategies. The few diffusion-based forecasters that exploit these *patterns* often overlook changes over time and uncertainties in pattern representation (Wang et al., 2024; Westny et al., 2024; Zhao et al., 2024).

In this paper, we present Pattern-Guided Diffusion Models (PGDM) for forecasting temporal data with inherent patterns. Using archetypal analysis (Cutler & Breiman, 1994), we extract patterns from training data, then train a guidance function to predict future pattern contributions to the data. PGDM then forecasts future points guided by these predictions. To handle evolving patterns, we introduce a novel uncertainty metric that dynamically tunes the scale of pattern guidance.

We evaluate PGDM on two impactful applications. First, we consider the clinical application of visual field (VF) prediction. Visual field tests measure a patient's functional vision, and the resulting measurements manifest common patterns across patients due to eye anatomy. Furthermore, forecasting future VF measurements can serve as a decision aid for clinicians. On a real-world VF dataset, we find that pattern guidance improves the performance (MAE / CRPS) of PGDM by up to 40.67% / 56.26% on average, surpassing baseline models by up to 65.58% / 84.83% on average. Next, we consider the application of forecasting future motion capture frames for human motion prediction. Pose patterns frequently appear in common human movements, such as walking and running. Predicting human motion may aid advancements in human robot collaboration and autonomous driving. On motion capture frames for a variety of dance genres, we show that PGDM is able to leverage even the diverse, highly dynamic patterns that present in dance motion. Pattern guidance improves the performance (MAE / CRPS) of PGDM by up to 14.12% / 14.10% on average, allowing PGDM to surpass baselines by up to 93.64% / 92.55% on average.

In summary, our contributions are as follows.

1. We present Pattern-Guided Diffusion Models (PGDM), which leverage inherent patterns within temporal data.

2. We introduce a novel uncertainty quantification method based on archetypal analysis, and we show that this uncertainty metric captures geometric distance from the training set.

3. We show that the proposed uncertainty quantification metric approximately lower bounds the error of the pattern predictions that guide PGDM.

4. We propose a method to dynamically tune the level of pattern guidance based on the proposed uncertainty metric.

## 2 RELATED WORKS

Diffusion models have been widely used for temporal forecasting. TimeGrad (Rasul et al., 2021) and LDT (Feng et al., 2024) forecast multivariate time series conditioned on histories, while BVAE (Li et al., 2022) uses a bi-directional VAE for the reverse diffusion process. Models like USTD (Hu et al., 2024) and DiffSTG (Wen et al., 2023) capture spatial dependencies using spatio-temporal graphs. Other applications include pedestrian trajectory (Lv et al., 2024; Gu et al., 2022) and medical sensor signal prediction (Chang et al., 2024). For a comprehensive overview, see Yang et al. (2024).

Few such diffusion-based forecasters attempt to leverage patterns within the data. Hypothesizing that past patterns likely reappear later, Diff-MGR (Zhao et al., 2024) conditions predictions on previous patterns. Westny et al. (2024) also proposed to guide predictions of traffic trajectories using patterns, formalized as environment maps, as agent behaviors are often dictated by the environment (e.g., cars stay within the road lanes). Similarly to Diff-MGR, Westny et al. (2024) assumes that manifested patterns will remain constant over time, as predictions are conditioned on a fixed map. Wang et al. (2024) instead proposed to guide predictions with dynamically changing patterns, captured by segmented real-time camera readings, as humans are most likely to walk towards specific destinations within a scene, such as a door, stairs, or a hallway. However, these approaches do not account for uncertainty in the guiding patterns. In contrast, our PGDM model adapts the level of pattern guidance based on the estimated reliability of dynamically evolving patterns.

## 3 BACKGROUND

Let the data of interest be $x \in \mathbb{R}^d$ sampled from distribution $p(x)$, which arrives in a temporal sequence $\{x_t\}$ with time index $t$. We are concerned with such data that contains *patterns*, or repeating structures. Given an observed history of length $T$ over time $t \in \{1, 2, \ldots, T\}$, we aim to predict a horizon of length $H$ over time $t \in \{T + 1, T + 2, \ldots, T'\}$ with $T' = T + H$. Denote a set of $n$ of history and horizon pairs by $\{x_{1:T}^i, x_{T:T'}^i\}_{i=1}^n$. We use archetypal analysis, which identifies patterns resembling real data rather than an abstraction, to overcome the challenge of extracting useful patterns. Here, we provide a brief overview of diffusion models and archetypal analysis.

### 3.1 DIFFUSION MODELS

Diffusion models (Ho et al., 2020; Sohl-Dickstein et al., 2015) aim to learn and generate data from a distribution $p(x)$ through a forward and reverse process, in which noise is iteratively added to and removed from the data, respectively. Given $x_{t,0} \sim p(x_t)$ at time $t$, the fixed $S$-step forward process creates a sequence of increasingly noisy samples $x_{t,1}, x_{t,2}, \ldots, x_{t,S}$. Note that here we use the notation $x_{t,s}$, where $t$ denotes the time index and $s$ denotes the diffusion step. The noisy samples are drawn from the distribution $q(x_{t,s}|x_{t,s-1}) := \mathcal{N}(\sqrt{1 - \beta_s} x_{t,s-1}, \beta_s \mathbf{I}, )$, where $\beta_1, \beta_2, \ldots, \beta_S$ is a noise variance schedule. With appropriately chosen variance schedule, this distribution approaches a standard normal as $S \to \infty$. Conveniently, the noising step $x_{t,s}$ can be sampled in closed form given $x_{t,0}$, $q(x_{t,s}|x_{t,0}) = \mathcal{N}(\sqrt{\bar{\alpha}_s} x_{t,0}, (1 - \bar{\alpha}_s)\mathbf{I})$, where $\alpha_s = 1 - \beta_s$ and $\bar{\alpha}_s = \prod_{i=1}^s \alpha_i$.

Conversely, the reverse process removes noise from $x_{t,S} \sim \mathcal{N}(\mathbf{0}, \mathbf{I})$ to ultimately recover the training distribution. The goal is to learn the distribution $p_\theta(x_{t,s-1}|x_{t,s}) := \mathcal{N}(\mu_\theta(x_{t,s}, s), \sigma_s^2 \mathbf{I})$. The denoising parameters $\theta$ are learned by optimizing the evidence lower bound (ELBO) on negative

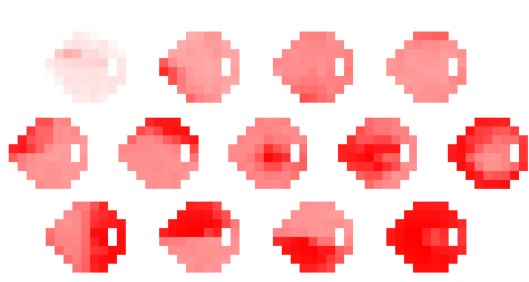
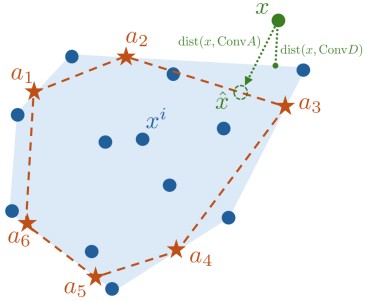

(a) Example visual field archetypes.

(b) Geometric interpretation.

Figure 1: Overview of archetypal analysis (AA). (a) Example archetypes extracted from a visual field dataset. The archetypes capture visual loss patterns that consistently appear across glaucoma patients. Darker regions indicate greater vision loss. (b) Given a dataset $D$ (blue dots), AA identifies a set of archetypes $A$ (red stars). Within $\mathrm{Conv}A \subseteq \mathrm{Conv}D$ (within red dashed line), any point can be reconstructed from $A$ without error. For any point $x \notin \mathrm{Conv}A$, the reconstruction error is the distance between $x$ and $\mathrm{Conv}A$.

log likelihood, $-\log p_\theta\left(x_{t,0}\right)$. At inference time, samples from the learned distribution $p_\theta\left(x_{t,0}\right)$ are generated by applying the reverse process over $S$ steps to noisy samples.

## 3.2 ARCHETYPAL ANALYSIS

Archetypal analysis (AA) (Cutler & Breiman, 1994) extracts extremal patterns, or archetypes, from a given dataset. These archetypes are themselves a combination of the data and therefore are realistic and interpretable representations of the data's significant patterns. Figure 1a shows examples extracted from a visual field dataset. Furthermore, any point within the given dataset can be constructed as a combination of these archetypes, allowing for the contribution of each pattern to be quantified.

More formally, given dataset $D = \left\{x^i\right\}_{i=1}^n$, AA finds the $p$ archetypes $a_1, a_2, \ldots a_p \in \mathbb{R}^d$ that minimize the residual sum of squares (RSS) error,

$$\min_{c_j^i, a_j^i \ \forall i,j} \sum_{i=1}^n \left\| x^i - \sum_{j=1}^p c_j^i a_j \right\|^2, \tag{1}$$

subject to $c_j^i \geq 0 \ \forall j$ and $\sum_{j=1}^p c_j^i = 1$ for each $i \in \{1, 2, \ldots, n\}$, and where $a_j = \sum_{k=1}^n \beta_j^k x^k$ with $\beta_j^k \geq 0 \ \forall k$ and $\sum_{k=1}^n \beta_j^k = 1$ for each $j \in \{1, , 2, \ldots, p\}$. That is, if zero reconstruction error is achieved, then each $x^i \in D$ can be reconstructed as a convex combination of the archetypes, and the archetypes are themselves a convex combination of the data. Then, given a set of identified archetypes, any new data point can be reconstructed as a convex combination of the archetypes, with coefficients found by minimizing the objective in equation 1 with fixed $a_1, a_2, \ldots, a_p$.

Figure 1b visualizes a geometric interpretation of AA. Intuitively, AA identifies a region in which any point can be perfectly reconstructed by the archetypes. This region is the convex hull of the archetypes, denoted $\mathrm{Conv}A$ for the archetype set $A$. In the ideal case when $p = n$, the set of archetypes is exactly $D$, and the region of reconstructible points fully encompasses the dataset. In practice, $p < n$ is typically chosen, and the resulting archetypes instead lie on the boundary of $\mathrm{Conv}D$ (Cutler & Breiman, 1994, Proposition 1). The archetype set therefore defines a reconstructible region that closely, but often not fully, captures the dataset $D$. Furthermore, for any $x \notin \mathrm{Conv}A$, the reconstruction error can be thought of as the distance between $x$ and $\mathrm{Conv}A$. In Section 4.2, we show that this reconstruction error can be used to estimate the distance between $x$ and the dataset $D$.

## 4 METHODS

Our goal is to learn the conditional distribution $p(x_{T:T'}|x_{1:T}, \hat{P}_{T:T'})$, where $\hat{P}_{T,T'}$ represents the patterns estimated to manifest in the future. To determine a pattern representation space, we first extract archetype set $A$ from the training data. The representation space is the contribution of archetypes to each data point. We then train our proposed Pattern-Guided Diffusion Model (PGDM) to learn the desired conditional distribution.

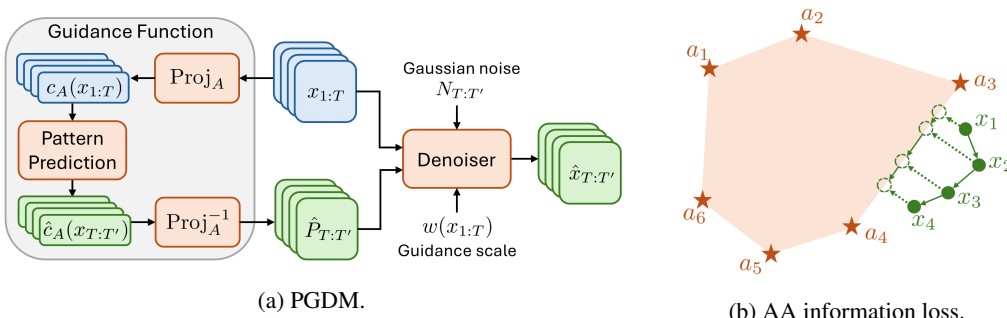

(a) PGDM.               (b) AA information loss.

Figure 2: (a) Overview of Pattern-Guided Diffusion Models. A pattern guidance function estimates the future patterns over a horizon. This prediction is performed in the archetype space, where the projected representation captures the contribution of each pattern to the data. The predicted future patterns are supplied as conditioning context to the diffusion model. The scale of the pattern guidance is dynamically tuned based on the uncertainty of the input sequence. (b) Projection into the lower-dimensional archetype space loses information for any point outside the convex hull of the archetypes $\text{Conv}A$. In this example, the sequence $x_1, x_2, x_3, x_4 \notin \text{Conv}A$ is perturbed to the boundary of $\text{Conv}A$ after reconstruction, losing important dynamical information.

Figure 2a summarizes PGDM, which is constructed from three key components. 1) A pattern guidance function predicts the patterns $\hat{P}$ that will appear over the future horizon. First, the input sequence $x_{1:T}$ is projected to the archetype space, where the projected pattern representation $c_A(x_{1:T})$ quantifies the contribution of each pattern to the data. The projected representation for the future steps is then predicted, and the resulting estimate $\hat{c}_A(x_{T:T'})$ is lifted back to the original data space as a predicted pattern $\hat{P}_{T:T'}$. 2) While prediction in the lower $p$-dimensional archetype space helps to evade the curse of dimensionality, the projection operation naturally incurs an information loss. By design, while any point $x_t \in \text{Conv}A$ can be projected with no information loss, this cannot be said for a point $x_t \notin \text{Conv}A$, which may appear when the data's patterns change over time. Figure 2b visualizes this loss of information. We therefore introduce a novel Archetypal Analysis Uncertainty Quantification (AAUQ) technique, which we show captures geometric distance from the training set. This uncertainty metric also estimates a lower bound on the loss of the guidance function. 3) We use the predicted pattern from the guidance function as additional conditioning context for the diffusion model, following the general methodology of classifier-free diffusion guidance (Ho & Salimans, 2022). To account for uncertainties in the guidance function, we dynamically tune the guidance scale based on our uncertainty metric. We now describe each of the three components in further detail.

## 4.1 GUIDANCE FUNCTION

First, we employ archetypal analysis to extract significant patterns from the training data. For convenience, when the meaning is clear, we overload the notation $A$ to indicate both the resulting set of $p < d$ archetypes $\{a_1, a_2, \ldots, a_p\} \subset \mathbb{R}^d$ and the matrix constructed from these archetypes $A \in \mathbb{R}^{d \times p}$. These archetypes define an archetype, or pattern, space in $p$ dimensions. Given archetypes $A$, let the projection function that determines the pattern representation (i.e., the coefficients minimizing objective equation 1) be $c_A : \mathbb{R}^d \to \mathbb{R}^p$. For any point $x_t$, the reconstruction is $\hat{x}_t = A c_A(x_t)$. We denote the error of this reconstruction as $L_{c_A}(x_t) = \|x_t - A c_A(x_t)\|$.

Next, we train a lightweight neural network $f_A : \mathbb{R}^{p \times T} \to \mathbb{R}^{p \times H}$ to predict $H$ future pattern representations based on $T$ past pattern representations. The error of this prediction function is $L_{f_A}(c_{1:T}) = \|A c_{T:T'} - A f_A(c_{1:T})\|$.

The guidance function $f_G : \mathbb{R}^{d \times T} \to \mathbb{R}^{d \times H}$ is

$$f_G(x_{1:T}) = A f_A \circ c_A(x_{1:T}). \tag{2}$$

Note that $f_G(x_{1:T})$ is $\hat{P}_{T:T'}$ in Figure 2a. We now show that a lower bound on the guidance function error $L_{f_G}$ is a function of the projection error $L_{c_A}$ and pattern prediction error $L_{f_A}$.

**Theorem 1** (Bound on Guidance Function Error). *For any sequence $x_{1:T}$ and horizon $x_{T:T'}$ pair, let the error of guidance function $f_G$ defined in Equation equation 2 be $L_{f_G}(x_{1:T}) =*

$\|x_{T:T'} - A f_A (c_A(x_{1:T}))\|$. *Then*

$$L_{f_G}(x_{1:T}) \geq L_{c_A}(x_{T:T'}) - L_{f_A}(x_{1:T}). \tag{3}$$

*Proof.* Please see Appendix A for the proof. □

It is clear from Theorem 3 that, if the prediction function $f_A$ has a reasonably low error, then the error of the projection function $c_A$ can serve as an approximate lower bound for the error of the guidance function $f_G$. Therefore, an estimate of $L_{c_A}$ may quantify the degree to which PGDM should "trust" the guidance function, allowing the level of pattern guidance to be dynamically tuned. Next, we introduce a novel uncertainty quantification technique that can be used as a proxy for $L_{c_A}$.

### 4.2 UNCERTAINTY QUANTIFICATION FOR ARCHETYPAL ANALYSIS

Projection to the lower-dimensional archetype space may lose information. We therefore introduce a novel uncertainty quantification metric based on archetypal analysis that captures this loss.

**Definition 1** (Archetypal Analysis Uncertainty Quantification). *For any sequence $x_{1:T}$ and archetype set $A$, the uncertainty $u_A$ of the archetype projection is*

$$u_A(x_{1:T}) = \tfrac{1}{T} \sum_{t=1}^{T} \|x_t - A c_A(x_t)\|. \tag{4}$$

Intuitively, this Archetypal Analysis Uncertainty Quantification (AAUQ) metric is simply the average reconstruction loss of the history sequence. AAUQ can also be geometrically interpreted as estimating the average distance of $x_{1:T}$ from the training dataset.

**Theorem 2** (AAUQ as Geometric Distance). *Assume that a set of archetypes $A = \{a_j\}_{j=1}^{p}$ is extracted from a dataset $D$. Define $d$ as the closest point in $\mathrm{Conv}D$ to $x_t$. For any $x_t$,*

$$u_A(x_t) - \delta \leq \mathrm{dist}(x_t, \mathrm{Conv}D) \leq u_A(x_t) + \delta, \tag{5}$$

*where $\delta = \|A c_A(x_t) - d\|$ and $\delta = 0$ when $p = n$.*

*Proof.* Please see Appendix B for the proof. □

**Remark 1.** *In Theorem 2, $\delta$ captures the ability of the archetypes to express the dataset $D$. This can be seen in the proof of Theorem 2. This can also be understood from the example in Figure 1b, in which $\mathrm{dist}(x, \mathrm{Conv}A)$ is exactly $u_A(x)$, and $\delta$ is the distance between $\hat{x}$ and the point on the boundary of $\mathrm{Conv}D$. This has interesting implications for the geometric interpretation of AAUQ in the case that the archetypes perfectly reconstruct the data, which occurs when the number of archetypes is equal to the size of the dataset. It is clear from Theorem 2 that if $p = n$,*

$$u_A(x_t) = \mathrm{dist}(x_t, \mathrm{Conv}D). \tag{6}$$

### 4.3 PATTERN-GUIDED DIFFUSION MODELS

PGDM predicts future sequences $x_{T:T'}$ conditioned on the pattern prediction from the guidance function. We follow the methodology of classifier-free diffusion guidance, in which the model is trained with conditioning dropout. This effectively learns two denoising models, $\epsilon_\theta(z_{T:T',s}, x_{1:T}, \hat{P}_{T:T'})$ and $\epsilon_\theta(z_{T:T',s}, x_{1:T}, \emptyset)$, where $\emptyset$ is a null value and $z_{T:T',s}$ is the sample to be denoised at diffusion step $s$. Algorithm 1 summarizes the training process of PGDM. For history and horizon sequences sampled in Line 2, the guidance conditioning is set to the pattern prediction or a null value in Line 4. With the loss function on Line 5, the denoising model learns to estimate the noise added to $x_{T:,T'}$.

When generating samples with traditional classifier-free guidance, each denoising step uses a linear combination of the conditional and unconditional predictions:

$$\hat{\epsilon}_\theta(z_{T:T',s}, x_{1:T}) = w\epsilon_\theta(z_{T:T',s}, x_{1:T}, \hat{P}_{T:T'}) + (1-w)\epsilon_\theta(z_{T:T',s}, x_{1:T}, \emptyset),$$

where $w \geq 0$ is the *guidance scale*. When $w = 0$, generation is unguided and sampled data are more diverse. The guidance level increases with $w$, leading to less diverse but higher quality samples. While traditional classifier-free guidance is performed with constant guidance scale $w$, we instead use a dynamic guidance scale $w(x_{1:T})$ that captures the trustworthiness of the guidance function:

$$\hat{\epsilon}_\theta(z_{T:T',s}, x_{1:T}) = w(x_{1:T})\epsilon_\theta(z_{T:T',s}, x_{1:T}, \hat{P}_{T:T'}) + (1-w(x_{1:T}))\epsilon_\theta(z_{T:T',s}, x_{1:T}, \emptyset), \tag{7}$$

---

**Algorithm 1** PGDM Training

---

1: **for** all epochs **do**
2:     Sample $x_{1:T}$, $x_{T:T'}$ from the training set
3:     $s \sim \text{Uniform}(1, \ldots, S)$, $\epsilon \sim \mathcal{N}(\mathbf{0}, \mathbf{I})$
4:     $\hat{P}_{T:T'} = \emptyset$ with probability $p_{\text{drop}}$, else $\hat{P}_{T:T'} = f_G(x_{1:T})$ from eqn. equation 2
5:     Take gradient descent step on $\nabla_\theta \| \epsilon - \epsilon_\theta(\sqrt{\overline{\alpha}_s} x_{T:T'} + \sqrt{1 - \overline{\alpha}_s} \epsilon, \; x_{1:T}, \; \hat{P}_{T:T'}) \|^2$
6: **end for**

---

**Algorithm 2** PGDM Inference

---

1: **given** $x_{1:T}$
2: $x_{T:T',S} \sim \mathcal{N}(\mathbf{0}, \mathbf{I})$
3: $\hat{P}_{T:T'} = f_G(x_{1:T})$ from eqn. equation 2
4: **for** s = S, ..., 1 **do**
5:     $n \sim \mathcal{N}(\mathbf{0}, \mathbf{I})$ if s > 1, else $n = 0$
6:     Compute $\hat{\epsilon}_\theta(z_{T:T',s}, \; x_{1:T})$ from eqn. equation 7
7:     Sample $x_{T:T',s-1} = \frac{1}{\sqrt{\alpha_s}} \left( x_{T:T',s} - \frac{1 - \alpha_s}{\sqrt{1 - \overline{\alpha}_s}} \right) \hat{\epsilon}_\theta(z_{T:T',s}, \; x_{1:T}) + \sqrt{\beta_s} n$
8: **end for**
9: Compute $w^* = w(x_{1:T}, \overline{w}^*, \gamma)$ from eqn. equation 8, for $\overline{w}^* \in [0, 1]$
10: Mix $\hat{x}_{T:T'} = w^* \hat{P}_{T:T'} + (1 - w^*) x_{T:T',0}$

---

**Definition 2** (Dynamic Guidance Scale). *Let $A$ be the set of archetypes extracted from the training dataset. Given a maximum guidance scale $\overline{w}$ and maximum tolerable uncertainty $\gamma$, the dynamic guidance scale for sequence $x_{1:T}$ is*

$$w(x_{1:T}, \overline{w}, \gamma) = \text{ReLU}\left( -\frac{\overline{w}}{\gamma} u_A(x_{1:T}) + \overline{w} \right). \tag{8}$$

Here, we measure the trustworthiness of our guidance function by our AAUQ uncertainty metric. The uncertainty $u_A(x_{1:T})$ is easy to compute and in practice, we find that the $u_A(x_{1:T})$ is a good proxy for estimating $L_{c_A}(x_{T:T'})$ and therefore the lower bound in Theorem 1. Hence, we design the dynamic guidance scale so that, as uncertainty increases, the guidance scale $w \in (0, \overline{w})$ decreases. When the uncertainty exceeds $\gamma$, w = 0. In other words, PGDM follows the pattern guidance most strictly when the data is in-distribution. For out-of-distribution data with unseen patterns, PGDM relies less on pattern-guidance, reverting to a standard diffusion model in the extreme case.

Algorithm 2 summarizes the inference process of PGDM. In Line 2, noisy data is sampled from a standard normal distribution. In Line 3, the patterns of the future sequence are predicted. Then, over $S$ reverse diffusion steps in Lines 4–7, the guided and unguided denoising models are combined with dynamic guidance scale to iteratively remove noise. The resulting sequence prediction is sampled from the learned distribution, which we expect to be tightly centered around the pattern prediction. Finally in Lines 9 and 10, we mix the raw pattern prediction with the pattern-guided sequence prediction using a dynamic mixing scale with maximum value $\overline{w}^*$. We include this mixing step to mitigate some potential practical challenges of PGDM (see Appendix C for a full discussion).

## 5 APPLICATIONS

We validate PGDM on two applications, visual field prediction and human motion prediction. To help demonstrate the benefits of pattern guidance, we show the performance of two PGDM models selected from our hyperparameter search. The first model, $\text{PGDM}_{\text{MAE}}$, achieved the lowest validation mean absolute error (MAE) with pattern guidance. The second, $\text{PGDM}_{\text{GDE}}$, achieved the highest capacity for pattern guidance, or the greatest achievable improvement in MAE by using guided predictions ($\overline{w} > 0$) over unguided predictions ($\overline{w} = 0$). That is, we selected $\text{PGDM}_{\text{GDE}}$ as the model that achieved the largest gain in MAE by adding any level of guidance in the range of 1 to 5.

For these two models, we compare performance with guidance to performance without guidance and multiple baselines. While we would ideally compare PGDM to baselines that use some pattern

conditioning (Wang et al., 2024; Westny et al., 2024; Zhao et al., 2024), most existing methods formalize patterns in a manner that does not translate to our applications. For others, we were unable to obtain sufficient implementation details or code. Therefore, we instead select more general baseline techniques that do not use pattern conditioning. We compare PGDM to multiple diffusion-based baselines for forecasting, TimeGrad (Rasul et al., 2021), CSDI (Tashiro et al., 2021), and ARMD (Gao et al., 2025). While CSDI is a data imputation technique, it can easily be extended for forecasting. For the visual field application, we further compare PGDM to GenViT (Yang et al., 2022), a diffusion model with a vision transformer backbone that has been applied to VF prediction by Tian et al. (2023). Finally, we perform multiple ablations, one serving as a proxy for evaluating Westny et al. (2024).

We evaluate all models using mean absolute error (MAE). Additionally, to evaluate PGDM as a probabilistic forecaster, we measure the continuous ranked probability score (CRPS). Following Rasul et al. (2021) and Tashiro et al. (2021), we report $CRPS_{SUM}$, computed as the CRPS over the summed dimensions of the data.

Source code is supplied in the supplementary material. Complete implementation details for pattern extraction and training, including hyperparameter selection, model architectures, compute resources, parameter counts, and inference latency are provided in Appendix D.

**Visual Field Prediction.** Pattern-Guided Diffusion models are especially useful in medical settings, where data often reflects consistent patterns due to anatomy. For instance, 24-2 visual field (VF) tests measure light sensitivity in decibels (dB) at 52 central points of vision, with specific loss patterns linked to structural eye damage (e.g., nerve fiber bundle loss) (Keltner et al., 2003). Figure 1a illustrates archetypal patterns from VF data, where darker areas indicate reduced vision. Forecasting VF outcomes can support clinicians in diagnosis, progression identification, and treatment planning.

We evaluate PGDM on the public the University of Washington Humphrey Visual Field (UWHVF) dataset (Montesano et al., 2022). To the best of our knowledge, UWHVF is the only publicly available 24-2 VF dataset, containing 7,428 sequences from 3,871 patients. The UWHVF measurements capture the patient's light sensitivity compared to normative data, ranging from -38 dB to 50 dB. That is, a negative (positive) dB indicates worse (better) vision than typical. Due to the few follow-up visits per patient, we predict $H = 1$ step into the future based on the past $T = 3$ steps. Additionally, as VF measurements are taken at non-constant time increments, we also condition predictions on the recorded age at each VF measurement and the desired time horizon for prediction. Thus, the one-step-ahead prediction can be made for an arbitrary length time period. We create multiple forecasting sequences from each patient in a sliding window fashion, resulting in 6,171 sequences.

**Human Motion Prediction.** We also apply PGDM to predict future frames of human motion capture data, a task relevant to domains including human robot interaction and autonomous driving (Lyu et al., 2022). Human motion often involves repeated body positions when executing common movements (e.g., walking, running, dancing). While our visual field application demonstrates PGDM's utility in the clinical domain, the motion prediction application presents a more challenging task with more rapidly evolving signals and longer prediction horizons.

We evaluate PGDM on the AIST++ dataset (Li et al., 2021), which contains motion capture frames capturing 3D motion from 10 dance genres. Predicting dance motion is more challenging for PGDM compared to locomotion, as the variety of dance genres and styles leads to a rich set of patterns with less periodic progressions over time. The AIST++ dataset captures the skeleton with 3D pose data for 17 keypoints, which represent specific joints or locations in the body. For consistency across heights, we normalize all data to the scale [0, 100]. We predict $H = 5$ steps into the future based on a past sequence of $T = 3$ steps. We create multiple forecasting sequences from each motion capture video, resulting between $36,739$ and $105,504$ sequences across the 10 genres.

## 5.1 RESULTS

**Pattern Extraction.** We extract $p = 13$ archetypal patterns from UWHVF, shown in Figure 1a. We extract between $p = 12$ and $p = 22$ archetypes for each genre of AIST++. Figure 3 shows the archetypes extracted from the break dancing frames (see Appendix E for remaining genres). Appendix F reports the reconstruction error of the extracted patterns and the guidance function error.

**AAUQ approximately lower bounds the guidance function error.** Motivated by Theorem 1, PGDM uses AAUQ to determine the appropriate level of guidance. In Figure 4, we compare AAUQ

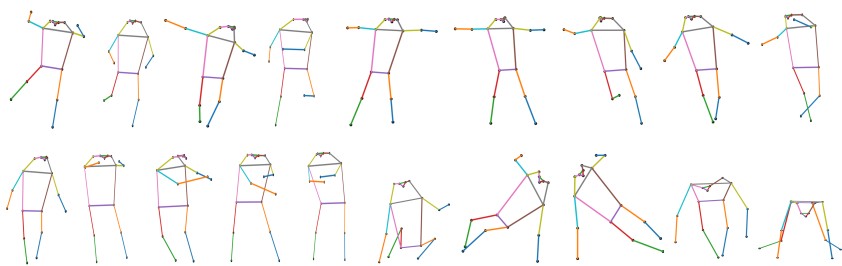

Figure 3: Nineteen archetypes extracted from AIST++ break dancing frames.

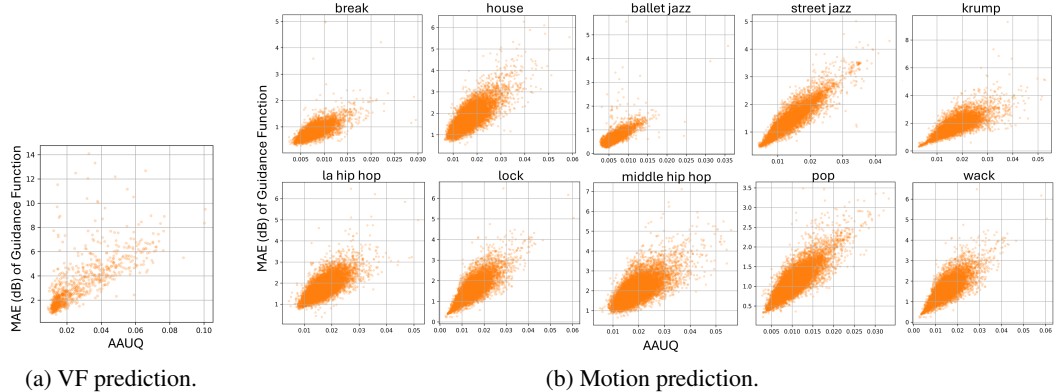

(a) VF prediction.    (b) Motion prediction.

Figure 4: AAUQ approximately lower bounds the guidance function error for both applications.

measurements to the MAE of the guidance function. In both applications, we observe that AAUQ is indeed proportional to a linear lower bound on the guidance function error.

**Pattern guidance improves predictions.** Tables 1 and 2 report the MAE and CRPS of $PGDM_{MAE}$ and $PGDM_{GDE}$ with and without pattern guidance. Compared to unguided predictions, pattern guidance reduces prediction error and CRPS significantly. In particular, on UWHVF, guidance reduces the MAE / CRPS of $PGDM_{GDE}$ by up to 40.67% / 56.26%. On AIST++, guidance achieves a reduction of up to 14.12% / 14.10%. To further study the impact of pattern guidance, we also show in Table 3 the MAE of $PGDM_{MAE}$ and $PGDM_{GDE}$ on UWHVF with $\overline{w} = 1, 2, 3, 4, 5$. Similar results on AIST++ and with CRPS, as well as qualitative examples, are shown in Appendix G. In general, the standard deviation of MAE decreases as the guidance scale increases towards the optimal $\overline{w}$ choice, indicating that guidance improves both prediction consistency and quality. Notably, we also find that excessive pattern guidance may lead to diminishing returns. In practice, an appropriate $\overline{w}$ may be selected by a hyperparameter search.

**PGDM outperforms baselines.** Tables 1 and 2 also compare the performance of $PGDM_{MAE}$ and $PGDM_{GDE}$ to our baselines. Across the board, $PGDM_{MAE}$ with pattern guidance achieves significantly lower MAE than baselines. On UWHVF, $PGDM_{MAE}$ with guidance surpasses GenViT, TimeGrad, CSDI, and ARMD by up to 65.68%, 29.36%, 7.20%, and 24.30%, respectively. On AIST++, $PGDM_{MAE}$ surpasses TimeGrad, CSDI, and ARMD by up to 82.60%, 36.71%, 93.64%. In terms of CRPS, on AIST++, $PGDM_{MAE}$ with guidance outperforms TimeGrad, CSDI, and ARMD by up to 82.87%, 40.86%, and 92.55%, respectively. On UWHVF, PGDM outperforms GenViT and ARMD by up to 84.83% and 15.38%, but does not outperform TimeGrad and CSDI. We note that all models achieve significantly higher CRPS on UWHVF than AIST++, indicating that this small dataset provides limited signal for learning well-calibrated predictive distributions. We emphasize that PGDM achieves the lowest MAE on UWHVF, demonstrating that explicit pattern modeling improves point prediction even when the distribution itself is challenging to estimate.

**PGDM can outperform baselines with guidance alone.** We note that $PGDM_{MAE}$ achieves lower MAE than baselines even without guidance ($\overline{w} = 0$) on some AIST++ genres (e.g., break dancing). In these cases, to better demonstrate that PGDM's performance comes from pattern guidance, rather than

Table 1: Mean absolute error of PGDM$_{\text{MAE}}$, PGDM$_{\text{GDE}}$, and baselines for both the visual field prediction (UWHVF dataset) and the human motion prediction (10 dance genres from AIST++ dataset) applications. For PGDM$_{\text{MAE}}$ and PGDM$_{\text{GDE}}$ with $\overline{w} > 0$, we show the guidance and mixing scale result that achieved the lowest error (for $\overline{w}$ and $\overline{w}^*$ choices, see Appendix D). Mean and standard deviation are taken across five samples.

| Visual Field Prediction | | | | | | | |
|---|---|---|---|---|---|---|---|
| GenViT | TimeGrad | CSDI | ARMD | PGDM$_{\text{GDE}}$ $\overline{w}=0$ | PGDM$_{\text{GDE}}$ $\overline{w}>0$ | PGDM$_{\text{MAE}}$ $\overline{w}=0$ | PGDM$_{\text{MAE}}$ $\overline{w}>0$ |
| 8.61$_{\pm0.0018}$ | 4.19$_{\pm0.0327}$ | 3.19$_{\pm0.0421}$ | 3.91$_{\pm0.0083}$ | 5.20$_{\pm0.0407}$ | 3.08$_{\pm0.0153}$ | 3.75$_{\pm0.0437}$ | 2.96$_{\pm0.0117}$ |

| Human Motion Prediction | | | | | | | |
|---|---|---|---|---|---|---|---|
| Genre | TimeGrad | CSDI | ARMD | PGDM$_{\text{GDE}}$ $\overline{w}=0$ | PGDM$_{\text{GDE}}$ $\overline{w}>0$ | PGDM$_{\text{MAE}}$ $\overline{w}=0$ | PGDM$_{\text{MAE}}$ $\overline{w}>0$ |
| Break | 2.10$_{\pm0.0064}$ | 0.47$_{\pm0.0010}$ | 4.26$_{\pm0.0026}$ | 0.52$_{\pm0.0032}$ | 0.45$_{\pm0.0015}$ | 0.41$_{\pm0.0013}$ | 0.38$_{\pm0.0009}$ |
| House | 3.71$_{\pm0.0159}$ | 1.02$_{\pm0.0045}$ | 9.11$_{\pm0.0035}$ | 0.90$_{\pm0.0013}$ | 0.82$_{\pm0.0014}$ | 0.79$_{\pm0.0017}$ | 0.74$_{\pm0.0014}$ |
| Ballet Jazz | 1.32$_{\pm0.0098}$ | 0.55$_{\pm0.0054}$ | 3.38$_{\pm0.0011}$ | 0.49$_{\pm0.0004}$ | 0.42$_{\pm0.0009}$ | 0.42$_{\pm0.0008}$ | 0.38$_{\pm0.0002}$ |
| Street Jazz | 1.65$_{\pm0.0102}$ | 0.56$_{\pm0.0054}$ | 7.57$_{\pm0.0049}$ | 0.60$_{\pm0.0016}$ | 0.54$_{\pm0.0005}$ | 0.52$_{\pm0.0017}$ | 0.48$_{\pm0.0008}$ |
| Krump | 2.37$_{\pm0.0067}$ | 0.77$_{\pm0.0017}$ | 8.90$_{\pm0.0046}$ | 0.88$_{\pm0.0016}$ | 0.79$_{\pm0.0005}$ | 0.77$_{\pm0.0016}$ | 0.70$_{\pm0.0013}$ |
| LA Hip Hop | 3.30$_{\pm0.0157}$ | 0.78$_{\pm0.0023}$ | 8.51$_{\pm0.0032}$ | 0.90$_{\pm0.0009}$ | 0.81$_{\pm0.0009}$ | 0.80$_{\pm0.0010}$ | 0.74$_{\pm0.0006}$ |
| Lock | 3.03$_{\pm0.0086}$ | 0.76$_{\pm0.0028}$ | 7.65$_{\pm0.0031}$ | 0.78$_{\pm0.0017}$ | 0.71$_{\pm0.0023}$ | 0.72$_{\pm0.0011}$ | 0.67$_{\pm0.0006}$ |
| Mid. Hip Hop | 3.35$_{\pm0.0113}$ | 1.04$_{\pm0.0048}$ | 8.98$_{\pm0.0051}$ | 1.05$_{\pm0.0034}$ | 0.95$_{\pm0.0038}$ | 0.88$_{\pm0.0014}$ | 0.82$_{\pm0.0009}$ |
| Pop | 2.55$_{\pm0.0105}$ | 0.70$_{\pm0.0053}$ | 6.43$_{\pm0.0019}$ | 0.52$_{\pm0.0013}$ | 0.48$_{\pm0.0018}$ | 0.47$_{\pm0.0008}$ | 0.44$_{\pm0.0018}$ |
| Wack | 1.03$_{\pm0.0047}$ | 0.44$_{\pm0.0042}$ | 3.39$_{\pm0.0026}$ | 0.49$_{\pm0.0054}$ | 0.45$_{\pm0.0079}$ | 0.44$_{\pm0.0044}$ | 0.39$_{\pm0.0028}$ |

Table 2: CRPS$_{\text{SUM}}$ (lower is better) of PGDM$_{\text{MAE}}$, PGDM$_{\text{GDE}}$, and baselines for both the visual field prediction (UWHVF dataset) and the human motion prediction (10 dance genres from AIST++ dataset) applications. For PGDM$_{\text{MAE}}$ and PGDM$_{\text{GDE}}$ with $\overline{w} > 0$, we show the guidance and mixing scale result that achieved the lowest error (for $\overline{w}$ and $\overline{w}^*$ choices, see Appendix D). Mean and standard deviation are taken across three seeds.

| Visual Field Prediction | | | | | | | |
|---|---|---|---|---|---|---|---|
| GenViT | TimeGrad | CSDI | ARMD | PGDM$_{\text{GDE}}$ $\overline{w}=0$ | PGDM$_{\text{GDE}}$ $\overline{w}>0$ | PGDM$_{\text{MAE}}$ $\overline{w}=0$ | PGDM$_{\text{MAE}}$ $\overline{w}>0$ |
| 5.119$_{\pm0.0004}$ | 0.751$_{\pm0.0020}$ | 0.658$_{\pm0.0025}$ | 0.918$_{\pm0.0029}$ | 1.887$_{\pm0.0119}$ | 0.825$_{\pm0.0032}$ | 0.794$_{\pm0.0122}$ | 0.777$_{\pm0.0040}$ |

| Human Motion Prediction | | | | | | | |
|---|---|---|---|---|---|---|---|
| Genre | TimeGrad | CSDI | ARMD | PGDM$_{\text{GDE}}$ $\overline{w}=0$ | PGDM$_{\text{GDE}}$ $\overline{w}>0$ | PGDM$_{\text{MAE}}$ $\overline{w}=0$ | PGDM$_{\text{MAE}}$ $\overline{w}>0$ |
| Break | 0.189$_{\pm0.0010}$ | 0.051$_{\pm0.0001}$ | 0.490$_{\pm0.0006}$ | 0.049$_{\pm0.0001}$ | 0.043$_{\pm0.0001}$ | 0.039$_{\pm0.0001}$ | 0.037$_{\pm<0.0001}$ |
| House | 0.340$_{\pm0.0011}$ | 0.106$_{\pm0.0002}$ | 0.679$_{\pm0.0008}$ | 0.085$_{\pm0.0005}$ | 0.079$_{\pm0.0003}$ | 0.077$_{\pm0.0004}$ | 0.073$_{\pm0.0002}$ |
| Ballet Jazz | 0.117$_{\pm0.0005}$ | 0.053$_{\pm0.0002}$ | 0.346$_{\pm0.0005}$ | 0.044$_{\pm0.0002}$ | 0.039$_{\pm0.0002}$ | 0.039$_{\pm0.0001}$ | 0.035$_{\pm0.0002}$ |
| Street Jazz | 0.184$_{\pm0.0002}$ | 0.058$_{\pm0.0003}$ | 0.448$_{\pm0.0004}$ | 0.055$_{\pm0.0002}$ | 0.052$_{\pm0.0001}$ | 0.050$_{\pm0.0002}$ | 0.047$_{\pm0.0002}$ |
| Krump | 0.298$_{\pm0.0010}$ | 0.093$_{\pm0.0003}$ | 0.911$_{\pm0.0003}$ | 0.091$_{\pm0.0004}$ | 0.078$_{\pm0.0003}$ | 0.079$_{\pm0.0003}$ | 0.071$_{\pm0.0003}$ |
| LA Hip Hop | 0.331$_{\pm0.0004}$ | 0.088$_{\pm0.0002}$ | 0.752$_{\pm0.0014}$ | 0.083$_{\pm0.0007}$ | 0.077$_{\pm0.0005}$ | 0.075$_{\pm0.0006}$ | 0.071$_{\pm0.0005}$ |
| Lock | 0.367$_{\pm0.0004}$ | 0.080$_{\pm0.0002}$ | 0.652$_{\pm0.0014}$ | 0.073$_{\pm0.0001}$ | 0.066$_{\pm0.0001}$ | 0.068$_{\pm0.0002}$ | 0.063$_{\pm0.0001}$ |
| Mid. Hip Hop | 0.368$_{\pm0.0018}$ | 0.109$_{\pm0.0002}$ | 0.832$_{\pm0.0005}$ | 0.097$_{\pm0.0003}$ | 0.089$_{\pm0.0003}$ | 0.085$_{\pm0.0002}$ | 0.081$_{\pm0.0003}$ |
| Pop | 0.252$_{\pm0.0004}$ | 0.076$_{\pm0.0005}$ | 0.460$_{\pm0.0001}$ | 0.051$_{\pm0.0002}$ | 0.047$_{\pm0.0002}$ | 0.046$_{\pm0.0002}$ | 0.045$_{\pm0.0002}$ |
| Wack | 0.112$_{\pm0.0005}$ | 0.051$_{\pm0.0002}$ | 0.400$_{\pm0.0012}$ | 0.049$_{\pm0.0002}$ | 0.044$_{\pm0.0002}$ | 0.043$_{\pm0.0002}$ | 0.039$_{\pm0.0003}$ |

model training or architecture design alone, we also emphasize the results for PGDM$_{\text{GDE}}$ in Tables 1 and 2. Even when unguided PGDM$_{\text{GDE}}$ performs worse than baselines, pattern guidance almost always reduces the error of PGDM$_{\text{GDE}}$ is to a lower or competitive level compared to baselines. These results demonstrate that pattern guidance is essential for higher quality predictions.

## 5.2 ABLATIONS

For all ablations, we report full results with additional discussion in Appendix H.

Table 3: Mean absolute error (MAE) of $\text{PGDM}_{\text{MAE}}$ and $\text{PGDM}_{\text{GDE}}$ on the VF prediction application (UWHVF dataset) with varying levels of guidance. Percent improvements over baselines are shown in the $\Delta$ MAE (%) columns. Mean and standard deviation are taken across five samples.

| Model | | MAE (dB) | $\Delta$ MAE (%) vs. GenViT | $\Delta$ MAE (%) vs. TimeGrad | $\Delta$ MAE (%) vs. CSDI | $\Delta$ MAE (%) vs. ARMD | $\Delta$ MAE (%) vs. $\overline{w} = 0$ |
|---|---|---|---|---|---|---|---|
| $\text{PGDM}_{\text{MAE}}$ | $\overline{w} = 0$ | $3.75_{\pm 0.0437}$ | $56.48_{\pm 0.50}$ | $10.69_{\pm 1.17}$ | $-17.32_{\pm 1.55}$ | $4.29_{\pm 1.21}$ | - |
| | $\overline{w} = 1$ | $2.96_{\pm 0.0117}$ | $65.58_{\pm 0.14}$ | $29.36_{\pm 0.67}$ | $7.20_{\pm 1.25}$ | $24.30_{\pm 0.31}$ | $20.90_{\pm 0.71}$ |
| | $\overline{w} = 2$ | $2.97_{\pm 0.0112}$ | $65.54_{\pm 0.13}$ | $29.29_{\pm 0.67}$ | $7.10_{\pm 1.27}$ | $24.22_{\pm 0.29}$ | $20.81_{\pm 0.73}$ |
| | $\overline{w} = 3$ | $2.97_{\pm 0.0108}$ | $65.51_{\pm 0.13}$ | $29.23_{\pm 0.67}$ | $7.03_{\pm 1.29}$ | $24.15_{\pm 0.26}$ | $20.75_{\pm 0.76}$ |
| | $\overline{w} = 4$ | $2.97_{\pm 0.0107}$ | $65.48_{\pm 0.13}$ | $29.17_{\pm 0.67}$ | $6.95_{\pm 1.31}$ | $24.09_{\pm 0.09}$ | $20.68_{\pm 0.78}$ |
| | $\overline{w} = 5$ | $2.97_{\pm 0.0104}$ | $65.45_{\pm 0.12}$ | $29.10_{\pm 0.66}$ | $6.86_{\pm 1.32}$ | $24.02_{\pm 0.02}$ | $20.60_{\pm 0.80}$ |
| $\text{PGDM}_{\text{GDE}}$ | $\overline{w} = 0$ | $5.20_{\pm 0.0407}$ | $39.60_{\pm 0.47}$ | $-23.95_{\pm 1.10}$ | $-62.83_{\pm 2.59}$ | $-32.84_{\pm 1.05}$ | - |
| | $\overline{w} = 1$ | $3.16_{\pm 0.0212}$ | $63.28_{\pm 0.24}$ | $24.65_{\pm 0.76}$ | $1.01_{\pm 1.44}$ | $19.25_{\pm 0.58}$ | $39.21_{\pm 0.23}$ |
| | $\overline{w} = 2$ | $3.13_{\pm 0.0201}$ | $63.67_{\pm 0.23}$ | $25.44_{\pm 0.76}$ | $2.06_{\pm 1.43}$ | $20.10_{\pm 0.55}$ | $39.85_{\pm 0.23}$ |
| | $\overline{w} = 3$ | $3.10_{\pm 0.0187}$ | $63.93_{\pm 0.21}$ | $25.99_{\pm 0.74}$ | $2.77_{\pm 1.41}$ | $20.68_{\pm 0.51}$ | $40.29_{\pm 0.24}$ |
| | $\overline{w} = 4$ | $3.09_{\pm 0.0170}$ | $64.09_{\pm 0.20}$ | $26.32_{\pm 0.72}$ | $3.20_{\pm 1.39}$ | $21.03_{\pm 0.47}$ | $40.55_{\pm 0.25}$ |
| | $\overline{w} = 5$ | $3.08_{\pm 0.0153}$ | $64.16_{\pm 0.18}$ | $26.47_{\pm 0.70}$ | $3.40_{\pm 1.39}$ | $21.19_{\pm 0.43}$ | $40.67_{\pm 0.27}$ |

**Impact of pattern mixing.** We study the impact of pattern mixing on the MAE and $\text{CRPS}_{\text{SUM}}$ of PGDM by varying the maximum mixing scale $\overline{w^*}$ from 0.0 to 1.0 while keeping the maximum guidance scale $\overline{w}$ fixed. On UWHVF, we find that a strong mixing signal substantially improves performance. In contrast, on AIST++, a weaker mixing scale is preferable, and an excessive mixing scale can degrade performance. This indicates that the pattern prediction model itself makes more accurate predictions on the small UWHVF dataset, even without the downstream denoiser. This suggests that explicit pattern modeling is particularly beneficial in low-data regimes.

**Impact of dynamic scaling.** We additionally perform ablations to demonstrate the benefits of dynamic scaling. We compare the MAE and $\text{CRPS}_{\text{SUM}}$ of PGDM with a dynamic scale to that with a constant scale, holding $\overline{w}$ and $\overline{w^*}$ fixed. We find that dynamic scaling consistently performs comparably or better than constant scaling. Dynamic scaling likely yields the greatest improvements over constant scaling when novel patterns are seen inference time.

**Impact of pattern prediction.** Finally, we compare PGDM with and without its pattern prediction model $f_A$. PGDM *without* pattern prediction inherently follows the assumption that patterns remain constant over time. This ablation therefore serves as a proxy for Diff-MGR (Zhao et al., 2024), which follows this same assumption but cannot be evaluated directly due to unavailable code and implementation details. PGDM with pattern prediction consistently outperforms PGDM without pattern prediction, most significantly when patterns change rapidly over time. This result highlights the importance of the pattern prediction model in accounting for the highly dynamic patterns that commonly appear in realistic temporal data.

## 6 CONCLUSIONS

In this paper, we proposed Pattern-Guided Diffusion Models (PGDM), which leverage inherent archetypal patterns to forecast future steps from multivariate time series data. PGDM is guided by a pattern guidance function that predicts future patterns within the data. To estimate the trustworthiness of this guidance function, we introduced a novel uncertainty quantification metric that approximately lower bounds the guidance function error. Finally, we proposed to dynamically tune the level to which PGDM follows the pattern guidance based on this uncertainty metric. We found that PGDM outperforms baseline models, and pattern guidance improves the prediction quality of PGDM. Two limitations of PGDM present interesting avenues for future work. First, PGDM has less benefit for out-of-distribution data exhibiting unseen patterns. Second, the use of AAUQ as an approximate lower-bound for guidance function error assumes that the temporal data is relatively continuous and does not rapidly change between the observed history and target prediction window. In some cases, this assumption is violated (e.g., rapidly changing signals sampled with a low frequency). Based on these limitations, PGDM may be further improved by updating the set of extracted patterns at inference time and accounting for signal dynamics when calculating the guidance scale.

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

## A  Proof of Theorem 1

For convenience, let $X = x_{1:T}$ and $Y = x_{T:T'}$. Then we have

$$
\begin{aligned}
L_{f_G}(X) &= \|Y - Af_A\left(c_A(X)\right)\| \\
&= \|Y - Af_A\left(c_A(X)\right) + Ac_A(Y) - Ac_A(Y)\| \\
&\geq \|Y - Ac_A(Y)\| - \|Af_A\left(c_A(X)\right) - Ac_A(Y)\| \\
&= L_{c_A}(Y) - L_{f_A}(X).
\end{aligned}
$$

The inequality above follows from the reverse triangle inequality.

## B  Proof of Theorem 2

First, observe that $u_A(x_t)$ is the distance between $x_t$ and the set $\mathrm{Conv}A$. This can be seen by noting that

$$
c_A(x_t) = \arg\min_c \|x_t - Ac\| = \arg\min_{\bar{x} \in \mathrm{Conv}A} \|x_t - \bar{x}\|, \tag{9}
$$

where $c \in \mathbb{R}^p$ has positive elements that sum to one. Let $\hat{x} = Ac_A(x_t)$. Recall from Definition 1 that $u_A(x_t) = \|x_t - \hat{x}\|$.

Similarly, $\text{dist}(x_t, \text{Conv}D)$ is defined as

$$\text{dist}(x_t, \text{Conv}D) = \min_{\bar{d} \in \text{Conv}D} \left\| x_t - \bar{d} \right\|.$$

Let $d$ be such that $\text{dist}(x_t, \text{Conv}D) = \|x_t - d\|$.

The remainder of the proof follows from a straightforward application of the reverse triangle inequality:

$$\|\hat{x} - d\| = \|\hat{x} - d + x_t - x_t\|$$
$$\geq \|\|x_t - d\| - \|x_t - \hat{x}\|\|.$$

Then, we have

$$-\|\hat{x} - d\| \leq \|x_t - d\| - \|x_t - \hat{x}\| \leq \|\hat{x} - d\|.$$

With some rearranging, we arrive at Equation equation 4 by letting $\delta = \|\hat{x} - d\|$.

Now note that if $p = n$, then selecting $A = D$ minimizes the archetypal analysis objective equation 1 (Cutler & Breiman, 1994, Proposition 1) with RSS of 0, and $A$ fully expresses $D$. Then $\text{Conv}A = \text{Conv}D$ and $\hat{x} = d$. Finally, $\delta = 0$. We briefly remark that $\delta$ therefore captures the expressiveness of the archetypes.

## C  PATTERN MIXING

In Lines 9 and 10 of Algorithm 2, we include an additional pattern mixing step in our sequence prediction process. The final PGDM prediction is a linear combination of the raw pattern prediction from the guidance function and the the pattern-guided output of the diffusion model. We include this step to overcome some of the practical challenges of PGDM. In practice, we find that PGDM's capacity for pattern guidance is highly dependent on appropriate architecture design. We therefore include pattern mixing as an additional step to overcome this challenge.

While pattern mixing improves prediction quality, pattern guidance is still necessary. Figure 5 illustrates the impact of pattern guidance and pattern mixing. Without guidance, the model may make highly varied predictions that are far from the groundtruth. With pattern guidance, PGDM narrows the distribution of predictions and shifts it towards the ground truth. Pattern mixing further shifts the distribution, without affecting sample diversity. Our results demonstrate exactly this. The unguided PGDM prediction has higher error and variance. With guidance and mixing, the error and standard deviation are significantly reduced, demonstrating that both pattern guidance and pattern mixing aid in improving predictions.

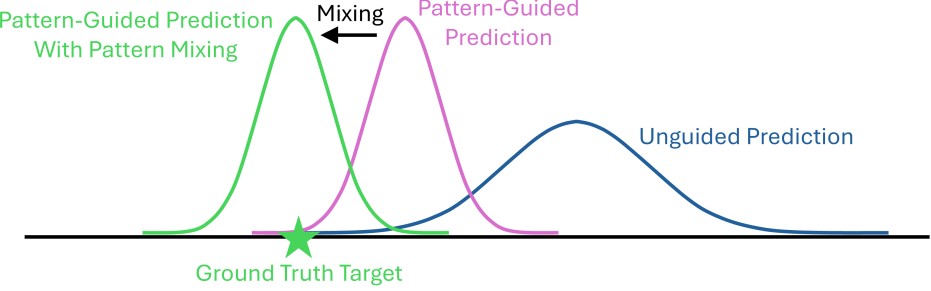

Figure 5: Impacts of pattern guidance and pattern mixing.

## D  IMPLEMENTATION DETAILS

For both case studies, we split our data into 70% training, 15% validation, and 15% test sets. For the motion capture data, we remove rotations around the vertical axis and supply the isolated rotation angles as additional inputs to PGDM and the baseline models. This normalizes the direction in which the motion capture skeletons are facing, allowing for more straightforward pattern extraction.

Table 4: Hyperparameter choices for pattern prediction model.

| | Pattern Prediction Model | | PGDM$_{\text{MAE}}$ | | | PGDM$_{\text{GDE}}$ | | |
|---|---|---|---|---|---|---|---|---|
| | Batch Size | LR | Batch Size | LR | Epochs | Batch Size | LR | Epochs |
| UWHVF | 32 | $1 \times 10^{-4}$ | 32 | $5 \times 10^{-5}$ | 200 | 64 | $1 \times 10^{-5}$ | 100 |
| Break | 32 | $5 \times 10^{-4}$ | 32 | $1 \times 10^{-3}$ | 300 | 64 | $5 \times 10^{-4}$ | 200 |
| House | 64 | $5 \times 10^{-4}$ | 32 | $1 \times 10^{-3}$ | 300 | 32 | $5 \times 10^{-4}$ | 200 |
| Ballet jazz | 64 | $5 \times 10^{-4}$ | 32 | $1 \times 10^{-3}$ | 300 | 64 | $5 \times 10^{-4}$ | 300 |
| Street jazz | 64 | $5 \times 10^{-4}$ | 32 | $1 \times 10^{-3}$ | 300 | 64 | $1 \times 10^{-3}$ | 200 |
| Krump | 64 | $5 \times 10^{-4}$ | 32 | $1 \times 10^{-3}$ | 300 | 32 | $5 \times 10^{-4}$ | 200 |
| LA Hip Hop | 64 | $5 \times 10^{-4}$ | 32 | $1 \times 10^{-3}$ | 300 | 32 | $5 \times 10^{-4}$ | 200 |
| Lock | 64 | $5 \times 10^{-4}$ | 32 | $1 \times 10^{-3}$ | 300 | 32 | $1 \times 10^{-3}$ | 200 |
| Middle Hip Hop | 64 | $5 \times 10^{-4}$ | 32 | $1 \times 10^{-3}$ | 300 | 64 | $5 \times 10^{-4}$ | 200 |
| Pop | 32 | $5 \times 10^{-4}$ | 32 | $1 \times 10^{-3}$ | 300 | 32 | $5 \times 10^{-4}$ | 200 |
| Wack | 32 | $5 \times 10^{-4}$ | 32 | $1 \times 10^{-3}$ | 300 | 32 | $5 \times 10^{-4}$ | 300 |

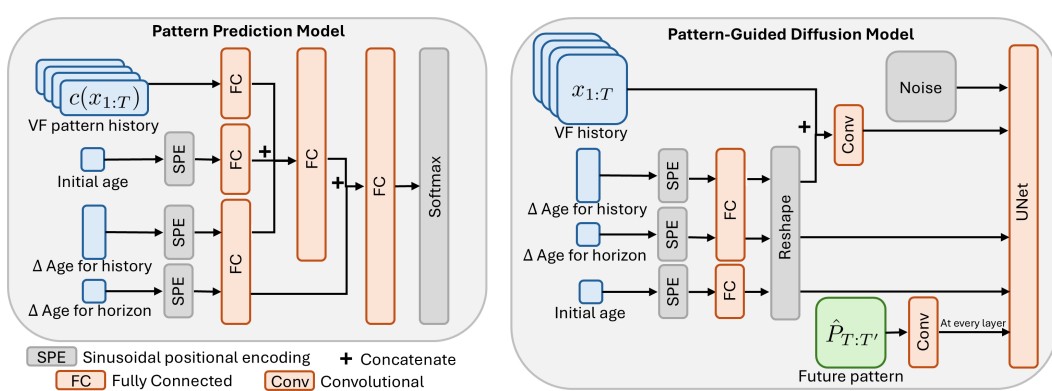

Figure 6: Pattern prediction model and pattern-guided diffusion model for visual field prediction.

Table 5: Selected values of $\overline{w}$ and $\overline{w^*}$ for Table 1.

| | PGDM$_{\text{MAE}}$ | | PGDM$_{\text{GDE}}$ | |
|---|---|---|---|---|
| | $\overline{w}$ | $w^*$ | $\overline{w}$ | $w^*$ |
| UWHVF | 1 | 1.0 | 5 | 1.0 |
| Break | 1 | 0.2 | 2 | 0.2 |
| House | 1 | 0.2 | 1 | 0.2 |
| Ballet jazz | 1 | 0.2 | 1 | 0.2 |
| Street jazz | 2 | 0.0 | 2 | 0.0 |
| Krump | 2 | 0.0 | 1 | 0.2 |
| LA Hip Hop | 1 | 0.2 | 1 | 0.2 |
| Lock | 1 | 0.2 | 1 | 0.2 |
| Middle Hip Hop | 1 | 0.2 | 1 | 0.2 |
| Pop | 1 | 0.2 | 1 | 0.2 |
| Wack | 1 | 0.2 | 1 | 0.2 |

To train our guidance function, we first extract archetypal patterns from the most recent VF $x_T$ of each sequence in the training set. By extracting archetypes from only a single point in each sequence, we avoid leakage between the training, validation, and test sets. We select the number of archetypes $p$ by a hyperparameter search through $p = 2, \ldots, 25$ with selection criterion following Elze et al. (2015). We then train our pattern prediction model (see Figure 6 for architecture) to predict the pattern representation of each sequence. We train the model with the Adam optimizer on a KL-divergence loss function with the hyperparameters shown in Table 4 and patience 20. These hyperparameters were selected over a search of batch size 32 to 64 and learning rate $10^{-4}$ to $5 \times 10^{-4}$, with mean absolute error (MAE) as selection criterion.

Table 6: Parameter counts.

| | PGDM Pattern Predictor | PGDM Denoiser | TimeGrad | CSDI | ARMD | GenViT |
|---|---|---|---|---|---|---|
| **UWHVF** | 61,581 | 394,567 | 60,103 | 610,945 | 375 | 4,393,874 |
| **Break** | 270,687 | | | | | |
| **House** | 282,222 | | | | | |
| **Ballet jazz** | 255,307 | | | | | |
| **Street jazz** | 251,462 | | | | | |
| **Krump** | 282,222 | | | | | |
| **LA Hip Hop** | 282,222 | 418,483 | 61,251 | 610,961 | 613 | - |
| **Lock** | 282,222 | | | | | |
| **Middle Hip Hop** | 282,222 | | | | | |
| **Pop** | 270,687 | | | | | |
| **Wack** | 243,772 | | | | | |

Table 7: Inference latency (ms). Latency is computed by measuring the average wall-clock time of a forward pass with batch size 32 (over 10 repeated runs on a fixed batch after warmup) and dividing the resulting batch latency by 32. Latency on the AIST++ dataset is measured using the break genre.

| | PGDM | TimeGrad | CSDI | ARMD | GenViT |
|---|---|---|---|---|---|
| **UWHVF** | 742.60 | <0.01 | 10.63 | 0.04 | 314.28 |
| **AIST++** | 732.30 | <0.01 | 27.20 | 0.03 | - |

We train our diffusion model (see Figure 6 for architecture) with the Adam optimizer on a mean square error loss function with the hyperparameters shown in Table 4. For the VF prediction application, these hyperparameters were selected over a search of batch size 32 to 64, learning rate $10^{-5}$ to $5 \times 10^{-5}$, and 100 to 1000 epochs. For the motion prediction application, the model was trained with learning rate scheduling, and the hyperparameters were selected over a search of batch size 32 to 64, learning rate $5 \times 10^{-4}$ to $10^{-3}$ and 200 to 300 epochs. In both applications, for our selection criterion, we measure MAE with maximum guidance scale $\bar{w} = 1, \ldots, 10$ and no pattern mixing, and we choose only from models with the highest capacity for pattern guidance (i.e., error continues to reduce with increasing $\bar{w}$). Of these, we select the models with lowest achievable MAE over the tested range of $\bar{w}$. To better evaluate the full effect of pattern guidance on model performance, we also select models with the highest impact of pattern guidance over the tested range of $\bar{w}$ (e.g., the greatest achievable percent decrease in error from applying guidance). For all PGDM models, we train with conditioning dropout probability $p_{\text{drop}} = 0.2$. We evaluate with maximum tolerable uncertainty $\gamma = 0.1, 0.03, 0.06, 0.04, 0.04, 0.05, 0.05, 0.06, 0.06, 0.03$, and $0.05$ for UWHVF, break, house, ballet jazz, street jazz, krump, LA hip hop, lock, middle hip hop, pop, and wack, respectively. These were chosen based on the range of uncertainties on the validation data. In Table 5, we report the choices of $\overline{w}$ and $\overline{w^*}$ that we use to generate Table 1.

For our baselines TimeGrad, CSDI, and ARMD, we select hyperparameters following the published implementation details. For the GenViT model, we select hyperparameters from a hyperparameter search, as those published in Tian et al. (2023) were for a simpler task with $H = 1$ and $T = 1$. We train with batch size 16, learning rate $10^{-5}$, and 50 epochs. These hyperparameters were selected from a search over batch size 8 to 16, learning rate $10^{-5}$ to $5 \times 10^{-5}$, and 10 to 50 epochs, with MAE as selection criterion. These ranges were chosen to remain consistent with the settings of Tian et al. (2023).

We use default architectures for all baseline models except ARMD. The original ARMD implementation uses a single layer linear network, with layer size equal to the prediction horizon, which can be extremely small in our application settings and therefore insufficiently expressive. For a fairer comparison, we instead apply a lightweight two layer MLP with hidden dimension 32 and layer normalization. In addition, ARMD's evolution (i.e., forward) and devolution (i.e., reverse) processes assume equal history and prediction lengths. To accommodate unequal horizons, we adapt both processes so that a window of length $H$ slides from time step 1 through $T$. Finally, following ARMD's DDIM-style sampling procedure, we adopt $\eta = 0.1$ to introduce a small amount of stochasticity. These minimal adjustments preserve the core design of ARMD while enabling its application to our forecasting setting.

All experiments were performed on a machine with 42 GB of GPU memory. Each model requires less than 1 GB. We report parameter counts for each model in Table 6 and inference latency in Table 7. The sampling efficiency of our model can be substantially improved using accelerated samplers such as DDIM. Exploring these optimizations is beyond the scope of this work.

## E    PATTERNS EXTRACTED FROM AIST++

Figures 7 to 15 show the patterns extracted from each genre of the AIST++ dataset.

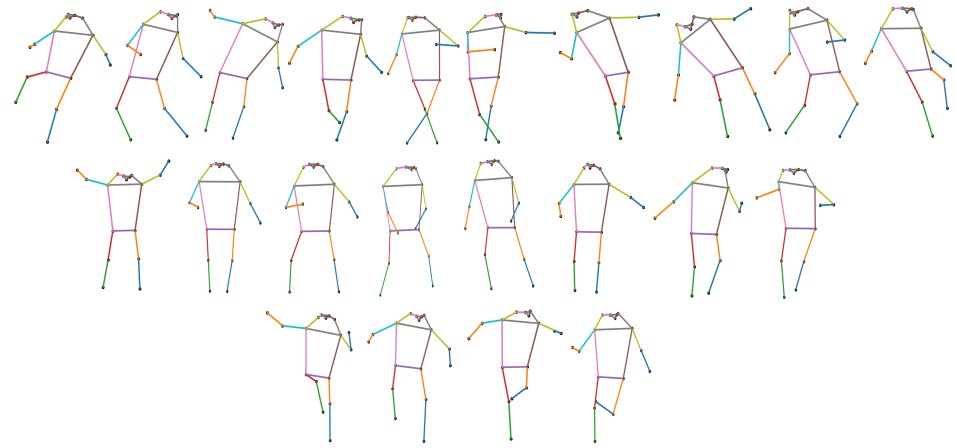

Figure 7: Twenty two archetypes extracted from AIST++ house dancing frames.

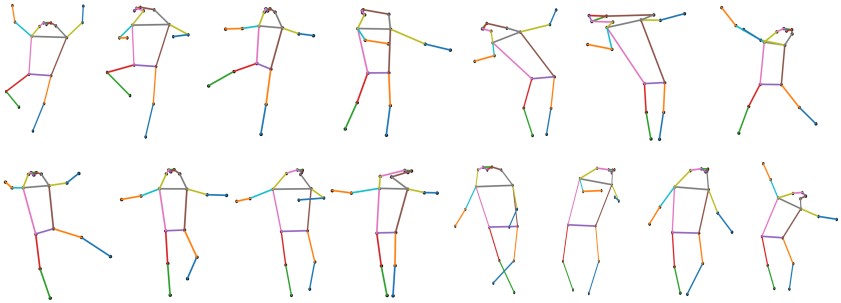

Figure 8: Fifteen archetypes extracted from AIST++ ballet jazz dancing frames.

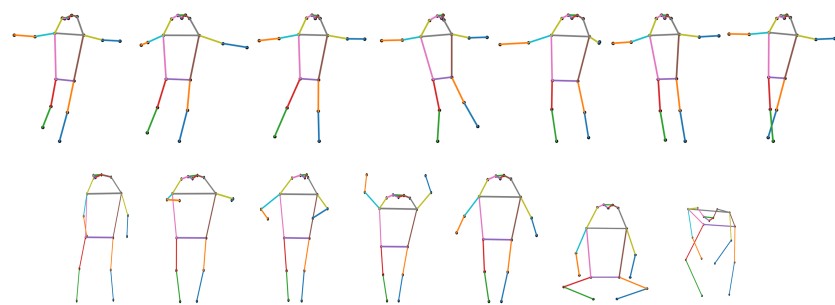

Figure 9: Fourteen archetypes extracted from AIST++ street jazz dancing frames.

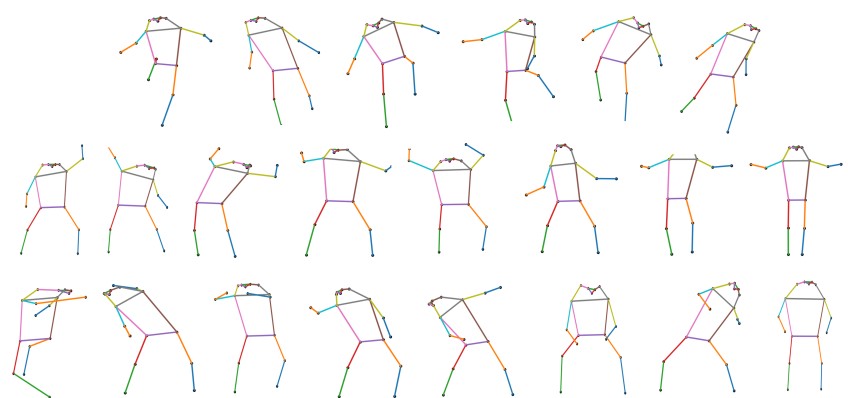

Figure 10: Twenty two archetypes extracted from AIST++ krump dancing frames.

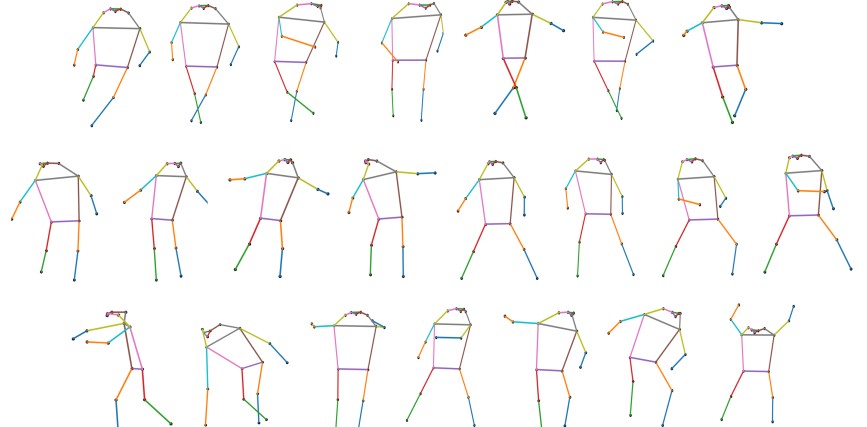

Figure 11: Twenty two archetypes extracted from AIST++ LA hip hop dancing frames.

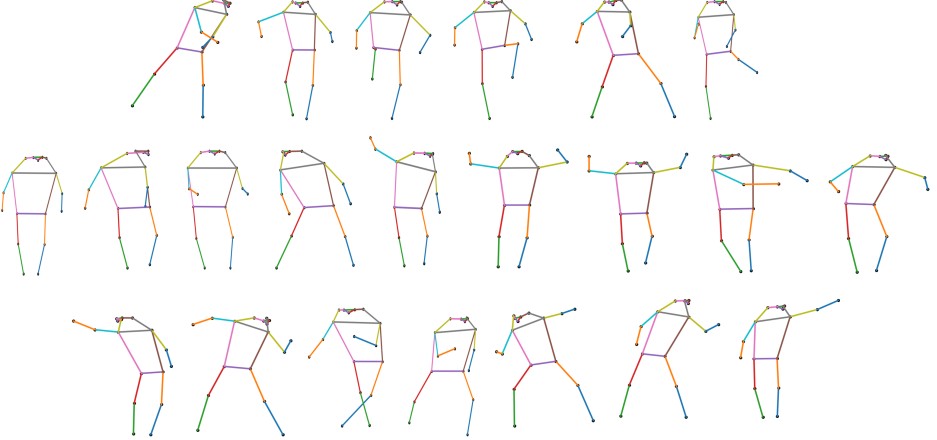

Figure 12: Twenty two archetypes extracted from AIST++ lock dancing frames.

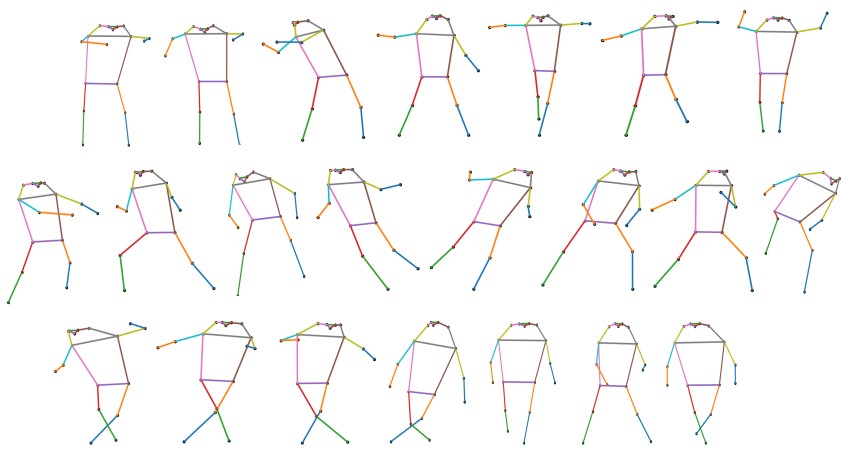

Figure 13: Twenty two archetypes extracted from AIST++ middle hip hop dancing frames.

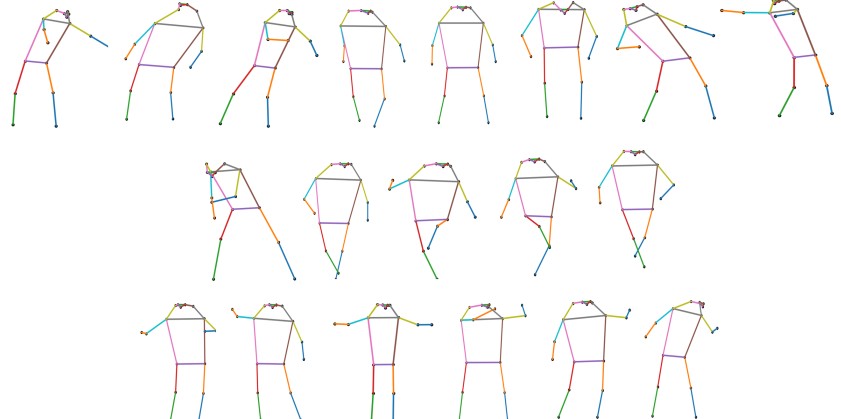

Figure 14: Nineteen archetypes extracted from AIST++ pop dancing frames.

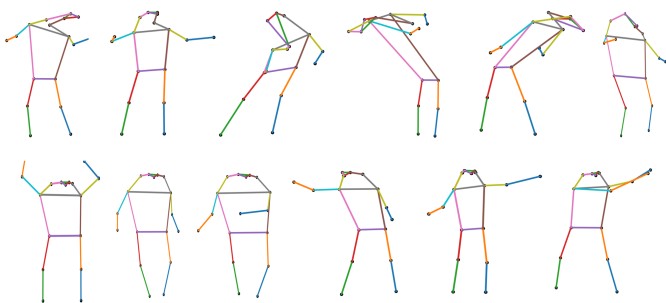

Figure 15: Twelve archetypes extracted from AIST++ wack dancing frames.

## F  EVALUATION OF PGDM COMPONENTS

Table 8 shows the reconstruction error of the extracted patterns, the error of the pattern prediction model, and the error of the guidance function.

Table 8: Mean absolute error (MAE) of the guidance function and its components. Note that the pattern prediction is performed in the pattern representation space, with range $(0, 1)$.

|  | Archetypal Analysis | Pattern Prediction | Guidance Function |
|---|---|---|---|
| **UWHVF** | 2.3146 | 0.0466 | 3.1560 |
| **Break** | 0.7929 | 0.0111 | 0.8440 |
| **House** | 1.6591 | 0.0111 | 1.7636 |
| **Ballet Jazz** | 0.6269 | 0.0156 | 0.6698 |
| **Street Jazz** | 1.3009 | 0.0108 | 1.3438 |
| **Krump** | 1.6768 | 0.0085 | 1.7677 |
| **LA Hip Hop** | 1.6707 | 0.0107 | 1.7622 |
| **Lock** | 1.3892 | 0.0102 | 1.4743 |
| **Middle Hip Hop** | 1.8489 | 0.0113 | 1.9682 |
| **Pop** | 0.9886 | 0.0106 | 1.0572 |
| **Wack** | 0.5353 | 0.0166 | 0.5973 |

## G  IMPACT OF PATTERN GUIDANCE

In the main text, we observed that pattern guidance reduces the error of PGDM's predictions. To further illustrate this point, qualitative examples for both applications are shown in Figure 16. For VF prediction, we show five example $H = 1$ step-ahead predictions from PGDM$_{\text{GDE}}$. When pattern guidance is not used ($\overline{w} = 0$), PGDM makes a noisy prediction based only on the past visual field data. When pattern guidance is added ($\overline{w} = 5$), PGDM incorporates the pattern prediction in its forecast. The outcome resembles a mixture of the pattern prediction and the unguided prediction (see Ex. 2 of 16a). For motion prediction, we show one example $H = 5$ step ahead prediction for PGDM$_{\text{GDE}}$. In this example, we highlight the bent right leg of the skeleton. Without pattern guidance ($\overline{w} = 0$), the model predicts nearly no motion in the leg across the horizon. In contrast, the guidance function predicts a set of patterns that change over time, matching the moving right leg of the ground truth frames. When guidance is used ($\overline{w} = 2$), PGDM incorporates this motion into its prediction and forecasts more accurate future frames.

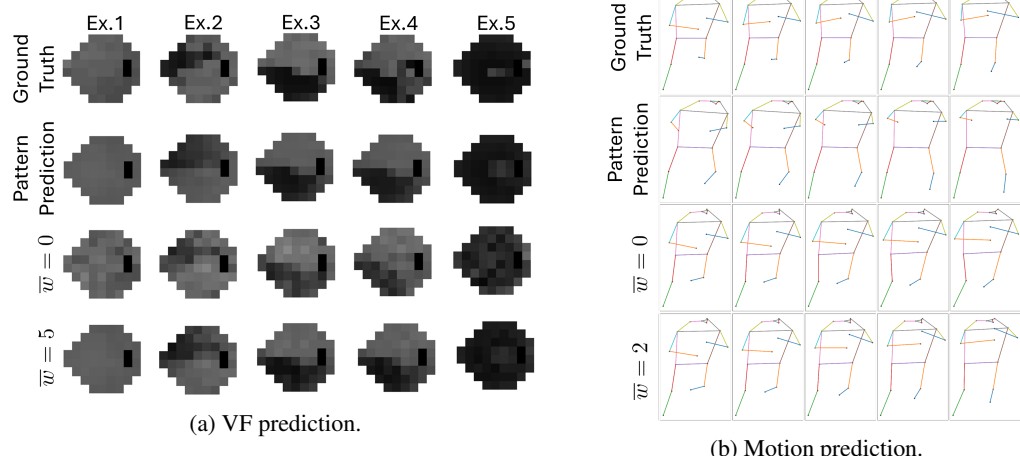

(a) VF prediction.

(b) Motion prediction.

Figure 16: Qualitative examples of pattern guidance for PGDM$_{\text{GDE}}$ on a) the visual field prediction application and b) the human motion prediction application.

In Tables 9 to 18, we show the quantitative effect of pattern guidance levels $\overline{w} = 1, 2, 3, 4, 5$ on PGDM's MAE for the human motion prediction applications, holding the mixing scale $\overline{w}^*$ constant at its optimal choice. Each table corresponds to one of the dance genres in the AIST++ dataset. Table 19 additionally shows the effect of pattern guidance levels on the CRPS of PGDM for both the visual field and human motion prediction applications.[1] In general $\text{PGDM}_{\text{MAE}}$ and $\text{PGDM}_{\text{GDE}}$ achieve their best performances with relatively light guidance. Beyond this point, the pattern guidance has diminishing returns, even increasing the prediction error when the guidance scale is too high. In practice, the appropriate $\overline{w}$ may be selected in a manner similar to a hyperparameter search. We also observe that, in most cases, the standard deviation of the MAE decreases as $\overline{w}$ increases up to the optimal $\overline{w}$. This indicates that pattern guidance improves both the quality and the consistency of predictions.

Table 9: Mean absolute error (MAE) of $\text{PGDM}_{\text{MAE}}$, $\text{PGDM}_{\text{GDE}}$, and baselines on the break dancing genre of the AIST++ dataset. Percent improvements over baselines are shown in the $\Delta$ MAE (%) columns. Mean and standard deviation are taken across five samples.

| Model | | MAE (dB) | $\Delta$ MAE (%) vs. TimeGrad | $\Delta$ MAE (%) vs. CSDI | $\Delta$ MAE (%) vs. ARMD | $\Delta$ MAE (%) vs. $\overline{w} = 0$ |
|---|---|---|---|---|---|---|
| TimeGrad | | $2.10_{\pm 0.0064}$ | - | - | - | - |
| CSDI | | $0.47_{\pm 0.0010}$ | - | - | - | - |
| ARMD | | $4.26_{\pm 0.0026}$ | - | - | - | - |
| $\text{PGDM}_{\text{MAE}}$ | $\overline{w} = 0$ | $0.41_{\pm 0.0013}$ | $80.33_{\pm 0.10}$ | $12.54_{\pm 0.15}$ | $90.31_{\pm 0.03}$ | - |
| | $\overline{w} = 1$ | $\textcolor{red}{0.38_{\pm 0.0009}}$ | $\textcolor{red}{81.91_{\pm 0.07}}$ | $\textcolor{red}{19.54_{\pm 0.14}}$ | $\textcolor{red}{91.08_{\pm 0.02}}$ | $8.01_{\pm 0.14}$ |
| | $\overline{w} = 2$ | $\textcolor{red}{0.38_{\pm 0.0009}}$ | $81.80_{\pm 0.07}$ | $19.04_{\pm 0.16}$ | $91.03_{\pm 0.02}$ | $7.43_{\pm 0.16}$ |
| | $\overline{w} = 3$ | $0.40_{\pm 0.0007}$ | $81.16_{\pm 0.06}$ | $16.22_{\pm 0.17}$ | $90.72_{\pm 0.02}$ | $4.21_{\pm 0.28}$ |
| | $\overline{w} = 4$ | $0.42_{\pm 0.0009}$ | $80.19_{\pm 0.09}$ | $11.91_{\pm 0.19}$ | $90.24_{\pm 0.02}$ | $-0.72_{\pm 0.33}$ |
| | $\overline{w} = 5$ | $0.44_{\pm 0.0010}$ | $79.02_{\pm 0.09}$ | $6.71_{\pm 0.14}$ | $89.66_{\pm 0.02}$ | $-6.66_{\pm 0.32}$ |
| $\text{PGDM}_{\text{GDE}}$ | $\overline{w} = 0$ | $0.52_{\pm 0.0032}$ | $75.25_{\pm 0.18}$ | $-10.06_{\pm 0.76}$ | $87.80_{\pm 0.07}$ | - |
| | $\overline{w} = 1$ | $0.45_{\pm 0.0023}$ | $78.32_{\pm 0.13}$ | $3.60_{\pm 0.64}$ | $89.32_{\pm 0.05}$ | $12.41_{\pm 0.29}$ |
| | $\overline{w} = 2$ | $0.45_{\pm 0.0015}$ | $78.75_{\pm 0.11}$ | $5.48_{\pm 0.45}$ | $89.53_{\pm 0.03}$ | $\textcolor{red}{14.12_{\pm 0.35}}$ |
| | $\overline{w} = 3$ | $0.46_{\pm 0.0008}$ | $78.24_{\pm 0.09}$ | $3.24_{\pm 0.28}$ | $89.28_{\pm 0.02}$ | $12.08_{\pm 0.44}$ |
| | $\overline{w} = 4$ | $0.48_{\pm 0.0010}$ | $77.34_{\pm 0.11}$ | $-0.79_{\pm 0.21}$ | $88.83_{\pm 0.02}$ | $8.42_{\pm 0.63}$ |
| | $\overline{w} = 5$ | $0.50_{\pm 0.0012}$ | $76.21_{\pm 0.12}$ | $-5.81_{\pm 0.21}$ | $88.27_{\pm 0.03}$ | $3.86_{\pm 0.74}$ |

---

[1]To avoid redundancy, we report CRPS alone and do not report percent improvements over baselines. Trends apparent in the reported MAE results are consistent with those reported in the CRPS results. Compared to unguided predictions, $\text{PGDM}_{\text{GDE}}$ achieves the greatest reduction in CRPS using guidance (56.26% on UWHVF and 14.10% on the LA hip hop genre of AIST++). Compared to baselines, $\text{PGDM}_{\text{MAE}}$ can surpass baselines by up to 84.55% on UWHVF and 92.55% on the break dancing genre of AIST++.

Table 10: Mean absolute error (MAE) of PGDM$_{\text{MAE}}$, PGDM$_{\text{GDE}}$, and baselines on the house dancing genre of the AIST++ dataset. Percent improvements over baselines are shown in the $\Delta$ MAE (%) columns. Mean and standard deviation are taken across five samples.

| Model | | MAE (dB) | $\Delta$ MAE (%) vs. TimeGrad | $\Delta$ MAE (%) vs. CSDI | $\Delta$ MAE (%) vs. ARMD | $\Delta$ MAE (%) vs. $\overline{w} = 0$ |
|---|---|---|---|---|---|---|
| TimeGrad | | $3.71_{\pm 0.0159}$ | - | - | - | - |
| CSDI | | $1.02_{\pm 0.0045}$ | - | - | - | - |
| ARMD | | $9.11_{\pm 0.0035}$ | - | - | - | - |
| PGDM$_{\text{MAE}}$ | $\overline{w} = 0$ | $0.79_{\pm 0.0017}$ | $78.63_{\pm 0.12}$ | $22.11_{\pm 0.35}$ | $91.30_{\pm 0.02}$ | - |
| | $\overline{w} = 1$ | $0.74_{\pm 0.0014}$ | $79.95_{\pm 0.11}$ | $26.93_{\pm 0.34}$ | $91.84_{\pm 0.02}$ | $6.19_{\pm 0.10}$ |
| | $\overline{w} = 2$ | $0.75_{\pm 0.0011}$ | $79.75_{\pm 0.11}$ | $26.20_{\pm 0.31}$ | $91.76_{\pm 0.01}$ | $6.96_{\pm 0.09}$ |
| | $\overline{w} = 3$ | $0.78_{\pm 0.0012}$ | $79.03_{\pm 0.12}$ | $23.57_{\pm 0.29}$ | $91.46_{\pm 0.01}$ | $1.87_{\pm 0.17}$ |
| | $\overline{w} = 4$ | $0.82_{\pm 0.0014}$ | $78.01_{\pm 0.12}$ | $19.85_{\pm 0.34}$ | $91.05_{\pm 0.02}$ | $-2.90_{\pm 0.20}$ |
| | $\overline{w} = 5$ | $0.86_{\pm 0.0019}$ | $76.80_{\pm 0.14}$ | $15.43_{\pm 0.41}$ | $90.56_{\pm 0.02}$ | $-8.57_{\pm 0.28}$ |
| PGDM$_{\text{GDE}}$ | $\overline{w} = 0$ | $0.90_{\pm 0.0013}$ | $75.64_{\pm 0.11}$ | $11.19_{\pm 0.39}$ | $90.08_{\pm 0.08}$ | - |
| | $\overline{w} = 1$ | $0.82_{\pm 0.0014}$ | $77.88_{\pm 0.12}$ | $19.38_{\pm 0.35}$ | $91.00_{\pm 0.02}$ | $9.22_{\pm 0.12}$ |
| | $\overline{w} = 2$ | $0.82_{\pm 0.0017}$ | $77.81_{\pm 0.12}$ | $19.13_{\pm 0.40}$ | $90.97_{\pm 0.02}$ | $8.94_{\pm 0.25}$ |
| | $\overline{w} = 3$ | $0.85_{\pm 0.0017}$ | $77.09_{\pm 0.13}$ | $16.51_{\pm 0.42}$ | $90.67_{\pm 0.02}$ | $5.98_{\pm 0.23}$ |
| | $\overline{w} = 4$ | $0.89_{\pm 0.0019}$ | $76.09_{\pm 0.15}$ | $12.84_{\pm 0.29}$ | $90.27_{\pm 0.02}$ | $1.85_{\pm 0.18}$ |
| | $\overline{w} = 5$ | $0.93_{\pm 0.0015}$ | $74.96_{\pm 0.14}$ | $8.73_{\pm 0.28}$ | $89.81_{\pm 0.02}$ | $-2.78_{\pm 0.15}$ |

Table 11: Mean absolute error (MAE) of PGDM$_{\text{MAE}}$, PGDM$_{\text{GDE}}$, and baselines on the ballet jazz dancing genre of the AIST++ dataset. Percent improvements over baselines are shown in the $\Delta$ MAE (%) columns. Mean and standard deviation are taken across five samples.

| Model | | MAE (dB) | $\Delta$ MAE (%) vs. TimeGrad | $\Delta$ MAE (%) vs. CSDI | $\Delta$ MAE (%) vs. ARMD | $\Delta$ MAE (%) vs. $\overline{w} = 0$ |
|---|---|---|---|---|---|---|
| TimeGrad | | $1.32_{\pm 0.0098}$ | - | - | - | - |
| CSDI | | $0.55_{\pm 0.0054}$ | - | - | - | - |
| ARMD | | $3.38_{\pm 0.0011}$ | - | - | - | - |
| PGDM$_{\text{MAE}}$ | $\overline{w} = 0$ | $0.42_{\pm 0.0008}$ | $67.94_{\pm 0.19}$ | $22.83_{\pm 0.84}$ | $87.53_{\pm 0.02}$ | - |
| | $\overline{w} = 1$ | $0.38_{\pm 0.0002}$ | $71.13_{\pm 0.21}$ | $30.50_{\pm 0.67}$ | $88.77_{\pm 0.01}$ | $9.94_{\pm 0.16}$ |
| | $\overline{w} = 2$ | $0.39_{\pm 0.0005}$ | $70.53_{\pm 0.24}$ | $29.07_{\pm 0.64}$ | $88.54_{\pm 0.02}$ | $8.08_{\pm 0.24}$ |
| | $\overline{w} = 3$ | $0.41_{\pm 0.0006}$ | $68.59_{\pm 0.27}$ | $24.39_{\pm 0.69}$ | $87.79_{\pm 0.02}$ | $2.02_{\pm 0.30}$ |
| | $\overline{w} = 4$ | $0.45_{\pm 0.0011}$ | $65.89_{\pm 0.29}$ | $17.90_{\pm 0.82}$ | $86.74_{\pm 0.03}$ | $-6.39_{\pm 0.42}$ |
| | $\overline{w} = 5$ | $0.49_{\pm 0.0012}$ | $62.85_{\pm 0.30}$ | $10.58_{\pm 0.93}$ | $85.55_{\pm 0.03}$ | $-15.87_{\pm 0.45}$ |
| PGDM$_{\text{GDE}}$ | $\overline{w} = 0$ | $0.49_{\pm 0.0004}$ | $62.75_{\pm 0.30}$ | $10.34_{\pm 0.83}$ | $85.52_{\pm 0.02}$ | - |
| | $\overline{w} = 1$ | $0.42_{\pm 0.0009}$ | $67.74_{\pm 0.27}$ | $22.34_{\pm 0.81}$ | $87.45_{\pm 0.03}$ | $13.39_{\pm 0.17}$ |
| | $\overline{w} = 2$ | $0.43_{\pm 0.0010}$ | $67.15_{\pm 0.29}$ | $20.92_{\pm 0.81}$ | $87.23_{\pm 0.03}$ | $11.81_{\pm 0.20}$ |
| | $\overline{w} = 3$ | $0.46_{\pm 0.0010}$ | $64.86_{\pm 0.32}$ | $15.40_{\pm 0.89}$ | $86.33_{\pm 0.03}$ | $5.65_{\pm 0.24}$ |
| | $\overline{w} = 4$ | $0.50_{\pm 0.0008}$ | $61.66_{\pm 0.33}$ | $7.70_{\pm 0.93}$ | $85.09_{\pm 0.03}$ | $-2.94_{\pm 0.21}$ |
| | $\overline{w} = 5$ | $0.55_{\pm 0.0011}$ | $57.90_{\pm 0.38}$ | $-1.34_{\pm 1.04}$ | $83.63_{\pm 0.04}$ | $-13.02_{\pm 0.26}$ |

Table 12: Mean absolute error (MAE) of PGDM$_{\text{MAE}}$, PGDM$_{\text{GDE}}$, and baselines on the street jazz dancing genre of the AIST++ dataset. Percent improvements over baselines are shown in the $\Delta$ MAE (%) columns. Mean and standard deviation are taken across five samples.

| Model | | MAE (dB) | $\Delta$ MAE (%) vs. TimeGrad | $\Delta$ MAE (%) vs. CSDI | $\Delta$ MAE (%) vs. ARMD | $\Delta$ MAE (%) vs. $\overline{w} = 0$ |
|---|---|---|---|---|---|---|
| TimeGrad | | $1.65_{\pm 0.0102}$ | - | - | - | - |
| CSDI | | $0.56_{\pm 0.0054}$ | - | - | - | - |
| ARMD | | $7.57_{\pm 0.0049}$ | - | - | - | - |
| PGDM$_{\text{MAE}}$ | $\overline{w} = 0$ | $0.52_{\pm 0.0017}$ | $68.58_{\pm 0.25}$ | $6.58_{\pm 0.79}$ | $93.14_{\pm 0.02}$ | - |
| | $\overline{w} = 1$ | $0.48_{\pm 0.0010}$ | $70.66_{\pm 0.23}$ | $12.76_{\pm 0.76}$ | $93.59_{\pm 0.01}$ | $6.62_{\pm 0.19}$ |
| | $\overline{w} = 2$ | $0.48_{\pm 0.0008}$ | $70.87_{\pm 0.22}$ | $13.39_{\pm 0.80}$ | $93.64_{\pm 0.01}$ | $7.29_{\pm 0.18}$ |
| | $\overline{w} = 3$ | $0.49_{\pm 0.0009}$ | $70.23_{\pm 0.22}$ | $11.49_{\pm 0.85}$ | $93.50_{\pm 0.01}$ | $5.26_{\pm 0.20}$ |
| | $\overline{w} = 4$ | $0.51_{\pm 0.0009}$ | $69.07_{\pm 0.23}$ | $8.03_{\pm 0.91}$ | $93.25_{\pm 0.01}$ | $1.56_{\pm 0.22}$ |
| | $\overline{w} = 5$ | $0.54_{\pm 0.0009}$ | $67.53_{\pm 0.25}$ | $3.45_{\pm 0.95}$ | $92.91_{\pm 0.01}$ | $-3.34_{\pm 0.25}$ |
| PGDM$_{\text{GDE}}$ | $\overline{w} = 0$ | $0.60_{\pm 0.0016}$ | $63.85_{\pm 0.26}$ | $-7.49_{\pm 1.02}$ | $92.11_{\pm 0.02}$ | - |
| | $\overline{w} = 1$ | $0.55_{\pm 0.0007}$ | $66.66_{\pm 0.23}$ | $0.88_{\pm 0.99}$ | $92.72_{\pm 0.01}$ | $7.79_{\pm 0.19}$ |
| | $\overline{w} = 2$ | $0.54_{\pm 0.0005}$ | $67.12_{\pm 0.20}$ | $2.25_{\pm 0.97}$ | $92.82_{\pm 0.01}$ | $9.06_{\pm 0.19}$ |
| | $\overline{w} = 3$ | $0.55_{\pm 0.0005}$ | $66.41_{\pm 0.21}$ | $0.12_{\pm 0.96}$ | $92.67_{\pm 0.01}$ | $7.08_{\pm 0.17}$ |
| | $\overline{w} = 4$ | $0.58_{\pm 0.0004}$ | $65.08_{\pm 0.22}$ | $-3.82_{\pm 0.97}$ | $92.38_{\pm 0.01}$ | $3.41_{\pm 0.21}$ |
| | $\overline{w} = 5$ | $0.60_{\pm 0.0005}$ | $63.39_{\pm 0.22}$ | $-8.84_{\pm 1.00}$ | $92.01_{\pm 0.01}$ | $-1.25_{\pm 0.21}$ |

Table 13: Mean absolute error (MAE) of PGDM$_{\text{MAE}}$, PGDM$_{\text{GDE}}$, and baselines on the krump dancing genre of the AIST++ dataset. Percent improvements over baselines are shown in the $\Delta$ MAE (%) columns. Mean and standard deviation are taken across five samples.

| Model | | MAE (dB) | $\Delta$ MAE (%) vs. TimeGrad | $\Delta$ MAE (%) vs. CSDI | $\Delta$ MAE (%) vs. ARMD | $\Delta$ MAE (%) vs. $\overline{w} = 0$ |
|---|---|---|---|---|---|---|
| TimeGrad | | $2.37_{\pm 0.0067}$ | - | - | - | - |
| CSDI | | $0.77_{\pm 0.0017}$ | - | - | - | - |
| ARMD | | $8.90_{\pm 0.0046}$ | - | - | - | - |
| PGDM$_{\text{MAE}}$ | $\overline{w} = 0$ | $0.77_{\pm 0.0016}$ | $67.56_{\pm 0.13}$ | $-0.04_{\pm 0.31}$ | $91.36_{\pm 0.02}$ | - |
| | $\overline{w} = 1$ | $0.71_{\pm 0.0014}$ | $70.27_{\pm 0.13}$ | $8.30_{\pm 0.31}$ | $92.08_{\pm 0.01}$ | $8.33_{\pm 0.08}$ |
| | $\overline{w} = 2$ | $0.70_{\pm 0.0013}$ | $70.40_{\pm 0.13}$ | $8.70_{\pm 0.31}$ | $92.11_{\pm 0.01}$ | $8.73_{\pm 0.12}$ |
| | $\overline{w} = 3$ | $0.73_{\pm 0.0011}$ | $69.31_{\pm 0.13}$ | $5.33_{\pm 0.30}$ | $91.82_{\pm 0.01}$ | $5.37_{\pm 0.12}$ |
| | $\overline{w} = 4$ | $0.77_{\pm 0.0010}$ | $67.53_{\pm 0.13}$ | $-0.14_{\pm 0.28}$ | $91.35_{\pm 0.01}$ | $-0.11_{\pm 0.13}$ |
| | $\overline{w} = 5$ | $0.82_{\pm 0.0008}$ | $65.26_{\pm 0.12}$ | $-7.15_{\pm 0.25}$ | $90.74_{\pm 0.01}$ | $-7.11_{\pm 0.16}$ |
| PGDM$_{\text{GDE}}$ | $\overline{w} = 0$ | $0.88_{\pm 0.0016}$ | $62.85_{\pm 0.12}$ | $-14.59_{\pm 0.26}$ | $90.10_{\pm 0.02}$ | - |
| | $\overline{w} = 1$ | $0.79_{\pm 0.0005}$ | $66.85_{\pm 0.10}$ | $-2.23_{\pm 0.20}$ | $91.17_{\pm 0.01}$ | $10.79_{\pm 0.11}$ |
| | $\overline{w} = 2$ | $0.79_{\pm 0.0005}$ | $66.83_{\pm 0.10}$ | $-2.32_{\pm 0.24}$ | $91.16_{\pm 0.01}$ | $10.71_{\pm 0.11}$ |
| | $\overline{w} = 3$ | $0.81_{\pm 0.0006}$ | $65.81_{\pm 0.11}$ | $-5.46_{\pm 0.27}$ | $90.89_{\pm 0.01}$ | $7.97_{\pm 0.12}$ |
| | $\overline{w} = 4$ | $0.85_{\pm 0.0007}$ | $64.38_{\pm 0.11}$ | $-9.86_{\pm 0.28}$ | $90.51_{\pm 0.01}$ | $4.13_{\pm 0.12}$ |
| | $\overline{w} = 5$ | $0.88_{\pm 0.0007}$ | $62.76_{\pm 0.11}$ | $-14.84_{\pm 0.30}$ | $90.08_{\pm 0.01}$ | $-0.22_{\pm 0.13}$ |

Table 14: Mean absolute error (MAE) of PGDM$_{\text{MAE}}$, PGDM$_{\text{GDE}}$, and baselines on the LA hip hop dancing genre of the AIST++ dataset. Percent improvements over baselines are shown in the $\Delta$ MAE (%) columns. Mean and standard deviation are taken across five samples.

| Model | | MAE (dB) | $\Delta$ MAE (%) vs. TimeGrad | $\Delta$ MAE (%) vs. CSDI | $\Delta$ MAE (%) vs. ARMD | $\Delta$ MAE (%) vs. $\overline{w} = 0$ |
|---|---|---|---|---|---|---|
| TimeGrad | | $3.30_{\pm 0.0157}$ | - | - | - | - |
| CSDI | | $0.78_{\pm 0.0023}$ | - | - | - | - |
| ARMD | | $8.51_{\pm 0.0032}$ | - | - | - | - |
| PGDM$_{\text{MAE}}$ | $\overline{w} = 0$ | $0.80_{\pm 0.0010}$ | $75.83_{\pm 0.13}$ | $-1.95_{\pm 0.36}$ | $90.62_{\pm 0.01}$ | - |
| | $\overline{w} = 1$ | $0.74_{\pm 0.0006}$ | $77.58_{\pm 0.11}$ | $5.46_{\pm 0.33}$ | $91.30_{\pm 0.01}$ | $7.26_{\pm 0.06}$ |
| | $\overline{w} = 2$ | $0.75_{\pm 0.0006}$ | $77.41_{\pm 0.11}$ | $4.71_{\pm 0.33}$ | $91.23_{\pm 0.01}$ | $6.53_{\pm 0.08}$ |
| | $\overline{w} = 3$ | $0.77_{\pm 0.0007}$ | $76.61_{\pm 0.11}$ | $1.37_{\pm 0.35}$ | $90.92_{\pm 0.01}$ | $3.25_{\pm 0.10}$ |
| | $\overline{w} = 4$ | $0.81_{\pm 0.0007}$ | $75.48_{\pm 0.11}$ | $-3.41_{\pm 0.36}$ | $90.48_{\pm 0.01}$ | $-1.44_{\pm 0.11}$ |
| | $\overline{w} = 5$ | $0.85_{\pm 0.0007}$ | $74.12_{\pm 0.12}$ | $-9.14_{\pm 0.37}$ | $89.96_{\pm 0.01}$ | $-7.05_{\pm 0.12}$ |
| PGDM$_{\text{GDE}}$ | $\overline{w} = 0$ | $0.90_{\pm 0.0009}$ | $72.82_{\pm 0.15}$ | $-14.62_{\pm 0.34}$ | $89.45_{\pm 0.01}$ | - |
| | $\overline{w} = 1$ | $0.81_{\pm 0.0009}$ | $75.42_{\pm 0.13}$ | $-3.65_{\pm 0.32}$ | $90.46_{\pm 0.01}$ | $9.57_{\pm 0.03}$ |
| | $\overline{w} = 2$ | $0.82_{\pm 0.0006}$ | $75.28_{\pm 0.13}$ | $-4.27_{\pm 0.33}$ | $90.41_{\pm 0.01}$ | $9.03_{\pm 0.05}$ |
| | $\overline{w} = 3$ | $0.85_{\pm 0.0005}$ | $74.34_{\pm 0.13}$ | $-8.23_{\pm 0.33}$ | $90.04_{\pm 0.01}$ | $5.58_{\pm 0.07}$ |
| | $\overline{w} = 4$ | $0.89_{\pm 0.0007}$ | $73.00_{\pm 0.14}$ | $-13.87_{\pm 0.40}$ | $89.52_{\pm 0.01}$ | $0.66_{\pm 0.10}$ |
| | $\overline{w} = 5$ | $0.94_{\pm 0.0009}$ | $71.44_{\pm 0.15}$ | $-20.47_{\pm 0.44}$ | $88.92_{\pm 0.01}$ | $-5.10_{\pm 0.10}$ |

Table 15: Mean absolute error (MAE) of PGDM$_{\text{MAE}}$, PGDM$_{\text{GDE}}$, and baselines on the lock dancing genre of the AIST++ dataset. Percent improvements over baselines are shown in the $\Delta$ MAE (%) columns. Mean and standard deviation are taken across five samples.

| Model | | MAE (dB) | $\Delta$ MAE (%) vs. TimeGrad | $\Delta$ MAE (%) vs. CSDI | $\Delta$ MAE (%) vs. ARMD | $\Delta$ MAE (%) vs. $\overline{w} = 0$ |
|---|---|---|---|---|---|---|
| TimeGrad | | $3.03_{\pm 0.0086}$ | - | - | - | - |
| CSDI | | $0.76_{\pm 0.0028}$ | - | - | - | - |
| ARMD | | $7.65_{\pm 0.0031}$ | - | - | - | - |
| PGDM$_{\text{MAE}}$ | $\overline{w} = 0$ | $0.72_{\pm 0.0011}$ | $76.12_{\pm 0.10}$ | $4.24_{\pm 0.23}$ | $90.53_{\pm 0.01}$ | - |
| | $\overline{w} = 1$ | $0.67_{\pm 0.0006}$ | $78.05_{\pm 0.07}$ | $11.98_{\pm 0.28}$ | $91.29_{\pm 0.01}$ | $8.07_{\pm 0.12}$ |
| | $\overline{w} = 2$ | $0.68_{\pm 0.0003}$ | $77.70_{\pm 0.06}$ | $10.60_{\pm 0.32}$ | $91.15_{\pm 0.01}$ | $6.63_{\pm 0.16}$ |
| | $\overline{w} = 3$ | $0.71_{\pm 0.0007}$ | $76.67_{\pm 0.07}$ | $6.47_{\pm 0.29}$ | $90.75_{\pm 0.01}$ | $2.32_{\pm 0.13}$ |
| | $\overline{w} = 4$ | $0.75_{\pm 0.0008}$ | $75.26_{\pm 0.09}$ | $0.81_{\pm 0.27}$ | $90.19_{\pm 0.01}$ | $-3.59_{\pm 0.12}$ |
| | $\overline{w} = 5$ | $0.80_{\pm 0.0011}$ | $73.57_{\pm 0.09}$ | $-5.98_{\pm 0.38}$ | $89.51_{\pm 0.02}$ | $-10.68_{\pm 0.24}$ |
| PGDM$_{\text{GDE}}$ | $\overline{w} = 0$ | $0.78_{\pm 0.0017}$ | $74.17_{\pm 0.09}$ | $-3.57_{\pm 0.29}$ | $89.75_{\pm 0.03}$ | - |
| | $\overline{w} = 1$ | $0.71_{\pm 0.0023}$ | $76.69_{\pm 0.12}$ | $6.54_{\pm 0.23}$ | $90.75_{\pm 0.03}$ | $9.76_{\pm 0.22}$ |
| | $\overline{w} = 2$ | $0.72_{\pm 0.0020}$ | $76.43_{\pm 0.11}$ | $5.50_{\pm 0.34}$ | $90.65_{\pm 0.03}$ | $8.76_{\pm 0.30}$ |
| | $\overline{w} = 3$ | $0.75_{\pm 0.0015}$ | $75.37_{\pm 0.11}$ | $1.23_{\pm 0.22}$ | $90.23_{\pm 0.02}$ | $4.64_{\pm 0.18}$ |
| | $\overline{w} = 4$ | $0.79_{\pm 0.0013}$ | $73.83_{\pm 0.10}$ | $-4.95_{\pm 0.26}$ | $89.62_{\pm 0.02}$ | $-1.33_{\pm 0.13}$ |
| | $\overline{w} = 5$ | $0.85_{\pm 0.0008}$ | $71.88_{\pm 0.09}$ | $-12.73_{\pm 0.35}$ | $88.85_{\pm 0.01}$ | $-8.85_{\pm 0.17}$ |

Table 16: Mean absolute error (MAE) of PGDM$_{\mathrm{MAE}}$, PGDM$_{\mathrm{GDE}}$, and baselines on the middle hip hop dancing genre of the AIST++ dataset. Percent improvements over baselines are shown in the $\Delta$ MAE (%) columns. Mean and standard deviation are taken across five samples.

| Model | | MAE (dB) | $\Delta$ MAE (%) vs. TimeGrad | $\Delta$ MAE (%) vs. CSDI | $\Delta$ MAE (%) vs. ARMD | $\Delta$ MAE (%) vs. $\overline{w} = 0$ |
|---|---|---|---|---|---|---|
| TimeGrad | | $3.35_{\pm 0.0113}$ | - | - | - | - |
| CSDI | | $1.04_{\pm 0.0048}$ | - | - | - | - |
| ARMD | | $8.98_{\pm 0.0051}$ | - | - | - | - |
| PGDM$_{\mathrm{MAE}}$ | $\overline{w} = 0$ | $0.88_{\pm 0.0014}$ | $73.79_{\pm 0.11}$ | $15.40_{\pm 0.46}$ | $90.21_{\pm 0.01}$ | - |
| | $\overline{w} = 1$ | $0.82_{\pm 0.0009}$ | $75.40_{\pm 0.10}$ | $20.61_{\pm 0.45}$ | $90.82_{\pm 0.01}$ | $6.15_{\pm 0.11}$ |
| | $\overline{w} = 2$ | $0.83_{\pm 0.0013}$ | $75.11_{\pm 0.09}$ | $19.68_{\pm 0.43}$ | $90.71_{\pm 0.01}$ | $5.05_{\pm 0.11}$ |
| | $\overline{w} = 3$ | $0.87_{\pm 0.0008}$ | $74.14_{\pm 0.09}$ | $16.54_{\pm 0.45}$ | $90.34_{\pm 0.01}$ | $1.34_{\pm 0.15}$ |
| | $\overline{w} = 4$ | $0.91_{\pm 0.0011}$ | $72.74_{\pm 0.10}$ | $12.03_{\pm 0.50}$ | $89.82_{\pm 0.01}$ | $-3.98_{\pm 0.20}$ |
| | $\overline{w} = 5$ | $0.97_{\pm 0.0010}$ | $71.07_{\pm 0.10}$ | $6.64_{\pm 0.49}$ | $89.20_{\pm 0.01}$ | $-10.36_{\pm 0.17}$ |
| PGDM$_{\mathrm{GDE}}$ | $\overline{w} = 0$ | $1.05_{\pm 0.0034}$ | $68.67_{\pm 0.13}$ | $-1.11_{\pm 0.37}$ | $88.30_{\pm 0.03}$ | - |
| | $\overline{w} = 1$ | $0.95_{\pm 0.0038}$ | $71.69_{\pm 0.11}$ | $8.64_{\pm 0.25}$ | $89.43_{\pm 0.04}$ | $9.64_{\pm 0.20}$ |
| | $\overline{w} = 2$ | $0.95_{\pm 0.0031}$ | $71.56_{\pm 0.05}$ | $8.21_{\pm 0.32}$ | $89.38_{\pm 0.03}$ | $9.22_{\pm 0.31}$ |
| | $\overline{w} = 3$ | $0.99_{\pm 0.0026}$ | $70.50_{\pm 0.06}$ | $4.78_{\pm 0.36}$ | $88.98_{\pm 0.03}$ | $5.83_{\pm 0.25}$ |
| | $\overline{w} = 4$ | $1.04_{\pm 0.0022}$ | $69.06_{\pm 0.07}$ | $0.16_{\pm 0.45}$ | $88.45_{\pm 0.02}$ | $1.25_{\pm 0.35}$ |
| | $\overline{w} = 5$ | $1.09_{\pm 0.0027}$ | $67.37_{\pm 0.11}$ | $-5.30_{\pm 0.58}$ | $87.82_{\pm 0.02}$ | $-4.14_{\pm 0.36}$ |

Table 17: Mean absolute error (MAE) of PGDM$_{\mathrm{MAE}}$, PGDM$_{\mathrm{GDE}}$, and baselines on the pop dancing genre of the AIST++ dataset. Percent improvements over baselines are shown in the $\Delta$ MAE (%) columns. Mean and standard deviation are taken across five samples.

| Model | | MAE (dB) | $\Delta$ MAE (%) vs. TimeGrad | $\Delta$ MAE (%) vs. CSDI | $\Delta$ MAE (%) vs. ARMD | $\Delta$ MAE (%) vs. $\overline{w} = 0$ |
|---|---|---|---|---|---|---|
| TimeGrad | | $2.55_{\pm 0.0105}$ | - | - | - | - |
| CSDI | | $0.70_{\pm 0.0053}$ | - | - | - | - |
| ARMD | | $6.43_{\pm 0.0019}$ | - | - | - | - |
| PGDM$_{\mathrm{MAE}}$ | $\overline{w} = 0$ | $0.47_{\pm 0.0008}$ | $81.57_{\pm 0.09}$ | $32.95_{\pm 0.58}$ | $92.69_{\pm 0.01}$ | - |
| | $\overline{w} = 1$ | $0.44_{\pm 0.0018}$ | $82.60_{\pm 0.08}$ | $36.71_{\pm 0.68}$ | $93.10_{\pm 0.03}$ | $5.60_{\pm 0.38}$ |
| | $\overline{w} = 2$ | $0.45_{\pm 0.0018}$ | $82.40_{\pm 0.07}$ | $35.97_{\pm 0.67}$ | $93.02_{\pm 0.03}$ | $4.49_{\pm 0.37}$ |
| | $\overline{w} = 3$ | $0.47_{\pm 0.0015}$ | $81.64_{\pm 0.09}$ | $33.20_{\pm 0.69}$ | $92.72_{\pm 0.02}$ | $0.37_{\pm 0.36}$ |
| | $\overline{w} = 4$ | $0.50_{\pm 0.0009}$ | $80.52_{\pm 0.10}$ | $29.14_{\pm 0.66}$ | $92.28_{\pm 0.01}$ | $-5.69_{\pm 0.23}$ |
| | $\overline{w} = 5$ | $0.53_{\pm 0.0007}$ | $79.07_{\pm 0.10}$ | $23.88_{\pm 0.67}$ | $91.70_{\pm 0.01}$ | $-13.54_{\pm 0.23}$ |
| PGDM$_{\mathrm{GDE}}$ | $\overline{w} = 0$ | $0.52_{\pm 0.0013}$ | $79.44_{\pm 0.09}$ | $25.21_{\pm 0.57}$ | $91.85_{\pm 0.02}$ | - |
| | $\overline{w} = 1$ | $0.48_{\pm 0.0018}$ | $81.01_{\pm 0.12}$ | $30.92_{\pm 0.50}$ | $92.47_{\pm 0.03}$ | $7.64_{\pm 0.39}$ |
| | $\overline{w} = 2$ | $0.49_{\pm 0.0016}$ | $80.96_{\pm 0.12}$ | $30.73_{\pm 0.61}$ | $92.45_{\pm 0.03}$ | $7.39_{\pm 0.48}$ |
| | $\overline{w} = 3$ | $0.50_{\pm 0.0014}$ | $80.29_{\pm 0.13}$ | $28.31_{\pm 0.59}$ | $92.19_{\pm 0.02}$ | $4.15_{\pm 0.43}$ |
| | $\overline{w} = 4$ | $0.53_{\pm 0.0011}$ | $79.16_{\pm 0.09}$ | $24.20_{\pm 0.57}$ | $91.74_{\pm 0.02}$ | $-1.35_{\pm 0.45}$ |
| | $\overline{w} = 5$ | $0.57_{\pm 0.0015}$ | $77.76_{\pm 0.11}$ | $19.11_{\pm 0.51}$ | $91.18_{\pm 0.02}$ | $-8.15_{\pm 0.50}$ |

Table 18: Mean absolute error (MAE) of PGDM$_\text{MAE}$, PGDM$_\text{GDE}$, and baselines on the wack dancing genre of the AIST++ dataset. Percent improvements over baselines are shown in the $\Delta$ MAE (%) columns. Mean and standard deviation are taken across five samples.

| Model | | MAE (dB) | $\Delta$ MAE (%) vs. TimeGrad | $\Delta$ MAE (%) vs. CSDI | $\Delta$ MAE (%) vs. ARMD | $\Delta$ MAE (%) vs. $\overline{w}=0$ |
|---|---|---|---|---|---|---|
| TimeGrad | | $1.03_{\pm0.0047}$ | - | - | - | - |
| CSDI | | $0.44_{\pm0.0042}$ | - | - | - | - |
| ARMD | | $3.39_{\pm0.0026}$ | - | - | - | - |
| PGDM$_\text{MAE}$ | $\overline{w}=0$ | $0.44_{\pm0.0044}$ | $57.55_{\pm0.45}$ | $0.61_{\pm1.39}$ | $87.15_{\pm0.12}$ | - |
| | $\overline{w}=1$ | $0.39_{\pm0.0028}$ | $61.54_{\pm0.28}$ | $9.96_{\pm1.37}$ | $88.36_{\pm0.08}$ | $9.40_{\pm0.64}$ |
| | $\overline{w}=2$ | $0.40_{\pm0.0027}$ | $60.53_{\pm0.26}$ | $7.59_{\pm1.36}$ | $88.05_{\pm0.07}$ | $7.01_{\pm0.75}$ |
| | $\overline{w}=3$ | $0.43_{\pm0.0018}$ | $58.40_{\pm0.20}$ | $2.59_{\pm1.19}$ | $87.41_{\pm0.05}$ | $1.99_{\pm0.69}$ |
| | $\overline{w}=4$ | $0.45_{\pm0.0015}$ | $55.70_{\pm0.20}$ | $-3.71_{\pm1.06}$ | $86.59_{\pm0.05}$ | $-4.36_{\pm0.76}$ |
| | $\overline{w}=5$ | $0.48_{\pm0.0019}$ | $52.72_{\pm0.25}$ | $-10.69_{\pm0.96}$ | $85.69_{\pm0.05}$ | $-11.38_{\pm0.83}$ |
| PGDM$_\text{GDE}$ | $\overline{w}=0$ | $0.49_{\pm0.0054}$ | $52.11_{\pm0.44}$ | $-12.12_{\pm1.81}$ | $85.50_{\pm0.16}$ | - |
| | $\overline{w}=1$ | $0.45_{\pm0.0079}$ | $56.17_{\pm0.63}$ | $-2.63_{\pm2.45}$ | $86.73_{\pm0.23}$ | $8.47_{\pm0.75}$ |
| | $\overline{w}=2$ | $0.46_{\pm0.0079}$ | $55.41_{\pm0.61}$ | $-4.40_{\pm2.37}$ | $86.50_{\pm0.24}$ | $6.89_{\pm0.83}$ |
| | $\overline{w}=3$ | $0.48_{\pm0.0075}$ | $53.20_{\pm0.57}$ | $-9.58_{\pm2.32}$ | $85.83_{\pm0.22}$ | $2.28_{\pm0.66}$ |
| | $\overline{w}=4$ | $0.51_{\pm0.0059}$ | $50.33_{\pm0.45}$ | $-16.29_{\pm2.06}$ | $84.96_{\pm0.18}$ | $-3.72_{\pm0.28}$ |
| | $\overline{w}=5$ | $0.54_{\pm0.0046}$ | $46.95_{\pm0.30}$ | $-24.20_{\pm1.83}$ | $83.94_{\pm0.14}$ | $-10.78_{\pm0.55}$ |

Table 19: Continuous ranked probability score (CRPS) of PGDM$_\text{MAE}$ and PGDM$_\text{GDE}$ for both the visual field prediction (UWHVF dataset) and the human motion prediction (10 dance genres from the AIST++ dataset) case studies. Mean and standard deviation are taken across three seeds

| | **PGDM$_\text{MAE}$** | | | | | |
|---|---|---|---|---|---|---|
| | $\overline{w}=0$ | $\overline{w}=1$ | $\overline{w}=2$ | $\overline{w}=3$ | $\overline{w}=4$ | $\overline{w}=5$ |
| UWHVF | $0.794_{\pm0.0122}$ | $0.777_{\pm0.0040}$ | $0.781_{\pm0.0040}$ | $0.786_{\pm0.0040}$ | $0.790_{\pm0.0039}$ | $0.794_{\pm0.0038}$ |
| Break | $0.039_{\pm0.0001}$ | $0.037_{\pm<0.0001}$ | $0.037_{\pm0.0001}$ | $0.040_{\pm0.0002}$ | $0.043_{\pm0.0002}$ | $0.046_{\pm0.0002}$ |
| House | $0.077_{\pm0.0004}$ | $0.073_{\pm0.0002}$ | $0.075_{\pm0.0001}$ | $0.080_{\pm0.0001}$ | $0.085_{\pm0.0001}$ | $0.092_{\pm0.0001}$ |
| Ballet Jazz | $0.039_{\pm0.0001}$ | $0.035_{\pm0.0002}$ | $0.037_{\pm0.0002}$ | $0.040_{\pm0.0003}$ | $0.046_{\pm0.0003}$ | $0.051_{\pm0.0003}$ |
| Street Jazz | $0.050_{\pm0.0002}$ | $0.047_{\pm0.0002}$ | $0.047_{\pm0.0002}$ | $0.049_{\pm0.0002}$ | $0.052_{\pm0.0003}$ | $0.055_{\pm0.0003}$ |
| Krump | $0.079_{\pm0.0003}$ | $0.070_{\pm0.0003}$ | $0.071_{\pm0.0003}$ | $0.076_{\pm0.0003}$ | $0.083_{\pm0.0003}$ | $0.090_{\pm0.0003}$ |
| LA Hip Hop | $0.075_{\pm0.0006}$ | $0.071_{\pm0.0005}$ | $0.073_{\pm0.0005}$ | $0.078_{\pm0.0005}$ | $0.083_{\pm0.0005}$ | $0.089_{\pm0.0005}$ |
| Lock | $0.068_{\pm0.0002}$ | $0.063_{\pm0.0001}$ | $0.066_{\pm0.0002}$ | $0.070_{\pm0.0002}$ | $0.076_{\pm0.0001}$ | $0.083_{\pm0.0001}$ |
| Middle Hip Hop | $0.085_{\pm0.0002}$ | $0.081_{\pm0.0003}$ | $0.084_{\pm0.0002}$ | $0.089_{\pm0.0002}$ | $0.097_{\pm0.0001}$ | $0.104_{\pm0.0001}$ |
| Pop | $0.046_{\pm0.0002}$ | $0.045_{\pm0.0002}$ | $0.046_{\pm0.0002}$ | $0.048_{\pm0.0002}$ | $0.052_{\pm0.0002}$ | $0.057_{\pm0.0002}$ |
| Wack | $0.043_{\pm0.0002}$ | $0.039_{\pm0.0003}$ | $0.042_{\pm0.0002}$ | $0.046_{\pm0.0001}$ | $0.052_{\pm0.0001}$ | $0.057_{\pm0.0001}$ |

| | **PGDM$_\text{GDE}$** | | | | | |
|---|---|---|---|---|---|---|
| | $\overline{w}=0$ | $\overline{w}=1$ | $\overline{w}=2$ | $\overline{w}=3$ | $\overline{w}=4$ | $\overline{w}=5$ |
| UWHVF | $1.887_{\pm0.0119}$ | $0.927_{\pm0.0028}$ | $0.887_{\pm0.0024}$ | $0.857_{\pm0.0020}$ | $0.837_{\pm0.0024}$ | $0.825_{\pm0.0032}$ |
| Break | $0.049_{\pm0.0001}$ | $0.043_{\pm0.0001}$ | $0.043_{\pm0.0001}$ | $0.046_{\pm0.0001}$ | $0.051_{\pm0.0001}$ | $0.056_{\pm0.0001}$ |
| House | $0.085_{\pm0.0005}$ | $0.079_{\pm0.0003}$ | $0.080_{\pm0.0002}$ | $0.085_{\pm0.0001}$ | $0.090_{\pm0.0001}$ | $0.095_{\pm0.0001}$ |
| Ballet Jazz | $0.044_{\pm0.0002}$ | $0.039_{\pm0.0002}$ | $0.040_{\pm0.0002}$ | $0.045_{\pm0.0002}$ | $0.051_{\pm0.0003}$ | $0.058_{\pm0.0003}$ |
| Street Jazz | $0.055_{\pm0.0002}$ | $0.051_{\pm0.0002}$ | $0.052_{\pm0.0001}$ | $0.054_{\pm0.0001}$ | $0.057_{\pm0.0001}$ | $0.061_{\pm0.0001}$ |
| Krump | $0.091_{\pm0.0004}$ | $0.078_{\pm0.0003}$ | $0.080_{\pm0.0004}$ | $0.085_{\pm0.0004}$ | $0.091_{\pm0.0004}$ | $0.097_{\pm0.0004}$ |
| LA Hip Hop | $0.083_{\pm0.0007}$ | $0.077_{\pm0.0005}$ | $0.079_{\pm0.0004}$ | $0.084_{\pm0.0004}$ | $0.090_{\pm0.0004}$ | $0.097_{\pm0.0004}$ |
| Lock | $0.073_{\pm0.0001}$ | $0.066_{\pm0.0001}$ | $0.069_{\pm<0.0001}$ | $0.074_{\pm0.0001}$ | $0.080_{\pm<0.0001}$ | $0.087_{\pm0.0001}$ |
| Middle Hip Hop | $0.097_{\pm0.0003}$ | $0.089_{\pm0.0003}$ | $0.091_{\pm0.0003}$ | $0.097_{\pm0.0002}$ | $0.104_{\pm0.0002}$ | $0.111_{\pm0.0002}$ |
| Pop | $0.051_{\pm0.0002}$ | $0.047_{\pm0.0002}$ | $0.048_{\pm0.0002}$ | $0.051_{\pm0.0002}$ | $0.054_{\pm0.0002}$ | $0.059_{\pm0.0002}$ |
| Wack | $0.049_{\pm0.0002}$ | $0.044_{\pm0.0002}$ | $0.046_{\pm0.0001}$ | $0.050_{\pm0.0002}$ | $0.055_{\pm0.0002}$ | $0.060_{\pm0.0002}$ |

## H  ABLATIONS

We report ablations evaluating the impact of pattern mixing and dynamic guidance and mixing scales here. For these ablations, we fix either or both the maximum guidance scale $\overline{w}$ and the maximum mixing scale $\overline{w^*}$ to the optimal choice reported in Table 5 in Appendix D. We evaluate performance with both MAE and $\text{CRPS}_{\text{SUM}}$.

**Impact of pattern mixing.** In Algorithm 2, the final pattern mixing step mixes the deterministic prediction from the pattern predictor with the probabilistic prediction from the denoiser. The maximum mixing scale $\overline{w^*}$ determines the strength of the pattern signal compared to the denoiser signal in the final forecast. In Tables 20 and 21, we study the effect of pattern mixing by holding $\overline{w}$ constant and varying the mixing scale $\overline{w^*}$. On the UWHVF dataset, we find that a strong mixing signal provides a significant benefit to the prediction. In contrast, on the AIST++ dataset, a weaker mixing signal of 0.0 or 0.2 is preferable, and an excessive mixing scale can degrade performance. This suggests that the pattern prediction model itself makes much more accurate predictions on UWHVF. This observation highlights one benefit of the explicit pattern modeling used by PGDM. On a dataset as small as UWHVF, it is challenging to reliably learn the underlying distribution, as demonstrated by the high CRPS across all models on UWHVF in Table 2. In such low-data regimes, predicting in the low-dimensional pattern space is more sample-efficient, allowing the pattern prediction model to make strong point forecasts even when distributional estimation is unreliable. In this case, a designer may opt to use a higher mixing scale, leading to lowered MAE. We accomplish exactly this in Table 1.

**Impact of dynamic scale.** For both the pattern guidance and pattern mixing, PGDM applies a dynamic scale $w \in (0, \overline{w})$ and $w^* \in (0, \overline{w^*})$, respectively. In Tables 22 and 23, we compare the performances of PGDM and predictions made with a constant scale, holding $\overline{w}$ and $\overline{w^*}$ fixed. Across the UWHVF dataset and most genres of the AIST++ dataset, the dynamic scale reduces the prediction error of PGDM. On the ballet jazz and wack genres of AIST++, dynamic scaling performs comparably to constant scaling. It is likely that in these two cases, fewer novel patterns are seen at inference time. The dynamic scale, which is determined by the uncertainty of the pattern prediction, is most beneficial when the patterns extracted by archetypal analysis do not fully capture the patterns seen at inference time (i.e., distribution shift). The dynamic guidance scale ensures that, when novel patterns occur, the pattern guidance is not followed by the diffusion model. Similarly, the dynamic mixing scale ensures that pattern predictions on novel patterns are not heavily incorporated into the final forecast.

**Impact of pattern prediction.** To address the challenge of dynamically evolving patterns, PGDM relies on a model that predicts the future patterns appearing in the sequence. The most similar related work to PGDM, Diff-MGR (Zhao et al., 2024), does not account for this realistic and common feature of temporal data. Diff-MGR assumes that patterns will remain constant over time. As we were able to obtain neither code nor sufficient implementation details to evaluate Diff-MGR on our applications, we instead perform an ablation study of PGDM where we assume constant patterns as a proxy for this baseline. In Tables 22 and 23, we compare the performances of PGDM and predictions conditioned only on the most recently observed pattern $Ac_A(x_T)$ (i.e., without the pattern prediction model $f_A$). Across UWHVF and all ten genres of the AIST++ dataset, PGDM with $f_A$ outperforms PGDM without $f_A$. On UWHVF, the pattern prediction model adds a slight performance gain, as visual fields and their patterns may progress slowly over several decades (Saunders et al., 2016). In other cases, the performance gap from adding pattern prediction is more significant, e.g., the house, krump, and middle hip hop genres of AIST++. In these particular dance genres, patterns may change more rapidly due to dance style or music tempo. For example, the reported music tempo for the house genre is 110-130 BPM, compared to the 80-130 BPM tempo of the remaining genres (Li et al., 2021). This study highlights the importance of the pattern prediction model in accounting for dynamic patterns.

Table 20: Impact of maximum mixing scale $\overline{w}^*$ on mean absolute error (MAE) of $\text{PGDM}_{\text{MAE}}$ and $\text{PGDM}_{\text{GDE}}$. MAE is reported on UWHVF and all ten genres of the AIST++ dataset with fixed $\overline{w}$. Mean and standard deviation are taken across five samples.

| | $\text{PGDM}_{\text{MAE}}$ with Maximum Mixing Scale $\overline{w}^* =$ | | | | | |
| | **0.0** | **0.2** | **0.4** | **0.6** | **0.8** | **1.0** |
|---|---|---|---|---|---|---|
| **UWHVF** | $3.68_{\pm 0.0415}$ | $3.45_{\pm 0.0352}$ | $3.26_{\pm 0.0293}$ | $3.11_{\pm 0.0230}$ | $3.01_{\pm 0.0168}$ | $2.96_{\pm 0.0117}$ |
| **Break** | $0.39_{\pm 0.0011}$ | $0.38_{\pm 0.0009}$ | $0.41_{\pm 0.0007}$ | $0.48_{\pm 0.0006}$ | $0.56_{\pm 0.0006}$ | $0.65_{\pm 0.0006}$ |
| **House** | $0.74_{\pm 0.0012}$ | $0.74_{\pm 0.0014}$ | $0.83_{\pm 0.0012}$ | $0.97_{\pm 0.0010}$ | $1.14_{\pm 0.0009}$ | $1.34_{\pm 0.0008}$ |
| **Ballet jazz** | $0.39_{\pm 0.0002}$ | $0.38_{\pm 0.0002}$ | $0.40_{\pm 0.0002}$ | $0.45_{\pm 0.0002}$ | $0.52_{\pm 0.0003}$ | $0.59_{\pm 0.0002}$ |
| **Street jazz** | $0.48_{\pm 0.0008}$ | $0.50_{\pm 0.0008}$ | $0.57_{\pm 0.0008}$ | $0.68_{\pm 0.0007}$ | $0.80_{\pm 0.0006}$ | $0.94_{\pm 0.0005}$ |
| **Krump** | $0.70_{\pm 0.0013}$ | $0.71_{\pm 0.0007}$ | $0.80_{\pm 0.0004}$ | $0.92_{\pm 0.0002}$ | $1.08_{\pm 0.0002}$ | $1.25_{\pm 0.0002}$ |
| **LA Hip Hop** | $0.74_{\pm 0.0007}$ | $0.74_{\pm 0.0006}$ | $0.81_{\pm 0.0005}$ | $0.93_{\pm 0.0004}$ | $1.09_{\pm 0.0004}$ | $1.26_{\pm 0.0003}$ |
| **Lock** | $0.67_{\pm 0.0005}$ | $0.67_{\pm 0.0006}$ | $0.74_{\pm 0.0006}$ | $0.86_{\pm 0.0006}$ | $1.01_{\pm 0.0005}$ | $1.18_{\pm 0.0004}$ |
| **Middle Hip Hop** | $0.82_{\pm 0.0009}$ | $0.82_{\pm 0.0009}$ | $0.91_{\pm 0.0010}$ | $1.06_{\pm 0.0010}$ | $1.24_{\pm 0.0010}$ | $1.45_{\pm 0.0010}$ |
| **Pop** | $0.44_{\pm 0.0017}$ | $0.44_{\pm 0.0018}$ | $0.49_{\pm 0.0017}$ | $0.56_{\pm 0.0017}$ | $0.65_{\pm 0.0015}$ | $0.75_{\pm 0.0015}$ |
| **Wack** | $0.41_{\pm 0.0031}$ | $0.39_{\pm 0.0028}$ | $0.41_{\pm 0.0024}$ | $0.44_{\pm 0.0020}$ | $0.50_{\pm 0.0015}$ | $0.56_{\pm 0.0010}$ |

| | $\text{PGDM}_{\text{GDE}}$ with Maximum Mixing Scale $\overline{w}^* =$ | | | | | |
| | **0.0** | **0.2** | **0.4** | **0.6** | **0.8** | **1.0** |
|---|---|---|---|---|---|---|
| **UWHVF** | $4.00_{\pm 0.0217}$ | $3.73_{\pm 0.0205}$ | $3.49_{\pm 0.0198}$ | $3.30_{\pm 0.0193}$ | $3.16_{\pm 0.0180}$ | $3.08_{\pm 0.0153}$ |
| **Break** | $0.46_{\pm 0.0013}$ | $0.45_{\pm 0.0015}$ | $0.47_{\pm 0.0017}$ | $0.52_{\pm 0.0018}$ | $0.59_{\pm 0.0018}$ | $0.68_{\pm 0.0018}$ |
| **House** | $0.83_{\pm 0.0012}$ | $0.82_{\pm 0.0014}$ | $0.89_{\pm 0.0013}$ | $1.01_{\pm 0.0011}$ | $1.18_{\pm 0.0010}$ | $1.36_{\pm 0.0009}$ |
| **Ballet jazz** | $0.45_{\pm 0.0010}$ | $0.42_{\pm 0.0009}$ | $0.44_{\pm 0.0008}$ | $0.48_{\pm 0.0007}$ | $0.53_{\pm 0.0006}$ | $0.61_{\pm 0.0006}$ |
| **Street jazz** | $0.54_{\pm 0.0005}$ | $0.55_{\pm 0.0006}$ | $0.61_{\pm 0.0007}$ | $0.70_{\pm 0.0007}$ | $0.82_{\pm 0.0006}$ | $0.95_{\pm 0.0005}$ |
| **Krump** | $0.79_{\pm 0.0010}$ | $0.79_{\pm 0.0005}$ | $0.85_{\pm 0.0003}$ | $0.96_{\pm 0.0003}$ | $1.10_{\pm 0.0004}$ | $1.26_{\pm 0.0004}$ |
| **LA Hip Hop** | $0.82_{\pm 0.0009}$ | $0.81_{\pm 0.0009}$ | $0.87_{\pm 0.0008}$ | $0.98_{\pm 0.0007}$ | $1.12_{\pm 0.0006}$ | $1.28_{\pm 0.0005}$ |
| **Lock** | $0.71_{\pm 0.0022}$ | $0.71_{\pm 0.0023}$ | $0.77_{\pm 0.0023}$ | $0.89_{\pm 0.0022}$ | $1.03_{\pm 0.0020}$ | $1.19_{\pm 0.0019}$ |
| **Middle Hip Hop** | $0.96_{\pm 0.0040}$ | $0.95_{\pm 0.0038}$ | $1.02_{\pm 0.0036}$ | $1.14_{\pm 0.0034}$ | $1.31_{\pm 0.0032}$ | $1.50_{\pm 0.0030}$ |
| **Pop** | $0.49_{\pm 0.0015}$ | $0.48_{\pm 0.0018}$ | $0.52_{\pm 0.0018}$ | $0.58_{\pm 0.0018}$ | $0.67_{\pm 0.0018}$ | $0.77_{\pm 0.0018}$ |
| **Wack** | $0.47_{\pm 0.0083}$ | $0.45_{\pm 0.0079}$ | $0.46_{\pm 0.0075}$ | $0.49_{\pm 0.0070}$ | $0.55_{\pm 0.0064}$ | $0.61_{\pm 0.0057}$ |

Table 21: Impact of maximum mixing scale $\overline{w}^*$ on continuous ranked probability score (CRPS$_{\text{SUM}}$) of PGDM$_{\text{MAE}}$ and PGDM$_{\text{GDE}}$. CRPS$_{\text{SUM}}$ (lower is better) is reported on UWHVF and all ten genres of the AIST++ dataset with fixed $\overline{w}$. Mean and standard deviation are taken across three seeds.

| | PGDM$_{\text{MAE}}$ with Maximum Mixing Scale $\overline{w}^* =$ | | | | | |
| --- | --- | --- | --- | --- | --- | --- |
| | **0.0** | **0.2** | **0.4** | **0.6** | **0.8** | **1.0** |
| UWHVF | $0.795_{\pm 0.0102}$ | $0.771_{\pm 0.0096}$ | $0.756_{\pm 0.0086}$ | $0.750_{\pm 0.0071}$ | $0.756_{\pm 0.0054}$ | $0.777_{\pm 0.0040}$ |
| Break | $0.037_{\pm <0.0001}$ | $0.037_{\pm <0.0001}$ | $0.041_{\pm <0.0001}$ | $0.048_{\pm <0.0001}$ | $0.059_{\pm 0.0001}$ | $0.072_{\pm 0.0001}$ |
| House | $0.072_{\pm 0.0003}$ | $0.073_{\pm 0.0002}$ | $0.082_{\pm 0.0002}$ | $0.096_{\pm 0.0001}$ | $0.114_{\pm 0.0001}$ | $0.135_{\pm <0.0001}$ |
| Ballet jazz | $0.036_{\pm 0.0002}$ | $0.035_{\pm 0.0002}$ | $0.038_{\pm 0.0002}$ | $0.045_{\pm 0.0001}$ | $0.055_{\pm 0.0001}$ | $0.067_{\pm 0.0001}$ |
| Street jazz | $0.047_{\pm 0.0002}$ | $0.051_{\pm 0.0002}$ | $0.064_{\pm 0.0002}$ | $0.082_{\pm 0.0002}$ | $0.104_{\pm 0.0002}$ | $0.127_{\pm 0.0001}$ |
| Krump | $0.071_{\pm 0.0003}$ | $0.073_{\pm 0.0003}$ | $0.081_{\pm 0.0003}$ | $0.093_{\pm 0.0002}$ | $0.109_{\pm 0.0002}$ | $0.128_{\pm 0.0001}$ |
| LA Hip Hop | $0.070_{\pm 0.0006}$ | $0.071_{\pm 0.0005}$ | $0.078_{\pm 0.0004}$ | $0.090_{\pm 0.0003}$ | $0.106_{\pm 0.0002}$ | $0.125_{\pm 0.0001}$ |
| Lock | $0.062_{\pm 0.0001}$ | $0.063_{\pm 0.0001}$ | $0.071_{\pm 0.0001}$ | $0.084_{\pm 0.0001}$ | $0.102_{\pm 0.0002}$ | $0.122_{\pm 0.0002}$ |
| Middle Hip Hop | $0.078_{\pm 0.0002}$ | $0.081_{\pm 0.0003}$ | $0.090_{\pm 0.0003}$ | $0.106_{\pm 0.0002}$ | $0.126_{\pm 0.0002}$ | $0.149_{\pm 0.0001}$ |
| Pop | $0.044_{\pm 0.0002}$ | $0.045_{\pm 0.0002}$ | $0.049_{\pm 0.0002}$ | $0.056_{\pm 0.0002}$ | $0.065_{\pm 0.0001}$ | $0.077_{\pm 0.0001}$ |
| Wack | $0.039_{\pm 0.0003}$ | $0.039_{\pm 0.0003}$ | $0.042_{\pm 0.0002}$ | $0.050_{\pm 0.0001}$ | $0.060_{\pm 0.0001}$ | $0.072_{\pm 0.0001}$ |

| | PGDM$_{\text{GDE}}$ with Maximum Mixing Scale $\overline{w}^* =$ | | | | | |
| --- | --- | --- | --- | --- | --- | --- |
| | **0.0** | **0.2** | **0.4** | **0.6** | **0.8** | **1.0** |
| UWHVF | $0.888_{\pm 0.0065}$ | $0.847_{\pm 0.0045}$ | $0.816_{\pm 0.0031}$ | $0.801_{\pm 0.0024}$ | $0.803_{\pm 0.0023}$ | $0.825_{\pm 0.0032}$ |
| Break | $0.044_{\pm 0.0001}$ | $0.043_{\pm 0.0001}$ | $0.046_{\pm 0.0001}$ | $0.053_{\pm 0.0001}$ | $0.062_{\pm 0.0001}$ | $0.073_{\pm 0.0001}$ |
| House | $0.078_{\pm 0.0003}$ | $0.079_{\pm 0.0003}$ | $0.086_{\pm 0.0001}$ | $0.098_{\pm 0.0001}$ | $0.116_{\pm <0.0001}$ | $0.136_{\pm <0.0001}$ |
| Ballet jazz | $0.040_{\pm 0.0002}$ | $0.039_{\pm 0.0002}$ | $0.041_{\pm 0.0001}$ | $0.047_{\pm 0.0002}$ | $0.056_{\pm 0.0002}$ | $0.067_{\pm 0.0001}$ |
| Street jazz | $0.052_{\pm 0.0001}$ | $0.055_{\pm 0.0002}$ | $0.067_{\pm 0.0002}$ | $0.084_{\pm 0.0002}$ | $0.105_{\pm 0.0001}$ | $0.128_{\pm 0.0001}$ |
| Krump | $0.077_{\pm 0.0003}$ | $0.078_{\pm 0.0003}$ | $0.084_{\pm 0.0003}$ | $0.094_{\pm 0.0003}$ | $0.110_{\pm 0.0003}$ | $0.128_{\pm 0.0002}$ |
| LA Hip Hop | $0.076_{\pm 0.0006}$ | $0.077_{\pm 0.0005}$ | $0.083_{\pm 0.0004}$ | $0.094_{\pm 0.0003}$ | $0.109_{\pm 0.0003}$ | $0.126_{\pm 0.0002}$ |
| Lock | $0.065_{\pm <0.0001}$ | $0.066_{\pm 0.0001}$ | $0.073_{\pm 0.0001}$ | $0.086_{\pm 0.0001}$ | $0.103_{\pm 0.0002}$ | $0.122_{\pm 0.0002}$ |
| Middle Hip Hop | $0.087_{\pm 0.0003}$ | $0.089_{\pm 0.0003}$ | $0.097_{\pm 0.0003}$ | $0.111_{\pm 0.0002}$ | $0.130_{\pm 0.0001}$ | $0.152_{\pm 0.0001}$ |
| Pop | $0.047_{\pm 0.0001}$ | $0.047_{\pm 0.0002}$ | $0.051_{\pm 0.0002}$ | $0.057_{\pm 0.0002}$ | $0.066_{\pm 0.0001}$ | $0.077_{\pm 0.0001}$ |
| Wack | $0.045_{\pm 0.0002}$ | $0.044_{\pm 0.0002}$ | $0.048_{\pm 0.0002}$ | $0.055_{\pm 0.0002}$ | $0.064_{\pm 0.0002}$ | $0.076_{\pm 0.0002}$ |

Table 22: Impact of dynamic scaling and pattern prediction model $f_A$ on mean absolute error (MAE) of PGDM$_{\text{MAE}}$ and PGDM$_{\text{GDE}}$. MAE is reported for UWHVF and all ten genres of the AIST++ dataset with fixed $\overline{w}$ and $\overline{w}^*$. Mean and standard deviation are taken across five samples.

| | PGDM$_{\text{MAE}}$ | | | PGDM$_{\text{GDE}}$ | | |
| --- | --- | --- | --- | --- | --- | --- |
| | **Const. Scale** | **Without $f_A$** | **PGDM** | **Const. Scale** | **Without $f_A$** | **PGDM** |
| UWHVF | $3.16_{\pm 0.0000}$ | $2.99_{\pm 0.0101}$ | $2.96_{\pm 0.0117}$ | $3.16_{\pm 0.0000}$ | $3.19_{\pm 0.0119}$ | $3.08_{\pm 0.0153}$ |
| Break | $0.39_{\pm 0.0010}$ | $0.41_{\pm 0.0008}$ | $0.38_{\pm 0.0009}$ | $0.46_{\pm 0.0012}$ | $0.49_{\pm 0.0025}$ | $0.45_{\pm 0.0015}$ |
| House | $0.77_{\pm 0.0013}$ | $0.84_{\pm 0.0019}$ | $0.74_{\pm 0.0014}$ | $0.84_{\pm 0.0012}$ | $0.94_{\pm 0.0011}$ | $0.82_{\pm 0.0014}$ |
| Ballet jazz | $0.38_{\pm 0.0002}$ | $0.40_{\pm 0.0006}$ | $0.38_{\pm 0.0002}$ | $0.42_{\pm 0.0010}$ | $0.45_{\pm 0.0002}$ | $0.42_{\pm 0.0009}$ |
| Street jazz | $0.49_{\pm 0.0010}$ | $0.49_{\pm 0.0005}$ | $0.48_{\pm 0.0008}$ | $0.56_{\pm 0.0006}$ | $0.55_{\pm 0.0002}$ | $0.54_{\pm 0.0005}$ |
| Krump | $0.74_{\pm 0.0009}$ | $0.92_{\pm 0.0012}$ | $0.70_{\pm 0.0013}$ | $0.81_{\pm 0.0003}$ | $0.94_{\pm 0.0009}$ | $0.79_{\pm 0.0005}$ |
| LA Hip Hop | $0.77_{\pm 0.0007}$ | $0.81_{\pm 0.0005}$ | $0.74_{\pm 0.0006}$ | $0.84_{\pm 0.0009}$ | $0.89_{\pm 0.0009}$ | $0.81_{\pm 0.0009}$ |
| Lock | $0.69_{\pm 0.0002}$ | $0.75_{\pm 0.0011}$ | $0.67_{\pm 0.0006}$ | $0.72_{\pm 0.0015}$ | $0.79_{\pm 0.0019}$ | $0.71_{\pm 0.0023}$ |
| Middle Hip Hop | $0.86_{\pm 0.0009}$ | $0.94_{\pm 0.0009}$ | $0.82_{\pm 0.0009}$ | $0.97_{\pm 0.0022}$ | $1.10_{\pm 0.0039}$ | $0.95_{\pm 0.0038}$ |
| Pop | $0.47_{\pm 0.0019}$ | $0.46_{\pm 0.0020}$ | $0.44_{\pm 0.0018}$ | $0.50_{\pm 0.0017}$ | $0.51_{\pm 0.0014}$ | $0.48_{\pm 0.0018}$ |
| Wack | $0.39_{\pm 0.0025}$ | $0.41_{\pm 0.0020}$ | $0.39_{\pm 0.0028}$ | $0.45_{\pm 0.0074}$ | $0.47_{\pm 0.0070}$ | $0.45_{\pm 0.0079}$ |

Table 23: Impact of dynamic scaling and pattern prediction model $f_A$ on continuous ranked probability score (CRPS$_{\text{SUM}}$) of PGDM$_{\text{MAE}}$ and PGDM$_{\text{GDE}}$. CRPS$_{\text{SUM}}$ (lower is better) is reported for UWHVF and all ten genres of the AIST++ dataset with fixed $\overline{w}$ and $\overline{w}^*$. Mean and standard deviation are taken across three seeds.

| | PGDM$_{\text{MAE}}$ | | | PGDM$_{\text{GDE}}$ | | |
|---|---|---|---|---|---|---|
| | **Const. Scale** | **Without $f_A$** | **PGDM** | **Const. Scale** | **Without $f_A$** | **PGDM** |
| **UWHVF** | $0.865_{\pm 0.0000}$ | $0.843_{\pm 0.0034}$ | $0.777_{\pm 0.0040}$ | $0.865_{\pm 0.0000}$ | $0.973_{\pm 0.0014}$ | $0.825_{\pm 0.0032}$ |
| **Break** | $0.038_{\pm 0.0001}$ | $0.043_{\pm 0.0001}$ | $0.037_{\pm <0.0001}$ | $0.046_{\pm 0.0001}$ | $0.052_{\pm 0.0001}$ | $0.043_{\pm 0.0001}$ |
| **House** | $0.077_{\pm 0.0002}$ | $0.089_{\pm 0.0003}$ | $0.073_{\pm 0.0002}$ | $0.081_{\pm 0.0001}$ | $0.099_{\pm 0.0002}$ | $0.079_{\pm 0.0003}$ |
| **Ballet jazz** | $0.036_{\pm 0.0002}$ | $0.039_{\pm 0.0001}$ | $0.035_{\pm 0.0002}$ | $0.039_{\pm 0.0002}$ | $0.042_{\pm 0.0001}$ | $0.039_{\pm 0.0002}$ |
| **Street jazz** | $0.049_{\pm 0.0002}$ | $0.051_{\pm 0.0002}$ | $0.047_{\pm 0.0002}$ | $0.054_{\pm 0.0001}$ | $0.056_{\pm 0.0002}$ | $0.052_{\pm 0.0001}$ |
| **Krump** | $0.078_{\pm 0.0003}$ | $0.133_{\pm 0.0007}$ | $0.071_{\pm 0.0003}$ | $0.081_{\pm 0.0004}$ | $0.122_{\pm 0.0005}$ | $0.078_{\pm 0.0003}$ |
| **LA Hip Hop** | $0.075_{\pm 0.0004}$ | $0.085_{\pm 0.0006}$ | $0.071_{\pm 0.0005}$ | $0.080_{\pm 0.0004}$ | $0.093_{\pm 0.0005}$ | $0.077_{\pm 0.0005}$ |
| **Lock** | $0.066_{\pm 0.0002}$ | $0.079_{\pm 0.0002}$ | $0.063_{\pm 0.0001}$ | $0.068_{\pm 0.0001}$ | $0.082_{\pm 0.0001}$ | $0.066_{\pm 0.0001}$ |
| **Middle Hip Hop** | $0.086_{\pm 0.0003}$ | $0.107_{\pm 0.0002}$ | $0.081_{\pm 0.0003}$ | $0.093_{\pm 0.0003}$ | $0.122_{\pm 0.0006}$ | $0.089_{\pm 0.0003}$ |
| **Pop** | $0.047_{\pm 0.0002}$ | $0.049_{\pm 0.0002}$ | $0.045_{\pm 0.0002}$ | $0.049_{\pm 0.0002}$ | $0.052_{\pm 0.0003}$ | $0.047_{\pm 0.0002}$ |
| **Wack** | $0.039_{\pm 0.0002}$ | $0.044_{\pm 0.0003}$ | $0.039_{\pm 0.0003}$ | $0.045_{\pm 0.0002}$ | $0.050_{\pm 0.0003}$ | $0.044_{\pm 0.0002}$ |

