# OpenReview forum: "Pattern-Guided Diffusion Models"
_ICLR.cc/2026/Conference — Submitted to ICLR 2026_

### Official Review · Reviewer_KRkT · 2025-10-19

**Soundness:** 3
**Presentation:** 3
**Contribution:** 2
**Rating:** 6
**Confidence:** 3

**Summary:**

This paper introduces Pattern-Guided Diffusion Models (PGDM), a framework that enhances diffusion-based time-series forecasting by explicitly leveraging recurring patterns inherent in temporal data. The authors first apply Archetypal Analysis (AA) to extract interpretable “archetype” patterns from training data and represent each data as a convex combination of these archetypes. A lightweight neural network then predicts future pattern coefficients, which guide a diffusion model to generate realistic future sequences. To dynamically control how much the model relies on pattern guidance, the paper proposes Archetypal Analysis Uncertainty Quantification (AAUQ), which measures geometric distance from the training distribution and adjusts guidance strength accordingly.

**Strengths:**

(1) The proposed model enhances diffusion-based time-series forecasting by explicitly leveraging recurring patterns inherent in temporal data, where such "recurring patterns" is common in real-world datasets.

(2) The solution is well-motivated and the proposed model is technically sound although no novel components is proposed compared to previous work.

(3) Evaluation on two very different tasks—medical vision fields and motion capture—demonstrates the model’s generality and consistent performance gains over multiple baselines.

**Weaknesses:**

(1) The need to first learn the pattern estimator (via AA and the guidance network) before training the diffusion model introduces potential error propagation and optimization complexity. Is there any way to conduct such process in an end-to-end manner?

(2) Intuitively, the input of the proposed model and other diffusion-based model in the same, i.e., the time series data, the only difference is the proposed model add addtional ``predicted pattern'' from the raw time series. In that sense, no additional information is introduced for the proposed model. And for other diffusion-based models, we can consider them conducting the "pattern prediction" implicitly. When there is enough data, other diffusion-based models should be able to learn a good pattern prediction implicitly in end-to-end manner. In that sense, other diffusion-based models should performance no different compared to the proposed model. Can the author explain more on why the proposed model get better performance?

**Questions:**

please see the weakness above.

---

> ### Author Response · Authors · 2025-11-21
> **Response Part 1**
>
> Thank you for your comments. Based on feedback from all of the reviewers, we have added ablation studies. We have also added the continuous ranked probability score (CRPS) as an evaluation metric. These new results are included in the updated PDF of our submission with discussion. We are currently working on adding a new baseline, and we will add those results before the end of the rebuttal period.
>
> Additionally, while adding the ablation experiments, we noticed that in generating the human motion prediction results reported in the original manuscript, we accidentally omitted Line 10 of Algorithm 2 (i.e., the final pattern mixing step). After correcting our implementation, we found that using a maximum mixing scale of 1 (see Line 9 of Algorithm 2) provided too strong of a pattern signal in the final prediction. In our ablations, we therefore performed a sweep over maximum guidance scale $\bar{w} \in {0, 1, 2, 3, 4, 5}$ and maximum mixing scale in $\bar{w*} \in {0, 0.2, 0.4, 0.6, 0.8, 1.0}$. We have updated Table 1 to report the best performance over this range of $\bar{w}$ and $\bar{w*}$, which has led to a very minor improvement in our results on the AIST++ dataset. Overall, pattern guidance reduces PGDM’s prediction error by up to 14.12% (previously 11.10%), and PGDM achieves lower error by up to 82.60% (previously 82.54%). These updates have also been propagated to the tables in Appendix G.  Among our ablations, we additionally report sweeps over $\bar{w}$ and $\bar{w*}$. Our conclusions are not affected by this update, as the updated results show improved performance.
>
> In summary, the major changes to our PDF submission are as follows:
> * Updated Table 1 with corrected AIST++ results
> * Added probabilistic forecasting metric, continuous ranked probability score (CRPS), in Table 2 (Table 2 of previous version is now Table 3)
> * Added ablations for pattern mixing and dynamic scaling, discussed in Section 5.2 and fully reported in Appendix H
> * Minor edits for clarity and conciseness based on reviewer feedback and 10-page limit
>
> **Question 1**
>
> We thank the reviewer for this idea. Indeed, error propagation is a challenge in any multi-stage training approach such as ours. Our dynamic guidance scale, computed based on the estimated uncertainty of the pattern prediction, helps to mitigate some of these effects. However, conducting this process in an end-to-end manner would be worth exploring in the future. For example, the archetype extraction, guidance network, and diffusion model may be learned jointly. Many considerations make this a non-trivial but interesting task. First, archetypes are extracted through a constrained optimization problem. While a neural network may be able to approximate the solution to this problem, fulfilling soft constraints rather than hard constraints, this may limit our ability to apply Theorem 2 which motivates much of PGDM. On a related note, PGDM assumes a fixed set of archetypes extracted before training, as this “basis” is how uncertainty is quantified. By training the entire pipeline end-to-end, this assumption no longer holds. Finally, even if we separate archetype extraction and subsequently train the guidance network jointly with the diffusion model, we may still encounter challenges optimizing two networks at once.

---

> > ### Author Response · Authors · 2025-11-21
> > **Response Part 2**
> >
> > **Question 2**
> >
> > We thank the reviewer for this insightful question. Yes, traditional diffusion-models likely conduct this “pattern prediction” implicitly to some extent. A number of benefits to our explicit pattern extraction and prediction explain PGDM’s superior performance compared to baselines.
> >
> > By nature, the baseline models, which may implicitly learn patterns, would learn a set of average patterns most commonly appearing across the training distribution. In contrast, archetypal analysis prioritizes extracting extremal patterns. By using these extremal patterns, PGDM is therefore able to handle outliers or long-tail patterns in the data, unlike models learning patterns implicitly.
> >
> > Another benefit of PGDM is that, by extracting patterns explicitly and learning the “dynamics” of the data in the lower-dimensional pattern space, the model requires less data to make accurate point forecasts. This is reflected in PGDM’s ability to outperform baselines in terms of MAE on the UWHVF dataset, which contains only about 4,000 training samples. This advantage, however, does not necessarily extend to probabilistic forecasting. For our added metric, continuous ranked probability score (CRPS), PGDM surpasses TimeGrad and CSDI across AIST++, but not on UWHVF. Notably, all models, including baselines, achieve CRPS values on UWHVF that are roughly an order of magnitude higher than on AIST++. This suggests that this small dataset provides limited signal for learning well-calibrated predictive distributions. We emphasize that PGDM achieves the lowest MAE on UWHVF, demonstrating that explicit pattern modeling improves point prediction even when the distribution itself is challenging to estimate.
> >
> > Finally, explicit pattern extraction and prediction aids in interpretability. For example, this may be of particular use in high-risk clinical domains. In our visual field application, providing both a final prediction and our intermediate pattern prediction may aid a clinician in evaluating the result and using it in downstream clinical tasks.

---

> ### Comment · Reviewer_KRkT · 2025-11-23
>
> Thanks for the response. No major issues remain from my side.

---

> > ### Author Response · Authors · 2025-12-03
> > **Comment**
> >
> > Thank you for your response. We are pleased that we have addressed your concerns.

---

### Official Review · Reviewer_LqcP · 2025-10-24

**Soundness:** 2
**Presentation:** 2
**Contribution:** 1
**Rating:** 2
**Confidence:** 4

**Summary:**

This paper proposes Pattern-Guided Diffusion Models (PGDM), a time-series forecasting approach that combines Archetypal Analysis (AA) with diffusion models. The method first projects input sequences into an archetype space, uses a lightweight neural network to predict future archetype coefficients, and injects them as conditional guidance during diffusion. The paper also introduces an uncertainty metric AAUQ to dynamically adjust the guidance strength. Experiments are conducted on visual field prediction and human motion prediction.

**Strengths:**

1. The idea of incorporating archetypal analysis into diffusion-based forecasting is conceptually interesting and adds a degree of interpretability.
2. The paper is clearly written and well-structured; the method is presented coherently with theoretical support and illustrative figures.

**Weaknesses:**

1. **Insufficient experimental design.** The experimental scope is small (only two datasets in relatively narrow domains); the baselines are dated and omit stronger modern models (e.g., recent diffusion- or Transformer-based forecasters); the reason for selecting the two variants $PGDM_{MAE}$ and $PGDM_{GDE}$ are unclear; and there is a lack of comprehensive ablations to establish the contribution of key components (archetype space, AAUQ weighting, etc.).
2. **Limited analysis and discussion.** The narrative largely describes *what* was done and *what* the results are, with limited discussion of *why* these design choices are appropriate and *why* the observed results occur.
3. **Limited methodological novelty.** In essence, the method adds a predictive module as a conditional signal within an existing diffusion framework. Since using external predictive signals with classifier-free guidance is already known to be effective, the contribution feels more stylistic than fundamentally novel. The paper should provide deeper justification and empirical evidence, explaining why this specific form of “pattern guidance” is principled and how it outperforms standard conditional diffusion in practice.

**Questions:**

See Weaknesses.

---

> ### Author Response · Authors · 2025-11-21
> **Response Part 1**
>
> Thank you for your comments. Based on feedback from all of the reviewers, we have added ablation studies. We have also added the continuous ranked probability score (CRPS) as an evaluation metric. These new results are included in the updated PDF of our submission with discussion, and they summarized in our responses to the relevant questions below. We are currently working on adding a new baseline, and we will add those results before the end of the rebuttal period.
>
> Additionally, while adding the ablation experiments, we noticed that in generating the human motion prediction results reported in the original manuscript, we accidentally omitted Line 10 of Algorithm 2 (i.e., the final pattern mixing step). After correcting our implementation, we found that using a maximum mixing scale of 1 (see Line 9 of Algorithm 2) provided too strong of a pattern signal in the final prediction. In our ablations, we therefore performed a sweep over maximum guidance scale $\bar{w} \in {0, 1, 2, 3, 4, 5}$ and maximum mixing scale in $\bar{w*} \in {0, 0.2, 0.4, 0.6, 0.8, 1.0}$. We have updated Table 1 to report the best performance over this range of $\bar{w}$ and $\bar{w*}$, which has led to a very minor improvement in our results on the AIST++ dataset. Overall, pattern guidance reduces PGDM’s prediction error by up to 14.12% (previously 11.10%), and PGDM achieves lower error by up to 82.60% (previously 82.54%). These updates have also been propagated to the tables in Appendix G.  Among our ablations, we additionally report sweeps over $\bar{w}$ and $\bar{w*}$. Our conclusions are not affected by this update, as the updated results show improved performance.
>
> In summary, the major changes to our PDF submission are as follows:
> * Updated Table 1 with corrected AIST++ results
> * Added probabilistic forecasting metric, continuous ranked probability score (CRPS), in Table 2 (Table 2 of previous version is now Table 3)
> * Added ablations for pattern mixing and dynamic scaling, discussed in Section 5.2 and fully reported in Appendix H
> * Minor edits for clarity and conciseness based on reviewer feedback and 10-page limit

---

> > ### Author Response · Authors · 2025-11-21
> > **Response Part 2**
> >
> > **Question 1**
> >
> > We agree that ablations and a more timely baseline are necessary. We will add a new baseline before the discussion period ends. We have performed ablation studies evaluating the impact of pattern mixing and dynamic guidance on PGDM.
> >
> > *Impact of pattern mixing:* In Algorithm 2, the final pattern mixing step mixes the deterministic prediction from the pattern predictor with the probabilistic prediction from the denoiser. The maximum mixing scale $\overline{w^\*}$ determines the strength of the pattern signal compared to the denoiser signal in the final forecast. In Table 18, we study the effect of pattern mixing by holding $\overline{w}$ constant and varying the mixing scale $\overline{w^\*}$. On the UWHVF dataset, we find that a strong mixing signal provides a significant benefit to the prediction. In contrast, on the AIST++ dataset, a weaker mixing signal of 0.0 or 0.2 is preferable, and an excessive mixing scale can degrade performance. This suggests that the pattern prediction model itself makes much more accurate predictions on UWHVF. This observation highlights one benefit of the explicit pattern modeling used by PGDM. On a dataset as small as UWHVF, it is challenging to reliably learn the underlying distribution, as demonstrated by the high CRPS across all models on UWHVF in Table 2 of our revised PDF. In such low-data regimes, predicting in the low-dimensional pattern space is more sample-efficient, allowing the pattern prediction model to make strong point forecasts even when distributional estimation is unreliable. In this case, a designer may opt to use a higher mixing scale, leading to lowered MAE. We accomplish exactly this in Table 1 of our revised PDF.
> >
> > *Table 18: Impact of mixing scale $\overline{w^∗}$ on PGDM$_{\rm MAE}$ with fixed $\overline{w}$. Mean absolute error (MAE) on UWHVF and all ten genres of the AIST++ dataset. Mean and standard deviation are taken across five samples.*
> > |  |$\overline{w^∗}$ =0.0 |$\overline{w^∗}$ =0.2| $\overline{w^∗}$ =0.4 |$\overline{w^∗}$ =0.6 |$\overline{w^∗}$ =0.8 |$\overline{w^∗}$ =1.0|
> > |---|---|---|---|---|---|---|
> > |UWHVF |3.68±0.0415 |3.45±0.0352 |3.26±0.0293 |3.11±0.0230 |3.01±0.0168 |**2.96±0.0117**|
> > |Break |0.39±0.0011| **0.38±0.0009** |0.41±0.0007 |0.48±0.0006 |0.56±0.0006 |0.65±0.0006|
> > |House |**0.74±0.0012** |**0.74±0.0014** |0.83±0.0012| 0.97±0.0010 |1.14±0.0009 |1.34±0.0008|
> > |Ballet jazz |0.39±0.0002| **0.38±0.0002**| 0.40±0.0002 |0.45±0.0002| 0.52±0.0003| 0.59±0.0002|
> > |Street jazz |**0.48±0.0008** |0.50±0.0008 |0.57±0.0008 |0.68±0.0007 |0.80±0.0006 |0.94±0.0005|
> > |Krump |**0.70±0.0013** |0.71±0.0007 |0.80±0.0004 |0.92±0.0002 |1.08±0.0002 |1.25±0.0002|
> > |LA Hip Hop |**0.74±0.0007** |**0.74±0.0006**| 0.81±0.0005| 0.93±0.0004| 1.09±0.0004| 1.26±0.0003|
> > |Lock |**0.67±0.0005** |**0.67±0.0006**| 0.74±0.0006 |0.86±0.0006 |1.01±0.0005 |1.18±0.0004|
> > |Middle Hip Hop |**0.82±0.0009** |**0.82±0.0009** |0.91±0.0010 |1.06±0.0010 |1.24±0.0010| 1.45±0.0010|
> > |Pop |**0.44±0.0017** |**0.44±0.0018** |0.49±0.0017 |0.56±0.0017 |0.65±0.0015 |0.75±0.0015|
> > |Wack |0.41±0.0031 |**0.39±0.0028** |0.41±0.0024 |0.44±0.0020| 0.50±0.0015 |0.56±0.0010|

---

> > > ### Author Response · Authors · 2025-11-21
> > > **Response Part 3**
> > >
> > > *Impact of dynamic scaling:* For both the pattern guidance and pattern mixing, PGDM applies a dynamic scale $w\in(0, \overline{w})$ and $w^\*\in(0, \overline{w^\*})$, respectively. In Table 19, we compare the MAE of PGDM$\_{\rm MAE}$ with a dynamic scale to that with a constant scale, holding $\overline{w}$ and $\overline{w^\*}$ fixed. Across the UWHVF dataset and most genres of the AIST++ dataset, the dynamic scale reduces the prediction error of PGDM. On the ballet jazz and wack genres of AIST++, dynamic scaling performs comparably to constant scaling. It is likely that in these two cases, fewer novel patterns are seen at inference time. The dynamic scale, which is determined by the uncertainty of the pattern prediction, is most beneficial when the patterns extracted by archetypal analysis do not fully capture the patterns seen at inference time (i.e., distribution shift). The dynamic guidance scale ensures that, when novel patterns occur, the pattern guidance is not followed by the diffusion model. Similarly, the dynamic mixing scale ensures that pattern predictions on novel patterns are not heavily incorporated into the final forecast.
> > >
> > > *Table 19: Impact of dynamic guidance and mixing scales on PGDM$\_{\rm MAE}$ with fixed $\overline{w}$ and $\overline{w^\*}$. Mean absolute error (MAE) on UWHVF and all ten genres of the AIST++ dataset. Mean and standard deviation are taken across five samples.*
> > > | | Constant Scale | Dynamic Scale |
> > > |---|---|---|
> > > |UWHVF |3.16±0.0000 |**2.96±0.0117**|
> > > |Break |0.39±0.0010 |**0.38±0.0009**|
> > > |House |0.77±0.0013 |**0.74±0.0014**|
> > > |Ballet jazz |**0.38±0.0002**|**0.38±0.0002**|
> > > |Street jazz |0.49±0.0010 |**0.48±0.0008**|
> > > |Krump |0.74±0.0009 |**0.70±0.0013**|
> > > |LA Hip Hop |0.77±0.0007 |**0.74±0.0006**|
> > > |Lock| 0.69±0.0002 |**0.67±0.0006**|
> > > |Middle Hip Hop |0.86±0.0009 |**0.82±0.0009**|
> > > |Pop |0.47±0.0019 |**0.44±0.0018**|
> > > |Wack| **0.39±0.0025** |**0.39±0.0028**|
> > >
> > > These are reported in full in Appendix H and discussed in Section 5.2 of our revised PDF.
> > >
> > > As for the purpose of selecting two variants of PGDM, we apologize for the confusion. We presented both the MAE and GDE variants in an effort to better demonstrate the benefits of pattern guidance. While the MAE variant achieves the best performance overall, we found that in many cases, the MAE variant surpasses baselines even without guidance. To better demonstrate the benefits of pattern guidance, we therefore additionally included a trained version of PGDM selected based on the GDE criterion. We found that this model performed worse than baselines without guidance in a majority of cases, but can almost always outperform baselines with guidance. By showing the performance of the GDE variant, we therefore further emphasize the benefits of guidance. The purpose of the GDE variant is stated in Section 5, but we have included a clearer description in our revised PDF.
> > >
> > > **Question 2**
> > >
> > > Our design choices for PGDM revolve around the challenges of uncertainties in pattern representation, which are not fully addressed by prior work as discussed in Section 2. In an effort to address these challenges, we base our technique around archetypal analysis. By formalizing patterns as archetypes, we are able to define a novel uncertainty metric with a natural geometric interpretation (Theorem 2). This metric furthermore provides a principled way to dynamically scale the guidance level (motivated by Theorem 1), guarding against highly uncertain pattern prediction.
> > >
> > > We agree that further discussion of why the observed results occur are necessary. For this reason, we have included the ablations reported in our answer to your Question 1. In summary, pattern mixing contributes to reducing prediction error, most significantly on small datasets such as UWHVF. Dynamic scaling further reduces prediction error by guarding against novel patterns seen at inference time.
> > >
> > > **Question 3**
> > >
> > > While our methodology uses a classifier-free guidance (CFG) inspired approach, PGDM is distinct from a simple stylistic application of CFG. Most importantly, our specific form of pattern guidance aims to overcome the challenges of uncertain pattern representation. As described in our answer to your Question 2, our use of archetypal analysis enables us to define a novel uncertainty quantification metric with a natural geometric interpretation, as well as a method for dynamically scaling guidance level. Theorems 1 and 2 motivate these design choices in a principled manner. In contrast, CFG alone does not account for uncertainties in the predictive signals provided to the diffusion model.
> > >
> > > We additionally provide empirical evidence of the importance of the dynamic guidance scale in our ablations, as summarized in our answer to your Question 1.

---

> ### Author Response · Authors · 2025-12-03
> **Additional Response Part 1**
>
> We have added one additional baseline (ARMD (Gao, 2025)) and one more ablation study. For completeness, we have also added the CRPS metric and our PGDM$\_{\rm GDE}$ variant to the reported results of our ablation studies, as we reported only MAE of PGDM$\_{\rm MAE}$ in the ablation studies added in our last response. Our findings from our ablation studies are further supported by the reported CRPS and PGDM$\_{\rm GDE}$ results. Finally, we have added timing measurements for PGDM and our baselines. These new results are included in the updated PDF of our submission with discussion, and relevant results are summarized in our responses to the appropriate questions below.
>
> **Additional Response to Question 1:**
>
> We have added one more ablation to study the impact of our pattern prediction model $f_A$.  PGDM without pattern prediction inherently follows the assumption that patterns remain constant over time. This ablation therefore serves as a proxy for Diff-MGR (Zhao, 2024), which follows this same assumption but cannot be evaluated directly due to unavailable code and insufficient implementation details. We compare PGDM without the pattern prediction model $f_A$ (i.e., predictions are conditioned only on historical patterns, rather than predicted future patterns) to that without $f_A$. Across UWHVF and all ten genres of the AIST++ dataset, PGDM with $f_A$ outperforms PGDM without $f_A$.  On UWHVF, the pattern prediction model adds a slight performance gain, as visual fields and their patterns may progress slowly over several decades (Saunders, 2016). In other cases, the performance gap from adding pattern prediction is more significant, e.g., the house, krump, and middle hip hop genres of AIST++. In these particular dance genres, patterns may change more rapidly due to dance style or music tempo. For example, the reported music tempo for the house genre is 110-130 BPM, compared to the 80-130 BPM tempo of the remaining genres (Li, 2021). This study highlights the importance of the pattern prediction model in accounting for the highly dynamic patterns that commonly appear in realistic temporal data.
>
> For brevity, we report below the MAE of PGDM$\_{\rm MAE}$ for this ablation. However, the MAE and CRPS for both PGDM$\_{\rm MAE}$ and PGDM$\_{\rm GDE}$ are included with discussion in Section 5.2 and Appendix H of our revised PDF. Our conclusions remain consistent across both metrics and both variants of PGDM.
>
> *Table: Impact of dynamic scaling and pattern prediction model $f\_A$ on mean absolute error (MAE) of PGDM$_{\rm MAE}$. MAE is reported for UWHVF and all ten genres of the AIST++ dataset with fixed $\overline{w}$ and $\overline{w^\*}$. Mean and standard deviation are taken across five samples.*
>
> ||Without $f\_A$ | With $f\_A$|
> |---|---|---|
> |UWHVF | 2.99±0.0101 |**2.96±0.0117**
> |Break | 0.41±0.0008 |**0.38±0.0009**
> |House | 0.84±0.0019 |**0.74±0.0014**
> |Ballet jazz |0.40±0.0006 |**0.38±0.0002**
> |Street jazz |0.49±0.0005 |**0.48±0.0008**
> |Krump | 0.92±0.0012 |**0.70±0.0013**
> |LA Hip Hop | 0.81±0.0005 |**0.74±0.0006**
> |Lock |0.75±0.0011 |**0.67±0.0006**
> |Middle Hip Hop |0.94±0.0009 |**0.82±0.0009**
> |Pop |0.46±0.0020 |**0.44±0.0018**
> |Wack |0.41±0.0020 |**0.39±0.0028**
>
> We have additionally added a more recent baseline, Auto-regressive moving diffusion models (ARMD), published at AAAI 2025. ARMD is a diffusion-based method that formulates the forward diffusion process as evolving the future series into the historical series, and the reverse process as a devolution from the historical series to the future series. By applying this devolution to time series data, ARMD can forecast future data. We report the MAE and CRPS of ARMD and PGDM$_{\rm MAE}$ below. In summary, PGDM outperforms ARMD in terms of both MAE and CRPS, by up to 93.64% and 92.55%, respectively. The greatest performance gap is on the motion capture frame prediction task. A likely explanation is that ARMD’s design is motivated by the linear ARMA model, which assumes that future data can be constructed as a linear combination of past data and noise terms. This assumption is well aligned with smooth, trend dominated time series (e.g., visual field progression), but limits ARMD’s expressiveness for the rapid, nonlinear dynamics of the AIST++ dancing data. In contrast, PGDM explicitly models pattern progression in a lower dimensional pattern representation space, allowing it to capture these nonlinear pattern dynamics with lower sample complexity.

---

> > ### Author Response · Authors · 2025-12-03
> > **Additional Response Part 2**
> >
> > *Table: MAE and CRPS$\_{\rm SUM}$ of PGDM$\_{\rm MAE}$ and ARMD for both the visual field prediction (UWHVF dataset) human motion prediction (10 dance genres from AIST++ dataset) applications. For PGDM$\_{\rm MAE}$, we show the guidance and mixing scale result that achieved the lowest error (for $\overline{w}$ and $\overline{w^\*}$ choices, see Appendix H). Mean and standard deviation are taken across five and three samples for MAE and CRPS$\_{\rm SUM}$, respectively*
> >
> > ||ARMD (MAE) | PGDM$_{\rm MAE}$ (MAE) | ARMD (CRPS) | PGDM$_{\rm MAE}$ (CRPS)|
> > |---|---|---|---|---|
> > |UWHVF |3.91±0.0083| **2.96±0.0117** | 0.918±0.0029  | **0.777±0.0040**|
> > |Break |4.26±0.0026 | **0.38±0.0009**| 0.490±0.0006 |**0.037±<0.0001**|
> > |House |9.11±0.0035 | **0.74±0.0014**|0.679±0.0008 |**0.073±0.0002**|
> > |Ballet Jazz |3.38±0.0011| **0.38±0.0002**|0.346±0.0005 |**0.035±0.0002**|
> > |Street Jazz |7.57±0.0049 | **0.48±0.0008**|0.448±0.0004 |**0.047±0.0002**|
> > |Krump |8.90±0.0046| **0.70±0.0013**|0.911±0.0003 |**0.071±0.0003**|
> > |LA Hip Hop | 8.51±0.0032 | **0.74±0.0006**|0.752±0.0014 |**0.071±0.0005**|
> > |Lock |7.65±0.0031 | **0.67±0.0006**|0.652±0.0014 |**0.063±0.0001**|
> > |Middle. Hip Hop |8.98±0.0051 | **0.82±0.0009**|0.832±0.0005 |**0.081±0.0003**|
> > |Pop| 6.43±0.0019 | **0.44±0.0018**|0.460±0.0001|**0.045±0.0002**|
> > |Wack |3.39±0.0026 | **0.39±0.0028**|0.400±0.0012 |**0.039±0.0003**|
> >
> > These added results and discussion can be found in Section 5.1 of our revised PDF.
> >
> > **References:**
> >
> > Jiaxin Gao, Qinglong Cao, and Yuntian Chen. Auto-regressive moving diffusion models for time series forecasting. In Proceedings of the AAAI Conference on Artificial Intelligence, volume 39, pp. 16727–16735, 2025.
> >
> > Ruilong Li, Shan Yang, David A Ross, and Angjoo Kanazawa. Ai choreographer: Music conditioned 3d dance generation with aist++. In Proceedings of the IEEE/CVF international conference on computer vision, pp. 13401–13412, 2021.
> >
> > Luke J Saunders, Felipe A Medeiros, Robert N Weinreb, and Linda M Zangwill. What rates of
> > glaucoma progression are clinically significant? Expert review of ophthalmology, 11(3):227–234,
> > 2016.
> >
> > Tianlong Zhao, Guangle Song, Xuemei Li, Lizhen Cui, and Caiming Zhang. Diff-mgr: Dynamic causal graph attention and pattern reproduction guided diffusion model for multivariate time series probabilistic forecasting. Information Sciences, 675:120742, 2024.

---

### Official Review · Reviewer_qMFo · 2025-10-27

**Soundness:** 3
**Presentation:** 3
**Contribution:** 3
**Rating:** 4
**Confidence:** 3

**Summary:**

The paper introduces Pattern-Guided Diffusion Models (PGDM) for time-series forecasting. The approach first applies archetypal analysis  to construct a low-dimensional pattern space from the training data. A predictor forecasts future pattern contributions, which condition a diffusion model via classifier-free guidance. A data-driven uncertainty score (AAUQ) adaptively modulates the guidance scale, and a lightweight pattern-mixing step at inference further refines samples. The method is evaluated on visual-field progression (UWHVF) and human motion and it consistently improves MAE over strong baselines while providing informative ablations of the guidance weight. The presentation is clear, the theoretical rationale is coherent, and the empirical evidence substantiates the central claims regarding accuracy and stability.

**Strengths:**

1 The work targets recurring temporal structure and argues that guiding diffusion in a low-dimensional pattern space improves efficiency and interpretability, with a succinct, easy-to-follow pipeline.

2 The training and inference procedures are explicit; the guidance mechanism is formalized with equations; and AAUQ provides a principled, data-dependent way to scale guidance rather than relying on a fixed heuristic.

3 Bounds and geometric arguments link the uncertainty proxy to expected guidance behavior, offering credibility beyond purely empirical results.

4 Across two distinct domains, PGDM reduces MAE and variance; ablations reveal a stable operating range and diminishing returns as the guidance scale increases.

**Weaknesses:**

1 The authors argue that AAUQ explicitly modulates guidance based on uncertainty but this paper does not report probabilistic metrics (e.g., CRPS, NLL), coverage (Prediction Interval Coverage Probability, PICP), or calibration diagnostics (e.g., reliability diagrams).

2  The attribution of gains seem ambiguous. I suggest an abalation study for the AA representation or the dynamic guidance against standard alternatives under the same backbone and compute budget,.

3 I think  it would be better to add  parameter counts and basic timing to demonstrate contextualize efficiency. It seems that parameter counts are not reported; hardware and memory are only briefly noted (single 42-GB GPU; <1 GB per model). Training/throughput and sampling latency are also unspecified.

**Questions:**

See weaknesses.

---

> ### Author Response · Authors · 2025-11-21
> **Response Part 1**
>
> Thank you for your comments. Based on feedback from all of the reviewers, we have added ablation studies. We have also added the continuous ranked probability score (CRPS) as an evaluation metric. These new results are included in the updated PDF of our submission with discussion, and they summarized in our responses to the relevant questions below. We are currently working on adding a new baseline, and we will add those results before the end of the rebuttal period.
>
> Additionally, while adding the ablation experiments, we noticed that in generating the human motion prediction results reported in the original manuscript, we accidentally omitted Line 10 of Algorithm 2 (i.e., the final pattern mixing step). After correcting our implementation, we found that using a maximum mixing scale of 1 (see Line 9 of Algorithm 2) provided too strong of a pattern signal in the final prediction. In our ablations, we therefore performed a sweep over maximum guidance scale $\bar{w} \in {0, 1, 2, 3, 4, 5}$ and maximum mixing scale in $\bar{w*} \in {0, 0.2, 0.4, 0.6, 0.8, 1.0}$. We have updated Table 1 to report the best performance over this range of $\bar{w}$ and $\bar{w*}$, which has led to a very minor improvement in our results on the AIST++ dataset. Overall, pattern guidance reduces PGDM’s prediction error by up to 14.12% (previously 11.10%), and PGDM achieves lower error by up to 82.60% (previously 82.54%). These updates have also been propagated to the tables in Appendix G.  Among our ablations, we additionally report sweeps over $\bar{w}$ and $\bar{w*}$. Our conclusions are not affected by this update, as the updated results show improved performance.
>
> In summary, the major changes to our PDF submission are as follows:
> * Updated Table 1 with corrected AIST++ results
> * Added probabilistic forecasting metric, continuous ranked probability score (CRPS), in Table 2 (Table 2 of previous version is now Table 3)
> * Added ablations for pattern mixing and dynamic scaling, discussed in Section 5.2 and fully reported in Appendix H
> * Minor edits for clarity and conciseness based on reviewer feedback and 10-page limit

---

> > ### Author Response · Authors · 2025-11-21
> > **Response Part 2**
> >
> > **Question 1**
> >
> > Thank you for this suggestion. We have added CRPS as an evaluation metric in our experiments, as this is the probabilistic metric reported by TimeGrad and CSDI. Following TimeGrad and CSDI, we report CRPS$_{\rm SUM}$, computed as the CRPS over the summed dimensions of the data. On the AIST++ dataset, guidance reduces CRPS and PGDM surpasses baselines across all ten genres of dance. However, on UWHVF, PGDM does not outperform TimeGrad and CSDI. We note that both PGDM and the baseline models achieve a significantly higher CRPS on UWHVF than AIST++ (one order of magnitude), indicating that this small dataset provides limited signal for learning well-calibrated predictive distributions. We emphasize that PGDM achieves the lowest MAE on UWHVF, demonstrating that explicit pattern modeling improves point prediction even when the distribution itself is challenging to estimate.
> >
> > *Table 2 (UWHVF): CRPS$\_{\rm SUM}$ (lower is better) of PGDM$\_{\rm MAE}$, PGDM$\_{\rm GDE}$, and baselines for the visual field prediction (UWHVF dataset) application. For PGDM$\_{\rm MAE}$ and PGDM$\_{\rm GDE}$ with $\overline{w}>0$, we show the guidance and mixing scale result that achieved the lowest error.*
> > | GenViT | TimeGrad | CSDI | PGDM$_{\rm GDE}$,  $\overline{w}=0$ | PGDM$_{\rm GDE}$,  $\overline{w}>0$ | PGDM$_{\rm MAE}$,  $\overline{w}=0$ | PGDM$_{\rm MAE}$,  $\overline{w}>0$ |
> > |---|---|---|---|---|---|---|
> > |5.119±0.0004 |0.751±0.0020 |**0.658±0.0025**|1.887±0.0119| 0.825±0.0032| 0.794±0.0122 |0.777±0.0040|
> >
> > *Table 2 (AIST++): CRPS$\_{\rm SUM}$ (lower is better) of PGDM$\_{\rm MAE}$, PGDM$\_{\rm GDE}$, and baselines for the human motion prediction (10 dance genres from AIST++ dataset) application. For PGDM$\_{\rm MAE}$ and PGDM$\_{\rm GDE}$ with $\overline{w}>0$, we show the guidance and mixing scale result that achieved the lowest error.*
> >
> > |Genre| TimeGrad |CSDI |  PGDM$\_{\rm GDE}$,  $\overline{w}=0$ | PGDM$\_{\rm GDE}$,  $\overline{w}>0$ | PGDM$\_{\rm MAE}$,  $\overline{w}=0$ | PGDM$\_{\rm MAE}$,  $\overline{w}>0$ |
> > |---|---|---|---|---|---|---|
> > |Break |0.189±0.0010 |0.051±0.0001 |0.049±0.0001 |0.043±0.0001| 0.039±0.0001 |**0.037±<0.0001**|
> > |House |0.340±0.0011 |0.106±0.0002 |0.085±0.0005| 0.079±0.0003| 0.077±0.0004| **0.073±0.0002**|
> > |Ballet Jazz| 0.117±0.0005 |0.053±0.0002 |0.044±0.0002 |0.039±0.0002 |0.039±0.0001 |**0.035±0.0002**|
> > |Street Jazz |0.184±0.0002 |0.058±0.0003| 0.055±0.0002| 0.052±0.0013| 0.050±0.0002| **0.047±0.0002**|
> > |Krump |0.298±0.0010| 0.093±0.0003 |0.091±0.0004 |0.078±0.0003 |0.079±0.0003 |**0.071±0.0003**|
> > |LA Hip Hop| 0.331±0.0004 |0.088±0.0002| 0.083±0.0007 |0.077±0.0005| 0.075±0.0006 |**0.071±0.0005**|
> > |Lock |0.367±0.0004| 0.080±0.0002| 0.073±0.0001| 0.066±0.0001 |0.068±0.0002 |**0.063±0.0001**|
> > |Middle Hip Hop |0.368±0.0018 |0.109±0.0002 |0.097±0.0003| 0.089±0.0003 |0.085±0.0002 |**0.081±0.0003**|
> > |Pop |0.252±0.0004 |0.076±0.0005 |0.051±0.0002 |0.047±0.0002 |0.046±0.0002 |**0.045±0.0002**|
> > |Wack| 0.112±0.0005| 0.051±0.0002| 0.049±0.0002 |0.044±0.0002| 0.043±0.0002| **0.039±0.0003**|
> >
> > These added results and discussion can be found in Table 2 of our revised PDF.
> >
> > **Question 2**
> >
> > We agree that ablation studies are necessary to better understand the role of each distinct module of PGDM. Under the same backbone and compute budget, we have performed ablation studies evaluating the impact of pattern mixing and dynamic guidance on PGDM.
> >
> > *Impact of pattern mixing:* In Algorithm 2, the final pattern mixing step mixes the deterministic prediction from the pattern predictor with the probabilistic prediction from the denoiser. The maximum mixing scale $\overline{w^\*}$ determines the strength of the pattern signal compared to the denoiser signal in the final forecast. In Table 18, we study the effect of pattern mixing by holding $\overline{w}$ constant and varying the mixing scale $\overline{w^\*}$. On the UWHVF dataset, we find that a strong mixing signal provides a significant benefit to the prediction. In contrast, on the AIST++ dataset, a weaker mixing signal of 0.0 or 0.2 is preferable, and an excessive mixing scale can degrade performance. This suggests that the pattern prediction model itself makes much more accurate predictions on UWHVF. This observation highlights one benefit of the explicit pattern modeling used by PGDM. On a dataset as small as UWHVF, it is challenging to reliably learn the underlying distribution, as demonstrated by the high CRPS across all models on UWHVF in Table 2 of our revised PDF. In such low-data regimes, predicting in the low-dimensional pattern space is more sample-efficient, allowing the pattern prediction model to make strong point forecasts even when distributional estimation is unreliable. In this case, a designer may opt to use a higher mixing scale, leading to lowered MAE. We accomplish exactly this in Table 1 of our revised PDF.

---

> > > ### Author Response · Authors · 2025-11-21
> > > **Response Part 3**
> > >
> > > *Table 18: Impact of mixing scale $\overline{w^∗}$ on PGDM$_{\rm MAE}$ with fixed $\overline{w}$. Mean absolute error (MAE) on UWHVF and all ten genres of the AIST++ dataset. Mean and standard deviation are taken across five samples.*
> > > |  |$\overline{w^∗}$ =0.0 |$\overline{w^∗}$ =0.2| $\overline{w^∗}$ =0.4 |$\overline{w^∗}$ =0.6 |$\overline{w^∗}$ =0.8 |$\overline{w^∗}$ =1.0|
> > > |---|---|---|---|---|---|---|
> > > |UWHVF |3.68±0.0415 |3.45±0.0352 |3.26±0.0293 |3.11±0.0230 |3.01±0.0168 |**2.96±0.0117**|
> > > |Break |0.39±0.0011| **0.38±0.0009** |0.41±0.0007 |0.48±0.0006 |0.56±0.0006 |0.65±0.0006|
> > > |House |**0.74±0.0012** |**0.74±0.0014** |0.83±0.0012| 0.97±0.0010 |1.14±0.0009 |1.34±0.0008|
> > > |Ballet jazz |0.39±0.0002| **0.38±0.0002**| 0.40±0.0002 |0.45±0.0002| 0.52±0.0003| 0.59±0.0002|
> > > |Street jazz |**0.48±0.0008** |0.50±0.0008 |0.57±0.0008 |0.68±0.0007 |0.80±0.0006 |0.94±0.0005|
> > > |Krump |**0.70±0.0013** |0.71±0.0007 |0.80±0.0004 |0.92±0.0002 |1.08±0.0002 |1.25±0.0002|
> > > |LA Hip Hop |**0.74±0.0007** |**0.74±0.0006**| 0.81±0.0005| 0.93±0.0004| 1.09±0.0004| 1.26±0.0003|
> > > |Lock |**0.67±0.0005** |**0.67±0.0006**| 0.74±0.0006 |0.86±0.0006 |1.01±0.0005 |1.18±0.0004|
> > > |Middle Hip Hop |**0.82±0.0009** |**0.82±0.0009** |0.91±0.0010 |1.06±0.0010 |1.24±0.0010| 1.45±0.0010|
> > > |Pop |**0.44±0.0017** |**0.44±0.0018** |0.49±0.0017 |0.56±0.0017 |0.65±0.0015 |0.75±0.0015|
> > > |Wack |0.41±0.0031 |**0.39±0.0028** |0.41±0.0024 |0.44±0.0020| 0.50±0.0015 |0.56±0.0010|
> > >
> > > *Impact of dynamic scaling:* For both the pattern guidance and pattern mixing, PGDM applies a dynamic scale $w\in(0, \overline{w})$ and $w^*\in(0, \overline{w^\*})$, respectively. In Table 19, we compare the MAE of PGDM$_{\rm MAE}$ with a dynamic scale to that with a constant scale, holding $\overline{w}$ and $\overline{w^\*}$ fixed. Across the UWHVF dataset and most genres of the AIST++ dataset, the dynamic scale reduces the prediction error of PGDM. On the ballet jazz and wack genres of AIST++, dynamic scaling performs comparably to constant scaling. It is likely that in these two cases, fewer novel patterns are seen at inference time. The dynamic scale, which is determined by the uncertainty of the pattern prediction, is most beneficial when the patterns extracted by archetypal analysis do not fully capture the patterns seen at inference time (i.e., distribution shift). The dynamic guidance scale ensures that, when novel patterns occur, the pattern guidance is not followed by the diffusion model. Similarly, the dynamic mixing scale ensures that pattern predictions on novel patterns are not heavily incorporated into the final forecast.
> > >
> > > *Table 19: Impact of dynamic guidance and mixing scales on PGDM$\_{\rm MAE}$ with fixed $\overline{w}$ and $\overline{w^\*}$. Mean absolute error (MAE) on UWHVF and all ten genres of the AIST++ dataset. Mean and standard deviation are taken across five samples.*
> > > | | Constant Scale | Dynamic Scale |
> > > |---|---|---|
> > > |UWHVF |3.16±0.0000 |**2.96±0.0117**|
> > > |Break |0.39±0.0010 |**0.38±0.0009**|
> > > |House |0.77±0.0013 |**0.74±0.0014**|
> > > |Ballet jazz |**0.38±0.0002**|**0.38±0.0002**|
> > > |Street jazz |0.49±0.0010 |**0.48±0.0008**|
> > > |Krump |0.74±0.0009 |**0.70±0.0013**|
> > > |LA Hip Hop |0.77±0.0007 |**0.74±0.0006**|
> > > |Lock| 0.69±0.0002 |**0.67±0.0006**|
> > > |Middle Hip Hop |0.86±0.0009 |**0.82±0.0009**|
> > > |Pop |0.47±0.0019 |**0.44±0.0018**|
> > > |Wack| **0.39±0.0025** |**0.39±0.0028**|
> > >
> > > These are reported in full in Appendix H and discussed in Section 5.2 of our revised PDF.
> > >
> > > **Question 3**
> > >
> > > Thank you for this suggestion. We have added exact parameter counts to our revised PDF in Appendix D. On the UWHVF dataset, PGDM’s pattern predictor has about 60K parameters, and PGDM’s denoiser has about 400K parameters. On the AIST++ dataset, PGDM’s pattern predictor has less than 300K parameters across all genres, and PGDM’s denoiser has about 400K parameters. PGDM outperforms the significantly larger GenViT (\~4 million parameters on UWHVF), the comparably sized CSDI (\~600K on both datasets), and the smaller TimeGrad (\~60K on both datasets) models.
> > >
> > > For more accurate measurements, we will include timing statistics after our additional experiments have completed.

---

> > > > ### Comment · Reviewer_qMFo · 2025-11-27
> > > >
> > > > I appreciate the authors’ efforts and their comprehensive response. I will raise my score to 6 as an encouragement,

---

> > > > > ### Author Response · Authors · 2025-12-03
> > > > > **Additional Response Part 1**
> > > > >
> > > > > Thank you for increasing your score to a 6. To address any remaining concerns, we have added one additional baseline (ARMD (Gao, 2025)) and one more ablation study. For completeness, we have also added the CRPS metric and our PGDM$\_{\rm GDE}$ variant to the reported results of our ablation studies, as we reported only MAE of PGDM$\_{\rm MAE}$ in the ablation studies added in our last response. Our findings from our ablation studies are further supported by the reported CRPS and PGDM$\_{\rm GDE}$ results. Finally, we have added timing measurements for PGDM and our baselines.
> > > > >
> > > > > We report the evaluation of the new baseline below. In summary, PGDM outperforms ARMD in terms of both MAE and CRPS, by up to 93.64% and 92.55%, respectively. The greatest performance gap is on the motion capture frame prediction task. A likely explanation is that ARMD’s design is motivated by the linear ARMA model, which assumes that future data can be constructed as a linear combination of past data and noise terms. This assumption is well aligned with smooth, trend dominated time series (e.g., visual field progression), but limits ARMD’s expressiveness for the rapid, nonlinear dynamics of the AIST++ dancing data. In contrast, PGDM explicitly models pattern progression in a lower dimensional pattern representation space, allowing it to capture these nonlinear pattern dynamics with lower sample complexity.
> > > > >
> > > > > *Table: MAE and CRPS$\_{\rm SUM}$ of PGDM$\_{\rm MAE}$ and ARMD for both the visual field prediction (UWHVF dataset) human motion prediction (10 dance genres from AIST++ dataset) applications. For PGDM$\_{\rm MAE}$, we show the guidance and mixing scale result that achieved the lowest error (for $\overline{w}$ and $\overline{w^\*}$ choices, see Appendix H). Mean and standard deviation are taken across five and three samples for MAE and CRPS$\_{\rm SUM}$, respectively*
> > > > >
> > > > > ||ARMD (MAE) | PGDM$_{\rm MAE}$ (MAE) | ARMD (CRPS) | PGDM$_{\rm MAE}$ (CRPS)|
> > > > > |---|---|---|---|---|
> > > > > |UWHVF |3.91±0.0083| **2.96±0.0117** | 0.918±0.0029  | **0.777±0.0040**|
> > > > > |Break |4.26±0.0026 | **0.38±0.0009**| 0.490±0.0006 |**0.037±<0.0001**|
> > > > > |House |9.11±0.0035 | **0.74±0.0014**|0.679±0.0008 |**0.073±0.0002**|
> > > > > |Ballet Jazz |3.38±0.0011| **0.38±0.0002**|0.346±0.0005 |**0.035±0.0002**|
> > > > > |Street Jazz |7.57±0.0049 | **0.48±0.0008**|0.448±0.0004 |**0.047±0.0002**|
> > > > > |Krump |8.90±0.0046| **0.70±0.0013**|0.911±0.0003 |**0.071±0.0003**|
> > > > > |LA Hip Hop | 8.51±0.0032 | **0.74±0.0006**|0.752±0.0014 |**0.071±0.0005**|
> > > > > |Lock |7.65±0.0031 | **0.67±0.0006**|0.652±0.0014 |**0.063±0.0001**|
> > > > > |Middle. Hip Hop |8.98±0.0051 | **0.82±0.0009**|0.832±0.0005 |**0.081±0.0003**|
> > > > > |Pop| 6.43±0.0019 | **0.44±0.0018**|0.460±0.0001|**0.045±0.0002**|
> > > > > |Wack |3.39±0.0026 | **0.39±0.0028**|0.400±0.0012 |**0.039±0.0003**|
> > > > >
> > > > > The remaining new results are included in the updated PDF of our submission with discussion, and relevant results are summarized in our responses to the appropriate questions below.
> > > > >
> > > > > **Additional Response to Question 2:**
> > > > >
> > > > > We have added one more ablation to study the impact of our pattern prediction model $f_A$.  PGDM without pattern prediction inherently follows the assumption that patterns remain constant over time. This ablation therefore serves as a proxy for Diff-MGR (Zhao, 2024), which follows this same assumption but cannot be evaluated directly due to unavailable code and insufficient implementation details. We compare PGDM without the pattern prediction model $f_A$ (i.e., predictions are conditioned only on historical patterns, rather than predicted future patterns) to that without $f_A$. Across UWHVF and all ten genres of the AIST++ dataset, PGDM with $f_A$ outperforms PGDM without $f_A$.  On UWHVF, the pattern prediction model adds a slight performance gain, as visual fields and their patterns may progress slowly over several decades (Saunders, 2016). In other cases, the performance gap from adding pattern prediction is more significant, e.g., the house, krump, and middle hip hop genres of AIST++. In these particular dance genres, patterns may change more rapidly due to dance style or music tempo. For example, the reported music tempo for the house genre is 110-130 BPM, compared to the 80-130 BPM tempo of the remaining genres (Li, 2021). This study highlights the importance of the pattern prediction model in accounting for the highly dynamic patterns that commonly appear in realistic temporal data.

---

> > > > > > ### Author Response · Authors · 2025-12-03
> > > > > > **Additional Response Part 2**
> > > > > >
> > > > > > For brevity, we report below the MAE of PGDM$\_{\rm MAE}$ for this ablation. However, the MAE and CRPS for both PGDM$\_{\rm MAE}$ and PGDM$\_{\rm GDE}$ are included with discussion in Section 5.2 and Appendix H of our revised PDF. Our conclusions remain consistent across both metrics and both variants of PGDM.
> > > > > >
> > > > > > *Table: Impact of dynamic scaling and pattern prediction model $f\_A$ on mean absolute error (MAE) of PGDM$_{\rm MAE}$. MAE is reported for UWHVF and all ten genres of the AIST++ dataset with fixed $\overline{w}$ and $\overline{w^\*}$. Mean and standard deviation are taken across five samples.*
> > > > > >
> > > > > > ||Without $f\_A$ | With $f\_A$|
> > > > > > |---|---|---|
> > > > > > |UWHVF | 2.99±0.0101 |**2.96±0.0117**
> > > > > > |Break | 0.41±0.0008 |**0.38±0.0009**
> > > > > > |House | 0.84±0.0019 |**0.74±0.0014**
> > > > > > |Ballet jazz |0.40±0.0006 |**0.38±0.0002**
> > > > > > |Street jazz |0.49±0.0005 |**0.48±0.0008**
> > > > > > |Krump | 0.92±0.0012 |**0.70±0.0013**
> > > > > > |LA Hip Hop | 0.81±0.0005 |**0.74±0.0006**
> > > > > > |Lock |0.75±0.0011 |**0.67±0.0006**
> > > > > > |Middle Hip Hop |0.94±0.0009 |**0.82±0.0009**
> > > > > > |Pop |0.46±0.0020 |**0.44±0.0018**
> > > > > > |Wack |0.41±0.0020 |**0.39±0.0028**
> > > > > >
> > > > > > **Additional Response to Question 3:**
> > > > > >
> > > > > > We have added timing measurements to Appendix D of our revised PDF. In summary, PGDM has higher per-sample inference latency than the baselines GenViT, TimeGrad, CSDI, and (our new baseline) ARMD. The sampling efficiency of our model can be substantially improved using accelerated samplers such as DDIM. Exploring these optimizations is beyond the scope of this work.
> > > > > >
> > > > > > **References:**
> > > > > >
> > > > > > Jiaxin Gao, Qinglong Cao, and Yuntian Chen. Auto-regressive moving diffusion models for time series forecasting. In Proceedings of the AAAI Conference on Artificial Intelligence, volume 39, pp. 16727–16735, 2025.
> > > > > >
> > > > > > Ruilong Li, Shan Yang, David A Ross, and Angjoo Kanazawa. Ai choreographer: Music conditioned 3d dance generation with aist++. In Proceedings of the IEEE/CVF international conference on computer vision, pp. 13401–13412, 2021.
> > > > > >
> > > > > > Luke J Saunders, Felipe A Medeiros, Robert N Weinreb, and Linda M Zangwill. What rates of
> > > > > > glaucoma progression are clinically significant? Expert review of ophthalmology, 11(3):227–234,
> > > > > > 2016.
> > > > > >
> > > > > > Tianlong Zhao, Guangle Song, Xuemei Li, Lizhen Cui, and Caiming Zhang. Diff-mgr: Dynamic causal graph attention and pattern reproduction guided diffusion model for multivariate time series probabilistic forecasting. Information Sciences, 675:120742, 2024.

---

### Official Review · Reviewer_NBP2 · 2025-10-31

**Soundness:** 3
**Presentation:** 1
**Contribution:** 1
**Rating:** 2
**Confidence:** 3

**Summary:**

This paper proposes Patten-Guided Diffusion Models (PGDM), a framework for the diffusion-based forecasting of data sequences that form inherent patterns. PGDM introduces the use of Archetypal Analysis (AA) and learn the generative model with the condition of archetypal patterns. The authors further design a dynamic guidance scale based on the distance between historical samples and the archetypal patterns. Experiments on visual field and human motion prediction tasks demonstrate the effectiveness of PGDM.

**Strengths:**

- The introduction of Archetypal Analysis for capturing inherent patterns is interesting.
- The method is described clearly and is easy to implement.
- The dynamic guidance scale is a thoughtful addition that enhances model performance.

**Weaknesses:**

- The application scope is somewhat narrow and the baseline comparisons could be more extensive.
- The experiment setup is somewhat confusing.
- The dynamic guidance scale is insufficiently effective.

**Questions:**

1. **The scope of application, and baseline methods.** In my opinion, PGDM is build on a prior that data sequences have inherent patterns. However, this assumption may not hold for all types of data sequences. Could the authors discuss the potential limitations of PGDM when applied to data sequences that do not exhibit clear patterns? Furthermore, the baseline methods TimeGrad and CSDI are general-purpose diffusion models for time series forecasting. Could the authors consider including more specialized baseline methods that are specifically designed for pattern-based forecasting to provide a more comprehensive evaluation of PGDM's performance? Additionally, how can PGDM be adapted or extended to handle data sequences that lack inherent patterns?

2. **Experiment setup of MAE/GDE.** The authors describe in Sec.5 that *"The first model ... lowest validation mean absolute error... The secon model ...  highest capacity"*. However, I feel confused about the setup of proposing different model selection criteria. Could the authors clarify why MAE-based and GDE-based models should be simultaneously discussed. Furthermore, the GDE criterion is not clearly described. Could the authors provide more details?

3. **Effectiveness of dynamic guidance scale.** The dynamic guidance scale is designed to adjust the influence of archetypal patterns based on the distance between historical samples and the archetypal patterns. However, in Human the motion prediction task, as demonstrated in Table 1, the improvement of $w>0$ is relatively small. How can the authors explain the limited effectiveness of the dynamic guidance scale in this context? Are there specific scenarios or types of data where the dynamic guidance scale is more beneficial?

4. **Clarity of writing.** While the overall structure of the paper is logical, certain sections could benefit from clearer explanations. For instance, the authors could first provide a detailed high-level overview of data types with inherent patterns (such as Figure 3). This would help readers better understand the motivation behind PGDM before delving into the technical details. Could the authors consider revising these sections to enhance clarity and accessibility for a broader audience?

5. **Novelty of proposed method.** The use of Archetypal Analysis (AA) to capture inherent patterns in data sequences is an interesting approach. However, AA itself is not a novel technique, and I believe it is natural to apply AA in this context. Could the authors elaborate on the specific contributions of PGDM that distinguish it from existing methods that utilize AA or similar techniques? How does PGDM advance the state-of-the-art in diffusion-based forecasting beyond the application of AA?

---

> ### Author Response · Authors · 2025-11-21
> **Response Part 1**
>
> Thank you for your comments. Based on feedback from all of the reviewers, we have added ablation studies. We have also added the continuous ranked probability score (CRPS) as an evaluation metric. These new results are included in the updated PDF of our submission with discussion, and they summarized in our responses to the relevant questions below. We are currently working on adding a new baseline, and we will add those results before the end of the rebuttal period.
>
> Additionally, while adding the ablation experiments, we noticed that in generating the human motion prediction results reported in the original manuscript, we accidentally omitted Line 10 of Algorithm 2 (i.e., the final pattern mixing step). After correcting our implementation, we found that using a maximum mixing scale of 1 (see Line 9 of Algorithm 2) provided too strong of a pattern signal in the final prediction. In our ablations, we therefore performed a sweep over maximum guidance scale $\bar{w} \in {0, 1, 2, 3, 4, 5}$ and maximum mixing scale in $\bar{w*} \in {0, 0.2, 0.4, 0.6, 0.8, 1.0}$. We have updated Table 1 to report the best performance over this range of $\bar{w}$ and $\bar{w*}$, which has led to a very minor improvement in our results on the AIST++ dataset. Overall, pattern guidance reduces PGDM’s prediction error by up to 14.12% (previously 11.10%), and PGDM achieves lower error by up to 82.60% (previously 82.54%). These updates have also been propagated to the tables in Appendix G.  Among our ablations, we additionally report sweeps over $\bar{w}$ and $\bar{w*}$. Our conclusions are not affected by this update, as the updated results show improved performance.
>
> In summary, the major changes to our PDF submission are as follows:
> * Updated Table 1 with corrected AIST++ results
> * Added probabilistic forecasting metric, continuous ranked probability score (CRPS), in Table 2 (Table 2 of previous version is now Table 3)
> * Added ablations for pattern mixing and dynamic scaling, discussed in Section 5.2 and fully reported in Appendix H
> * Minor edits for clarity and conciseness based on reviewer feedback and 10-page limit
>
> **Question 1**
>
> Indeed, patterns being present in the data is a stated assumption of PGDM. Under this assumption, PGDM leverages the present patterns to aid in forecasting and surpass baselines. While this assumption holds easily in many domains, including the clinical and motion forecasting domains we consider, PGDM may be more challenging to apply in other domains. For example, on very rich image data, which may contain more latent patterns obscured by the high number of dimensions, it may be more challenging to extract meaningful and representative patterns using archetypal analysis. In such cases, PGDM may be extended by first learning an embedding space, then applying PGDM on these extracted features. Effectively identifying when PGDM’s pattern assumption holds is another important future step for the practical use of PGDM.
>
> Additionally, we agree that we would ideally like to compare PGDM to baselines that similarly aim to leverage patterns. We have noted three related diffusion-based techniques (DIFF-MGR (Zhao, 2024), Westny et al. (2024), and Wang et al. (2024)) that use patterns for forecasting temporal data. Diff-MGR and Wang et al. formalize the notion of “patterns” through a method specific to their intended applications, predicting the trajectories of traffic and pedestrians, respectively. Westny et al. represents patterns as environment maps, and Wang et al. represents patterns as segmented images. As these techniques are not generally applicable across domains, we were unfortunately unable to evaluate them in our applications. Finally, the most similar related work to PGDM is DIFF-MGR, which assumes that the manifested patterns tend to remain constant over time. We were unfortunately unable to reproduce DIFF-MGR due to inaccessible code and insufficient implementation details. For these reasons, we have opted to instead compare PGDM to more general purpose forecasting techniques.

---

> > ### Author Response · Authors · 2025-11-21
> > **Response Part 2**
> >
> > **Question 2**
> >
> > We apologize for the confusion. We presented both the MAE-based and GDE-based models in an effort to better demonstrate the benefits of pattern guidance. While the MAE-based model achieves the best performance overall, we found that in many cases, the MAE-based model surpasses baselines even without guidance. To better demonstrate the benefits of pattern guidance, we therefore additionally included a trained version of PGDM selected based on the GDE criterion. We found that this model performed worse than baselines without guidance in a majority of cases, but can almost always outperform baselines with guidance. By showing the performance of the GDE-based model, we therefore further emphasize the benefits of guidance.
> >
> > To select the GDE-based model, we chose the trained PGDM model that achieved the highest capacity for pattern guidance (i.e., the greatest achievable improvement in MAE by using guided predictions over unguided predictions). In more detail, we selected the model that achieved the largest gain in MAE by adding any level of guidance in the range of 1 to 5.
> >
> > The definition and purpose of the GDE-based model is stated in Section 5, but we have updated our PDF with a clearer description.
> >
> > **Question 3**
> >
> > The dynamic guidance scale guards against uncertainties in the pattern representation. This may occur, for example, when the patterns extracted by archetypal analysis do not fully capture the patterns seen at inference time (i.e., distribution shift). The dynamic guidance scale ensures that, when novel patterns occur, the pattern guidance is not followed by the diffusion model.
> >
> > To support this, we have added an ablation comparing constant guidance scale to dynamic guidance scale. For both the pattern guidance and pattern mixing, PGDM applies a dynamic scale $w\in(0, \overline{w})$ and $w^\* \in(0, \overline{w^\*})$, respectively. In Table 19, we compare the MAE of PGDM$\_{\rm MAE}$ with a dynamic scale to that with a constant scale, holding $\overline{w}$ and $\overline{w^*}$ fixed. Across the UWHVF dataset and most genres of the AIST++ dataset, the dynamic scale reduces the prediction error of PGDM. On the ballet jazz and wack genres of AIST++, dynamic scaling performs comparably to constant scaling. It is likely that in these two cases, fewer novel patterns are seen at inference time. The dynamic scale, which is determined by the uncertainty of the pattern prediction, is most beneficial when the patterns extracted by archetypal analysis do not fully capture the patterns seen at inference time (i.e., distribution shift). The dynamic guidance scale ensures that, when novel patterns occur, the pattern guidance is not followed by the diffusion model. Similarly, the dynamic mixing scale ensures that pattern predictions on novel patterns are not heavily incorporated into the final forecast.
> >
> > *Table 19: Impact of dynamic guidance and mixing scales on PGDM$\_{\rm MAE}$ with fixed $\overline{w}$ and $\overline{w^\*}$. Mean absolute error (MAE) on UWHVF and all ten genres of the AIST++ dataset. Mean and standard deviation are taken across five samples.*
> > | | Constant Scale | Dynamic Scale |
> > |---|---|---|
> > |UWHVF |3.16±0.0000 |**2.96±0.0117**|
> > |Break |0.39±0.0010 |**0.38±0.0009**|
> > |House |0.77±0.0013 |**0.74±0.0014**|
> > |Ballet jazz |**0.38±0.0002**|**0.38±0.0002**|
> > |Street jazz |0.49±0.0010 |**0.48±0.0008**|
> > |Krump |0.74±0.0009 |**0.70±0.0013**|
> > |LA Hip Hop |0.77±0.0007 |**0.74±0.0006**|
> > |Lock| 0.69±0.0002 |**0.67±0.0006**|
> > |Middle Hip Hop |0.86±0.0009 |**0.82±0.0009**|
> > |Pop |0.47±0.0019 |**0.44±0.0018**|
> > |Wack| **0.39±0.0025** |**0.39±0.0028**|
> >
> > This ablation and discussion is included in Section 5.2 and Appendix H of our revised PDF.
> >
> > **Question 4**
> >
> > PGDM is intended for data with inherent patterns. One example domain where this assumption holds frequently is in clinical applications, due to the physical and recurring structures of the human body. In our VF example, specific patterns of loss reappear across patients, because they are caused by degradation of certain parts of the retinal structure. In some domains, even richer data can also exhibit patterns. For example, basketball videos also contain repeated structures. Standardized courts and common strategies appear across basketball games in different countries played by different teams. Thus, across videos, we can reliably expect features related to these patterns to appear. We provide this high level overview of applicable data types in the introduction.

---

> ### Author Response · Authors · 2025-11-21
> **Response Part 3**
>
> **Question 5**
>
> Indeed, AA is an existing technique. One of our unique contributions is our use of AA to propose a novel uncertainty quantification metric. To the best of our knowledge, only one other work has applied archetypal analysis to measure the uncertainty of a predictive model, a recent preprint by Philips et al., 2025. Intuitively, this method formalizes semantic content in LLM predictions as archetypal patterns. After sampling multiple predictions from an LLM, this technique applies archetypal analysis, then measures the volume of the resulting archetype hull and the entropy of the assigned archetype contributions. A high volume or entropy indicates high variability in the LLM predictions, and therefore high uncertainty. While Philips et al. can only be applied to a probabilistic model, our uncertainty quantification technique can be applied to deterministic models like our pattern guidance function.
>
> Furthermore, we use our uncertainty quantification technique in the loop to improve prediction quality. Our remaining contributions lie in the use of our uncertainty quantification metric as a principled method for tuning the level of pattern guidance. We provide Theorems 1 and 2 to support this design. Furthermore, PGDM uniquely proposes a guidance function to predict future patterns. The uncertainty quantification metric and guidance function together serve to better handle dynamically changing patterns and uncertain pattern representations, challenges not yet addressed by existing work leveraging patterns.
>
> **References**
>
> Philips, E., et al. Geometric uncertainty for detecting and correcting hallucinations in LLMs. ArXiv preprint, 2025.
>
> Weizhuo Wang, C Karen Liu, and Monroe Kennedy III. Egonav: Egocentric scene-aware human trajectory prediction. arXiv preprint arXiv:2403.19026, 2024.
>
> Theodor Westny, Björn Olofsson, and Erik Frisk. Diffusion-based environment-aware trajectory prediction. arXiv preprint arXiv:2403.11643, 2024.
>
> Tianlong Zhao, Guangle Song, Xuemei Li, Lizhen Cui, and Caiming Zhang. Diff-mgr: Dynamic causal graph attention and pattern reproduction guided diffusion model for multivariate time series probabilistic forecasting. Information Sciences, 675:120742, 2024.

---

> ### Author Response · Authors · 2025-12-03
> **Additional Response Part 1**
>
> We have added one additional baseline (ARMD (Gao, 2025)) and one more ablation study. For completeness, we have also added the CRPS metric and our PGDM$\_{\rm GDE}$ variant to the reported results of our ablation studies, as we reported only MAE of PGDM$\_{\rm MAE}$ in the ablation studies added in our last response. Our findings from our ablation studies are further supported by the reported CRPS and PGDM$\_{\rm GDE}$ results. Finally, we have added timing measurements for PGDM and our baselines.
>
> We report the evaluation of the new baseline below. In summary, PGDM outperforms ARMD in terms of both MAE and CRPS, by up to 93.64% and 92.55%, respectively. The greatest performance gap is on the motion capture frame prediction task. A likely explanation is that ARMD’s design is motivated by the linear ARMA model, which assumes that future data can be constructed as a linear combination of past data and noise terms. This assumption is well aligned with smooth, trend dominated time series (e.g., visual field progression), but limits ARMD’s expressiveness for the rapid, nonlinear dynamics of the AIST++ dancing data. In contrast, PGDM explicitly models pattern progression in a lower dimensional pattern representation space, allowing it to capture these nonlinear pattern dynamics with lower sample complexity.
>
> *Table: MAE and CRPS$\_{\rm SUM}$ of PGDM$\_{\rm MAE}$ and ARMD for both the visual field prediction (UWHVF dataset) human motion prediction (10 dance genres from AIST++ dataset) applications. For PGDM$\_{\rm MAE}$, we show the guidance and mixing scale result that achieved the lowest error (for $\overline{w}$ and $\overline{w^\*}$ choices, see Appendix H). Mean and standard deviation are taken across five and three samples for MAE and CRPS$\_{\rm SUM}$, respectively*
>
> ||ARMD (MAE) | PGDM$_{\rm MAE}$ (MAE) | ARMD (CRPS) | PGDM$_{\rm MAE}$ (CRPS)|
> |---|---|---|---|---|
> |UWHVF |3.91±0.0083| **2.96±0.0117** | 0.918±0.0029  | **0.777±0.0040**|
> |Break |4.26±0.0026 | **0.38±0.0009**| 0.490±0.0006 |**0.037±<0.0001**|
> |House |9.11±0.0035 | **0.74±0.0014**|0.679±0.0008 |**0.073±0.0002**|
> |Ballet Jazz |3.38±0.0011| **0.38±0.0002**|0.346±0.0005 |**0.035±0.0002**|
> |Street Jazz |7.57±0.0049 | **0.48±0.0008**|0.448±0.0004 |**0.047±0.0002**|
> |Krump |8.90±0.0046| **0.70±0.0013**|0.911±0.0003 |**0.071±0.0003**|
> |LA Hip Hop | 8.51±0.0032 | **0.74±0.0006**|0.752±0.0014 |**0.071±0.0005**|
> |Lock |7.65±0.0031 | **0.67±0.0006**|0.652±0.0014 |**0.063±0.0001**|
> |Middle. Hip Hop |8.98±0.0051 | **0.82±0.0009**|0.832±0.0005 |**0.081±0.0003**|
> |Pop| 6.43±0.0019 | **0.44±0.0018**|0.460±0.0001|**0.045±0.0002**|
> |Wack |3.39±0.0026 | **0.39±0.0028**|0.400±0.0012 |**0.039±0.0003**|
>
> The remaining new results are included in the updated PDF of our submission with discussion, and relevant results are summarized in our responses to the appropriate questions below.
>
> **Additional Response to Question 1:**
>
> As we noted earlier, we were unable to evaluate additional baselines that also leverage patterns, as these have domain specific pattern formalizations, inaccessible code, or insufficient implementation details. We have added, however, an ablation study to serve as a proxy for evaluating the most similar pattern-based diffusion forecaster to PGDM, which is Diff-MGR (Zhao, 2024). Diff-MGR assumes that patterns will remain constant over time, conditioning future predictions on past patterns. We therefore evaluate a version of PGDM that follows this same assumption. We compare PGDM without the pattern prediction model $f_A$ (i.e., predictions are conditioned only on historical patterns, rather than predicted future patterns) to that without $f_A$. Across UWHVF and all ten genres of the AIST++ dataset, PGDM with $f_A$ outperforms PGDM without $f_A$.  On UWHVF, the pattern prediction model adds a slight performance gain, as visual fields and their patterns may progress slowly over several decades (Saunders, 2016). In other cases, the performance gap from adding pattern prediction is more significant, e.g., the house, krump, and middle hip hop genres of AIST++. In these particular dance genres, patterns may change more rapidly due to dance style or music tempo. For example, the reported music tempo for the house genre is 110-130 BPM, compared to the 80-130 BPM tempo of the remaining genres (Li, 2021). This study highlights the importance of the pattern prediction model in accounting for the highly dynamic patterns that commonly appear in realistic temporal data.

---

> > ### Author Response · Authors · 2025-12-03
> > **Additional Response Part 2**
> >
> > For brevity, we report below the MAE of PGDM$\_{\rm MAE}$ for this ablation. However, the MAE and CRPS for both PGDM$\_{\rm MAE}$ and PGDM$\_{\rm GDE}$ are included with discussion in Section 5.2 and Appendix H of our revised PDF. Our conclusions remain consistent across both metrics and both variants of PGDM.
> >
> > *Table: Impact of dynamic scaling and pattern prediction model $f\_A$ on mean absolute error (MAE) of PGDM$_{\rm MAE}$. MAE is reported for UWHVF and all ten genres of the AIST++ dataset with fixed $\overline{w}$ and $\overline{w^\*}$. Mean and standard deviation are taken across five samples.*
> >
> > ||Without $f\_A$ | With $f\_A$|
> > |---|---|---|
> > |UWHVF | 2.99±0.0101 |**2.96±0.0117**
> > |Break | 0.41±0.0008 |**0.38±0.0009**
> > |House | 0.84±0.0019 |**0.74±0.0014**
> > |Ballet jazz |0.40±0.0006 |**0.38±0.0002**
> > |Street jazz |0.49±0.0005 |**0.48±0.0008**
> > |Krump | 0.92±0.0012 |**0.70±0.0013**
> > |LA Hip Hop | 0.81±0.0005 |**0.74±0.0006**
> > |Lock |0.75±0.0011 |**0.67±0.0006**
> > |Middle Hip Hop |0.94±0.0009 |**0.82±0.0009**
> > |Pop |0.46±0.0020 |**0.44±0.0018**
> > |Wack |0.41±0.0020 |**0.39±0.0028**
> >
> > **References:**
> >
> > Jiaxin Gao, Qinglong Cao, and Yuntian Chen. Auto-regressive moving diffusion models for time series forecasting. In Proceedings of the AAAI Conference on Artificial Intelligence, volume 39, pp. 16727–16735, 2025.
> >
> > Ruilong Li, Shan Yang, David A Ross, and Angjoo Kanazawa. Ai choreographer: Music conditioned 3d dance generation with aist++. In Proceedings of the IEEE/CVF international conference on computer vision, pp. 13401–13412, 2021.
> >
> > Luke J Saunders, Felipe A Medeiros, Robert N Weinreb, and Linda M Zangwill. What rates of
> > glaucoma progression are clinically significant? Expert review of ophthalmology, 11(3):227–234,
> > 2016.
> >
> > Tianlong Zhao, Guangle Song, Xuemei Li, Lizhen Cui, and Caiming Zhang. Diff-mgr: Dynamic causal graph attention and pattern reproduction guided diffusion model for multivariate time series probabilistic forecasting. Information Sciences, 675:120742, 2024.

---

### Author Response · Authors · 2025-12-03
**Summary of Rebuttals**

We would like to thank the reviewers for their valuable feedback, based on which we have significantly improved our submitted paper. We believe we have comprehensively addressed the reviewers’ concerns, supported by Reviewer KRkT’s response and Reviewer qMFo’s score increase from 4 to 6.

In summary, we have added:
* A **new metric, continuous ranked probability score (CRPS)**. This added metric reinforces our conclusions and highlights that explicit pattern modeling improves point prediction even when the distribution itself is challenging to estimate.
* An additional **recent baseline, ARMD (Gao, 2025)**. PGDM outperforms ARMD across all settings and metrics, by **up to 93.64% in MAE and up to 92.55% in CRPS**.
* An **ablation demonstrating the benefits of pattern mixing**. Our results indicate that PGDM’s explicit pattern modeling is particularly beneficial in low-data regimes.
* An **ablation demonstrating the benefits of dynamic guidance scaling**. Dynamic scaling likely yields the greatest improvements over constant scaling when novel patterns are seen inference time.
* An **ablation demonstrating the benefits of the pattern prediction model**. PGDM benefits from pattern prediction most significantly in domains where patterns change rapidly over time. This also serves as a proxy for comparing PGDM to Diff-MGR (Zhao, 2024), which assumes patterns are constant over time, and which we have been unable to evaluate due to unavailable code and insufficient implementation details.
* **Parameter counts and timing measurements** for both PGDM and baselines.
* **Revised PDF** with added results, further discussion, and improved clarity based on reviewer questions.

For temporal data with inherent patterns, PGDM provides a principled method of leveraging these patterns for forecasting future data. Motivated by our theoretical contributions, we introduce a novel uncertainty quantification metric, which we use to dynamically scale the impact of predicted future patterns on PGDM’s forecast. Our added baseline and metric reinforce PGDM’s competitive performance. Furthermore, our added ablations demonstrate the advantages of PGDM in handling dynamically changing patterns and uncertain pattern representations, challenges not yet addressed by existing pattern-based diffusion forecasters.

**References:**

Jiaxin Gao, Qinglong Cao, and Yuntian Chen. Auto-regressive moving diffusion models for time series forecasting. In Proceedings of the AAAI Conference on Artificial Intelligence, volume 39, pp. 16727–16735, 2025.

Tianlong Zhao, Guangle Song, Xuemei Li, Lizhen Cui, and Caiming Zhang. Diff-mgr: Dynamic causal graph attention and pattern reproduction guided diffusion model for multivariate time series probabilistic forecasting. Information Sciences, 675:120742, 2024.

---

### Meta-Review · Area_Chair_6qnD · 2026-01-05

**Summary:**

Reviewers raise overlapping concerns. In particular, they note that the experimental evaluation is limited in scope, with a small number of datasets and relatively weak or outdated baselines, and lacks sufficient ablation studies to isolate the contributions of key components such as the archetypal space, weighting strategy, and dynamic guidance mechanism. In addition, the reviewers find that the methodological novelty is limited, as the core idea of incorporating pattern-based or predictive signals into diffusion models via guidance is closely related to existing classifier-free or conditional diffusion approaches. While the use of Archetypal Analysis is reasonable, the paper does not sufficiently articulate why this design is principled or how it advances the state of the art beyond a stylistic combination of known techniques.

During the rebuttal phase, the authors provided substantial additional experimental results and addressed several reviewer concerns, which led some reviewers to increase their scores. However, after considering the overall evaluations, the final scores still do not reach the acceptance threshold, and therefore the paper cannot be recommended for acceptance.

**Reviewer Concerns:**

Reviewer KRkT and Reviewer qMFo's concerns are addressed, Reviewer LqcP and Reviewer NBP2's concerns are still outstanding.

**Reviewer Scores:**

Reviewer KRkT and Reviewer qMFo ultimately provided clearly positive ratings. Reviewer LqcP and Reviewer NBP2 did not respond during the discussion period; however, I believe they might have increased their scores by 0–2 points had they been able to participate fully in the discussion.

---

### Decision · Program_Chairs · 2026-01-26

Reject